# Accessing acute care hospitals in the San Francisco Bay Area after a major hayward earthquake

Luis Ceferino [1,2] ✉, Charan Kukunoor [2], Jinyan Zhao [1], Dan Mao[3], Xinlu Xu[3], Jingzhe Wu[4] & Adam Zsarnóczay[5]

Earthquakes can severely disrupt healthcare access, especially in dense cities. Here, we provide a comprehensive assessment of how a magnitude 7.25 earthquake on the Hayward Fault would impact access to acute care hospitals in the San Francisco Bay Area. By integrating seismic hazard with hospital and transportation infrastructure's vulnerability and connectivity data, we analyze 76 hospitals (426 buildings with 16,639 beds) and 5163 bridges within a vast network of ~1.5 million edges and ~0.5 million nodes. We leverage the rich data to formulate a coupled risk-network model to quantify simultaneous failures and cascading disruptions across the healthcare and transportation systems. Our results revealed that hospital bed capacity could drop to 51%, with Alameda County retaining only 20%. Widespread transportation failures further restrict access, increasing regional travel times by 177% and exceeding 1000% in parts of East Bay, potentially fully isolating a hospital and an entire urban community. These findings underscore the urgent need for resilient healthcare and transportation infrastructure to mitigate life-threatening disruptions following major earthquakes.

Earthquakes wield the power to severely disrupt healthcare systems, placing vulnerable populations at significant risk[1]. In the 2023 M 7.8 Türkiye earthquake, more than 50 hospitals were damaged, leaving tens of thousands without access to medical care[2]. In other major earthquakes—such as 2005 Pakistan, 2010 Haiti, 2008 China, 2011 Japan, and 2023 Türkiye—thousands of injured individuals experienced worsening conditions, sometimes leading to fatal outcomes, due to delays in receiving medical attention[3–10]. Healthcare disruptions can persist long after the disaster: following the 2003 Bam earthquake in Iran, access to critical services like dialysis remained limited for up to five years[11,12].

Transportation failures further compound these healthcare disruptions. The 2008 Wenchuan earthquake in China and the 2010 Haiti earthquake demonstrated how disrupted transportation networks delayed or prevented access to emergency medical care[13–15]. In the U.S., the 1989 M 6.9 Loma Prieta earthquake damaged 91 state highway bridges and forced 13 closures, severely disrupting hospital access across the Bay Area.

Even moderate earthquakes in the U.S. have led to critical healthcare system failures, prompting major policy reforms. The 1971 M 6.6 San Fernando Earthquake caused the structural collapse of two major hospitals in the San Fernando Valley—most notably the Veterans Administration Hospital, where more than 40 people died, and Olive View Medical Center—alongside damage to other medical buildings[16]. This prompted the Alfred E. Alquist Hospital Seismic Safety Act of 1973, mandating that hospitals remain functional after disasters—but applying only to new construction. In 1994, the M 6.7 Northridge Earthquake forced evacuations at eight acute care hospitals, and

[1]Department of Civil and Environmental Engineering, University of California, Berkeley, Berkeley, California, USA. [2]Department of Civil and Urban Engineering, New York University, Brooklyn, NY, USA. [3]Center for Urban Science and Progress, New York University, Brooklyn, NY, USA. [4]The Global Facility for Disaster Risk Reduction, the World Bank Group, Washington, District of Columbia, USA. [5]Department of Civil and Environmental Engineering, Stanford University, Stanford, CA, USA. ✉e-mail: ceferino@berkeley.edu

rendered twelve pre-Alquist hospital buildings unsafe[17–19]. Though newer hospitals experienced less structural damage, non-structural failures were still extensive. These events led to Senate Bill 1953, requiring seismic retrofits of acute care hospitals by 2030[20,21]. Yet, many hospitals face financial barriers to meeting these mandates [22].

Past research has examined hospital seismic risk primarily at the building scale. Studies have used structural engineering methods, fault-tree analyses, and performance-based assessments to evaluate damage in isolated facilities[23–26]. Others have used flow models and discrete-event simulations to explore patient care delays and medical bottlenecks in single-hospital scenarios[27–30]. While valuable, these studies often overlook how failures can propagate across urban systems.

In parallel, a growing body of work has started to model access to hospital services at the regional scale. Studies in Lima, Peru[31,32], and Butte County, California[33], have shown that even small reductions in overall healthcare capacity can significantly affect local access to care. More recent efforts have begun to jointly consider disruptions to both healthcare and transportation systems, e.g., in Istanbul, Türkiye[34], and in a synthetic medium-sized city in China[35]. We contribute to this growing literature by modeling these compounding effects at scale across the San Francisco Bay Area using detailed, real-world datasets on hospital and bridge infrastructure.

Recent earthquakes in Chile, New Zealand, and Japan have also underscored the importance of treating hospitals as part of broader interdependent systems[24,36–39]. Failures at one facility can cascade across a regional network—overwhelming nearby hospitals, increasing patient travel distances, and worsening health outcomes. To support resilience planning across large metropolitan areas, there is a need for region-wide models grounded in real exposure data that capture operational interdependencies between healthcare and transportation infrastructure[40–42]. Developing such models remains challenging due to limited data, computational demands, and regional variation in infrastructure.

In this work, we evaluate post-earthquake healthcare access across the entire San Francisco Bay Area by simulating joint disruptions in healthcare and transportation systems. We integrate seismic risk models with regional-scale network analysis using publicly available data on 426 hospital buildings and 5163 bridges across a transportation network comprising approximately 1.5 million edges and 0.5 million nodes. Our study contributes to the growing field of integrated disaster risk modeling by providing one of the most extensive simulations to date of interdependent failures affecting healthcare access after an earthquake. Although grounded in the San Francisco Bay Area, our study highlights systemic barriers to accessing healthcare after earthquakes under conditions common to many dense urban regions—such as high seismic risk, aging infrastructure, complex emergency response systems, and population disparities. These findings underscore the need for integrated planning approaches that account for infrastructure interdependencies and can inform resilience policy in similarly exposed metropolitan areas.

## Results

The Bay Area is home to more than 7M people in Northern California, and as a major city, it has a large demand for healthcare. The region's 76 acute care hospitals provide critical inpatient care and specialized medical services for surgery, acute conditions, and injuries, making them essential for disaster response.

Much of the Bay Area's healthcare infrastructure faces significant seismic risks due to proximity to major active faults. Most hospitals lie near the San Andreas and the Hayward Fault, where major earthquakes (>7.0) can occur (Fig. 1). The Laguna Honda Hospital and Rehabilitation Center and the University of California, San Francisco (UCSF) Medical Center are the largest facilities, with 780 and 580 beds. Both are located in San Francisco, less than 2 km from each other, highlighting

the concentration of critical medical resources in geographically small areas. The three zip codes with the most beds are Palo Alto, San Francisco (where the two largest hospitals are), and San Jose, with 1410, 1360, and 932 beds, respectively.

### Acute Care Vulnerability in the San Francisco Bay

We found that a significant part of the acute care portfolio is seismically vulnerable due to structural or non-structural deficiencies (Fig. 1). To quantify these vulnerabilities, we compiled information from 426 buildings belonging to the 76 acute care hospitals in the Bay Area, including structural typologies, year of construction, number of stories, and seismic vulnerability ratings[21,43,44]. Supplementary Note 1 and Supplementary Fig. 1 describe and summarize the hospitals' years of construction, structural typologies, and the number of stories. The Methods section establishes how to use this information to model seismic vulnerability, as these building features indicate buildings' dynamic properties, strength, and ductility. In this paper, we used the Structural Performance Categories (SPC) and Non-structural Performance Categories (NPC), established and reported by California's Department of Health Care Access and Information (HCAI), to characterize structural and non-structural deficiencies[45]. Each hospital building receives vulnerability ratings (SPC or NPC) ranging from 1 to 5, from the most to the least vulnerable (see full descriptions in HCAI documentation[45]).

Structural vulnerabilities are pronounced across the portfolio. (Fig. 1). Sixteen (4%) hospital buildings have an SPC of 1. All these buildings, constructed before 1974, face a higher collapse risk due to earthquakes than regular modern buildings (e.g., those designed for life safety in 475-year return period events)[46]. None of these buildings were supposed to provide acute care services by 2020. An additional 87 (20%) hospital buildings have an SPC of 2; of these, 65 were built before 1974 and 20 between 1974 and 1994. SPC-2 buildings meet pre-1973 standards for regular buildings but not those for hospitals, requiring upgrading by 2030[47]. Only buildings rated SPC-3 or above may operate as acute care facilities after this deadline. Only buildings rated SPC-3 or above may operate as acute care facilities after this deadline. Buildings with an SPC-5 rating—163 buildings (38%), predominantly constructed after 1994—meet essential-facility standards under modern codes[46]; thus, they can remain operational immediately after 475-year return period events.

Non-structural vulnerabilities are even more widespread. Over half of the buildings (220, 52%) have an NPC rating of 2 or lower, with most constructed before 1994 (Fig. 1). NPC-2 buildings have anchorage and bracing only for limited components, typically for basic building access, and were mandated to be upgraded by 2002 to support acute care services. Buildings rated NPC-4 (131 buildings, predominantly built after 1994) have comprehensive anchorage and bracing for all non-structural components, meeting the standard required for acute care facilities after 2030. Only two buildings currently meet the stringent NPC-5 criteria, requiring additional onsite provisions for continuous 72-hour acute care operations. Even facilities meeting modern structural codes may not attain an NPC-5 rating due to the presence of vulnerable external resources, such as water tanks, necessary for sustained hospital operations during emergencies.

### Earthquake Scenario and Projected Building Damage

We studied an earthquake scenario of M 7.25 on the Hayward Fault in East Bay (Fig. 2), a fault that has accumulated energy for over more than a century. Its last major earthquake (M 6.8, 1868) caused extensive damage across the region. Our selected scenario magnitude is based on an established reference case widely used in resilience policymaking for the Bay Area[48].

We modeled the rupture geometry (see Methods) and found that 10 acute care hospitals with 2167 beds (13% of the Bay's total) are within just 5 km of the projected rupture. Our shaking intensity estimates

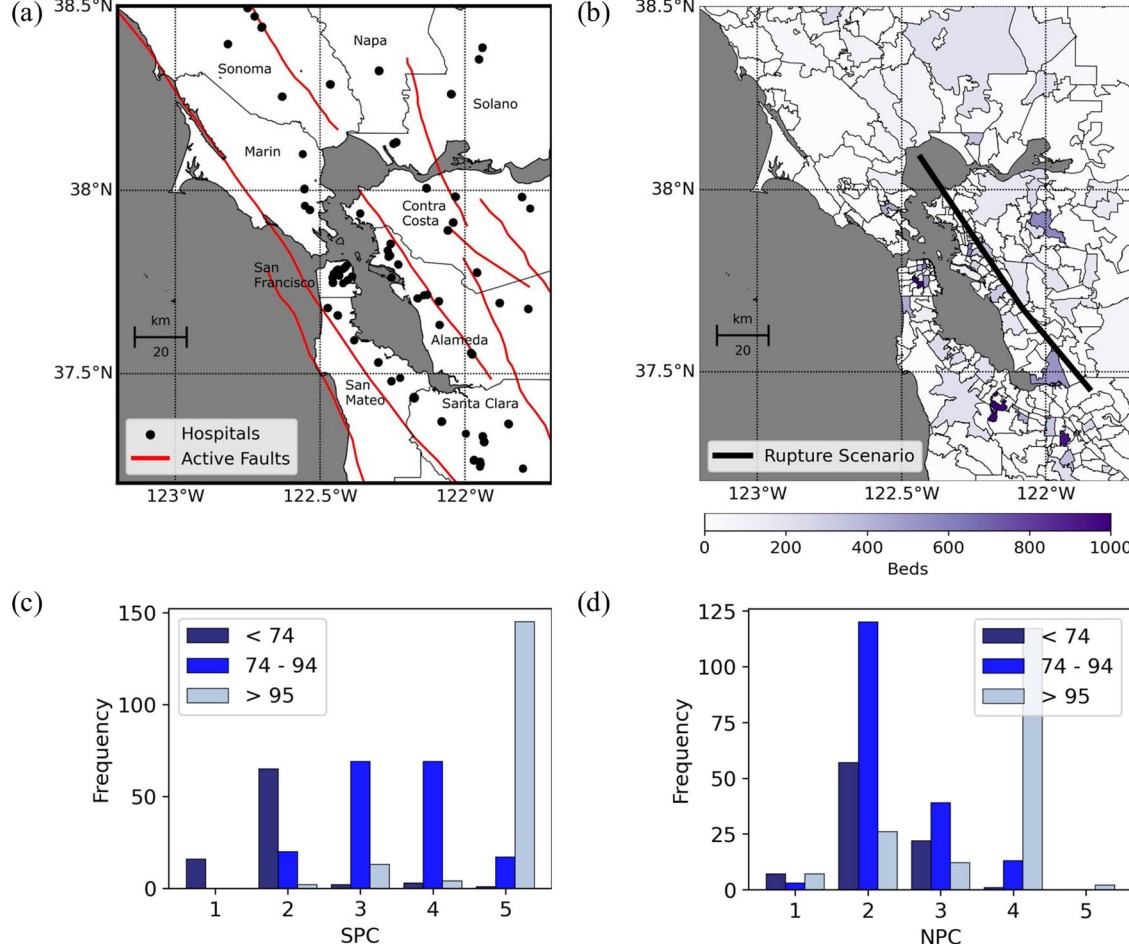

**Fig. 1 | Seismic hazard and hospital exposure and vulnerability in the San Francisco Bay Area, California.** The Bay Area's county names are also included for reference. **a** Acute care hospital inventory and all major active faults. Map data from https://www.openstreetmap.org/copyright OpenStreetMap ©contributors. **b** Bed count per zip code and M 7.25 rupture scenario on the Hayward Fault. Map data from https://www.openstreetmap.org/copyright OpenStreetMap ©contributors. **c** Histogram of hospital buildings' structural performance category (SPC) and their years of construction. **d** Histogram of hospital buildings' non-structural performance category (NPC) and their years of construction.

indicate that many hospitals could experience levels of ground motion not observed in recent earthquakes (See Methods). On average, our analysis predicts 51 hospitals will experience peak ground accelerations above 0.2g (Fig. 2). By comparison, the 1989 M 6.9 Loma Prieta Earthquake—previously the strongest since the devastating 1906 M 7.9 San Francisco Earthquake—subjected only 14 hospitals to similar shaking intensities (Fig. 2). Supplementary Fig. 2 further contextualizes these findings, illustrating how recent events such as the 2014 Napa Earthquake (M 6.0) exposed just three hospitals to significant shaking (>0.2g). In contrast, the historic 1906 earthquake exposed nearly the entire portfolio (72 out of 76 hospitals).

We then estimated structural and non-structural damage across all 426 acute care hospital buildings, as both types of damage critically disrupt continuous hospital operations. We employed damage thresholds as tipping points to model potential service interruptions at various levels of severity (see Methods). For simplicity, we refer to the probability of exceeding these thresholds as the "probability of damage" or "probability of failure." Our findings reveal extensive potential damage: 214 buildings (50%) have greater than a 25% probability of structural damage, and 254 buildings (60%) exceed this threshold for non-structural components. Notably, 71% and 94% of these at-risk buildings lie within 20 km of the Hayward Fault or have low SPC/NPC ratings (Supplementary Fig. 3, Supplementary Note 3).

## Post-earthquake hospital capacity

Our risk analysis predicts a substantial loss of hospital capacity throughout the Bay Area following the earthquake scenario (see Methods). We estimate a loss of 8165 hospital beds, resulting in only 51% of total acute care beds remaining functional (standard deviation: 21%). This significant reduction in capacity is highly uneven across the region (Fig. 3, Table 1). Our baseline predictions assume hospitals lose functionality when structural or non-structural damage exceeds a threshold of slight damage (see Methods). These damage conditions are often shown to disrupt hospital services[24,36–39]. See descriptions of multiple damage levels in HAZUS Earthquake Model Technical Manual[49]. We also analyzed more optimistic scenarios with moderate ("favorable") and extensive ("idealistic") damage thresholds, resulting in increased overall functionality of 79% and 93%, respectively (Fig. 3 and Supplementary Table 1). However, empirical evidence supports our baseline assumption, as hospitals often lose operations at early stages of damage [24,36–39].

Alameda County faces the most severe impacts due to its proximity to the Hayward Fault and the concentration of vulnerable hospitals (Table 1). Bed functionality there would drop dramatically, from 3221 to only 651 functional beds (20%, standard deviation: 19%). As Alameda houses 1.6 million residents—making it the Bay Area's second most populous county—this represents a critical healthcare disruption for numerous communities. Marin County, situated northward near

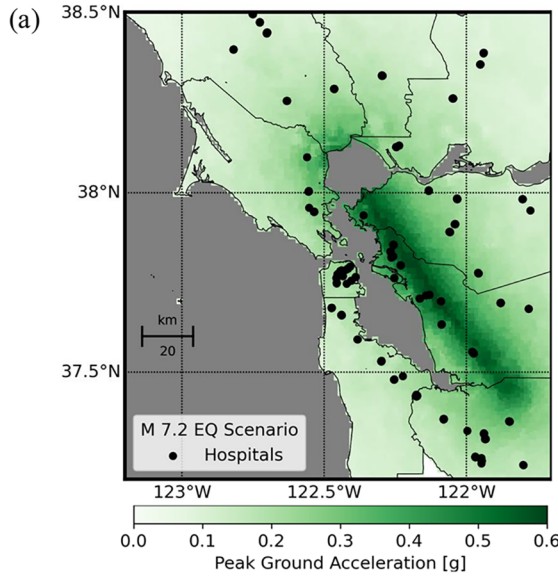

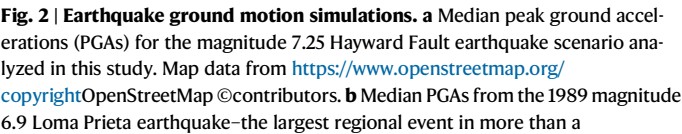

**Fig. 2 | Earthquake ground motion simulations. a** Median peak ground accelerations (PGAs) for the magnitude 7.25 Hayward Fault earthquake scenario analyzed in this study. Map data from https://www.openstreetmap.org/copyrightOpenStreetMap ©contributors. **b** Median PGAs from the 1989 magnitude 6.9 Loma Prieta earthquake–the largest regional event in more than a century–highlighting that current infrastructure has not experienced shaking intensities projected for the Hayward scenario. Map data from https://www.openstreetmap.org/copyrightOpenStreetMap ©contributors. We used 5000 Monte Carlo simulations to conduct the analysis.

the fault rupture, will be the second-most impacted, retaining only 45% of its original capacity. However, with a significantly smaller population (251,000 residents), Marin's absolute impact is lower, further highlighting Alameda's critical vulnerability.

Hospitals within 20 km of the fault retain, on average, only 25% functionality, compared to 51% at 20-40 km and 82% beyond 40 km (Supplementary Fig. 4, Supplementary Note 4). Facilities with over half their buildings rated structurally poor (SPC ≤2) have 14% lower functionality (40% vs. 54%), and those with predominantly poor non-structural ratings (NPC ≤ 3) have 17% lower functionality (45% vs. 62%). Structural damage dominates near the fault, but non-structural damage remains critical at greater distances (64% structural vs. 70% non-structural damage within 20 km; 30% vs. 38% beyond 40 km).

### Post-earthquake Bridge Functionality

Our risk analysis model predicts that 1469 bridges out of 5163 would be damaged due to the earthquake scenario (see Methods). Consequently, 3693 bridges (72%, standard deviation: 17%) are expected to remain undamaged and fully operational (Fig. 3). Some damaged bridges could continue operating, albeit at reduced capacities. Under our "baseline scenario", where only bridges with at most slight damage remain operational, a total of 3964 bridges (77%, standard deviation: 15%) would function. In a more optimistic "favorable scenario", where bridges with at most moderate damage also remain partially operational, 4069 bridges (79%, standard deviation: 14%) could provide service. In an idealistic scenario, which assumes that bridges even with extensive damage retain emergency functionality[50], 4527 bridges (88%, standard deviation: 11%) would remain operational.

Similar to hospitals, Alameda County faces the most severe transportation impacts (Fig. 3, Table 2). Under the baseline scenario, Alameda retains only 44% functional bridges (from 642 down to 282). Marin County is second most impacted, retaining 64% (from 195 to 125 bridges). Additionally, estimates of reductions in bridge travel speeds[50] (see Methods)–calculated from their damage probabilities–highlight Alameda's higher vulnerability due to proximity to the Hayward Fault rupture (Fig. 3, Table 2).

### Acute Care Accessibility

Our results show that the earthquake scenario will radically change healthcare access in the Bay. We coupled the risk model for the hospital and bridge portfolio with a network model to assess the cascading effects of earthquakes on healthcare access across the Bay (see Methods).

We evaluated accessibility based on travel times to functioning hospitals. Across the Bay Area, average travel times to the nearest operational acute care hospital increase from 6.1 minutes pre-earthquake to 16.9 minutes post-earthquake–a 177% increase. However, this impact varies significantly by county. Alameda County, the second most populous (1.6M residents), experiences the most drastic increase of 407%, from 5.1 to 25.6 minutes (Table 3). Marin and Contra Costa counties follow, with increases of 314% and 147%, respectively, demonstrating widespread disruption to healthcare access.

At a more detailed neighborhood scale, these disruptions are even more pronounced (Fig. 4). We identified six densely populated zip codes (each exceeding 15,000 residents, labeled #1 to #6 in Fig. 4) facing substantial reductions in healthcare access. The most severely impacted is Novato (#1, Marin County), where travel times increase nearly 25-fold, from 7.3 to 185.5 minutes, affecting approximately 18,000 residents. Similarly, a Fremont zip code (#6, Alameda County) with roughly 66,000 residents experiences an eightfold increase (from 8.4 to 42.2 minutes). Other significantly impacted zip codes in Richmond (#4), Oakland (#2 and #5, Alameda County), and San Jose (#3, Santa Clara County) face travel time increases of 10, 11, 9, and 11 times their pre-earthquake levels, respectively. These results underscore the dramatic reshaping of healthcare access throughout East Bay communities along the earthquake rupture.

Finally, we explored the relative contributions of hospital and transportation infrastructure disruptions through scenario analyses (Fig. 4). We compared the baseline (None, no infrastructure disruptions) against scenarios modeling probabilistic damage only to transportation infrastructure (Transp., hospitals fully resilient), only to hospital infrastructure (Hosp., transportation fully resilient), and simultaneous probabilistic disruptions to both (Transp. & Hosp.). When considering disruptions only to transportation infrastructure (Transp.), average Bay Area travel times increase by 41% (from 6.1 to 8.6 minutes), notably less

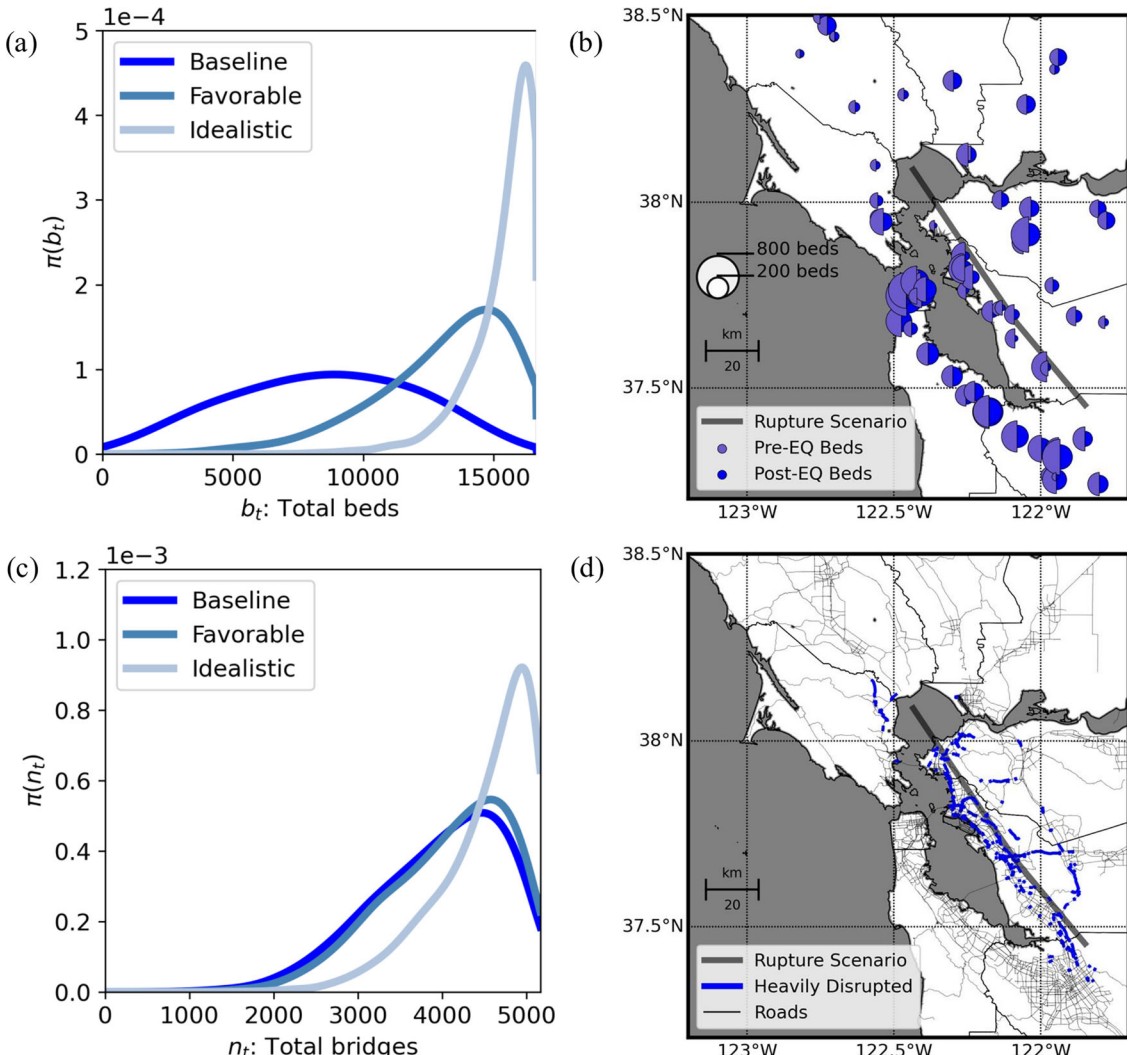

**Fig. 3 | Post-earthquake functionality of hospitals and bridges across the Bay Area. a** Probability distribution of functional acute care beds under baseline (slight), favorable (moderate), and idealistic (extensive) damage thresholds. **b** Spatial distribution of acute care beds before and after an earthquake scenario using the baseline (slight) damage threshold. The size of the circles indicates the number of functional beds before and after the earthquake. Map data from https:// www.openstreetmap.org/copyrightOpenStreetMap ©contributors. **c** Probability distribution of fully functional bridges under baseline (slight), favorable (moderate), and idealistic (extensive) damage thresholds. **d** Spatial distribution highlighting bridges with post-earthquake travel capacities below 75%. Map data from https://www.openstreetmap.org/copyrightOpenStreetMap ©contributors. Results generated from 5000 Monte Carlo simulations.

than the combined scenario (Transp. & Hosp., 177% increase). Conversely, modeling only hospital infrastructure disruptions (Hosp.) yields a 79% increase (from 6.1 to 10.9 minutes), highlighting greater relative fragility in hospital networks.

At localized urban scales, the compounding effects of simultaneous transportation and hospital infrastructure failures lead to disproportionately severe impacts (Fig. 4). for example, as mentioned earlier, in the most severely impacted zip code (novato, labeled #1 in Fig. 4), travel times dramatically increase from 7.3 to 185.5 minutes under combined disruptions (transp. & hosp.), compared to only 14.1 minutes when transportation disruptions are excluded (hosp. only). similarly, for other highly affected zip codes (labeled #2 to #6 in Fig. 4), travel-time ratios notably decrease—from 11, 11, 10, 9, and 8 under combined disruptions, to 6, 3, 6, 5, and 3 when transportation disruptions are not considered—highlighting the critical influence of compounding infrastructure failures on healthcare access.

### Isolated Hospitals and Communities
To understand the drivers behind the sharp increases in travel times observed earlier, we analyzed travel volumes to the closest acute care

hospitals before and after the earthquake. Fig. 5 shows pre-earthquake road usage, with line thickness representing travel volume. Roads in brown highlight those whose usage falls above the 90th percentile based on pre-earthquake conditions, emphasizing the most critical travel routes. As expected, the highest-usage roads prior to the earthquake were located near hospitals, especially in densely populated areas with fewer hospitals. For instance, communities in Fremont (#6 in Figs. 4 and 5) are served by only two hospitals, while Richmond (#4) relies on just one.

Figure 5 also displays post-earthquake road usage, using the same pre-earthquake 90th percentile threshold to highlight heavily traveled roads. After the earthquake, patient travel patterns shift significantly, with many communities forced to travel longer distances along major corridors. For example, residents in Oakland (#2 and #5 in Figs. 4 and 5), who previously relied on nearby hospitals, will now have to travel west to the San Francisco Peninsula or east to Walnut Creek due to simultaneous disruptions in both hospital and transportation infrastructure. These shifts help explain the steep increases in travel times observed in East Bay communities along the rupture zone.

**Table 1 | Predicted post-earthquake hospital bed capacities across Bay Area counties under a threshold of slight damage (baseline scenario)**

| County | Pre-earthquake Beds | Post-earthquake Beds | |
| --- | --- | --- | --- |
| | | Mean | Std. Dev. |
| Alameda | 3221 | 651 (20%) | 621 (19%) |
| Contra Costa | 1749 | 878 (50%) | 481 (28%) |
| Marin | 627 | 280 (45%) | 209 (33%) |
| Napa | 351 | 248 (71%) | 89 (25%) |
| San Francisco | 3618 | 2284 (63%) | 950 (26%) |
| San Mateo | 1444 | 944 (65%) | 367 (25%) |
| Santa Clara | 4323 | 2277 (53%) | 1069 (25%) |
| Solano | 722 | 476 (66%) | 179 (25%) |
| Sonoma | 584 | 436 (75%) | 124 (21%) |
| Total | 16,639 | 8474 (51%) | 3567 (21%) |

Numbers in parentheses represent the percentage relative to pre-earthquake capacities. Supplementary Table 1 presents results for thresholds of moderate (favorable scenario) and extensive (idealistic scenario) damage.

Four major bridges connect the East and West Bay: the Richmond-San Rafael, Oakland-San Francisco Bay, San Mateo-Hayward, and Dumbarton bridges. Before the earthquake, no patients needed to cross these bridges to reach their nearest acute care hospital (Fig. 5). After the earthquake, however, these bridges become critical, with travel volumes increasing to 3.6, 9.3, 6.3, and 13.6 times the pre-earthquake 90th percentile values—reflecting the large-scale redistribution of healthcare demand.

Our analysis also revealed a more complex post-disaster phenomenon: micro-scale isolation of a hospital and a community due to localized bridge failures. The hospital serving the most severely impacted zip code in Novato (#1 in Figs. 4 and 5), which has 64 beds, depends entirely on a single bridge for access. If this bridge fails, the hospital becomes inaccessible, increasing travel times from 7.3 to 185.5 minutes for the surrounding population.

More concerning, we found that an urban neighborhood near Fremont (#6) could become fully isolated (Fig. 5). In one area, failure of three bridges would trap one part of the neighborhood; in another, a different set of three bridge failures would isolate the remaining part. This community would lose access not only to hospitals but also to other essential services such as grocery stores and pharmacies—illustrating how cascading infrastructure failures can sever lifeline access in disaster scenarios.

## Discussion

We present a modeling framework that integrates probabilistic risk analysis with network modeling to study disruptions to acute care access after earthquakes. Using detailed data on 76 hospitals (426 buildings) and 5163 bridges, we assess infrastructure risks and interdependencies across Bay Area communities (see Methods).

First, we find that hospital services will face significant disruption. Nearly half of all acute care beds (49%) could be lost due to building damage, reducing regional capacity from 16,639 to 8474 beds. In Alameda County, that Fig. drops to just 20% of pre-earthquake capacity (from 3,221 to 651 beds). These results point to the urgent need for preparedness. While structural damage may render many buildings unusable, past earthquakes show that medical staff can adapt. For instance, Christchurch Hospital in New Zealand moved triage to the parking lot after the 2011 earthquake[51]. Similarly, during the 2023 Türkiye earthquake, personnel at Mustafa Kemal University Hospital relocated operations from upper floors to the ground floor and later to outdoor spaces[52]. If hospitals in vulnerable counties like Alameda prepare in advance—e.g., securing water and power supply to exterior

areas—they could partially recover functionality in an emergency. This rapid recovery can be lifesaving, especially for communities near the rupture that may see high numbers of severely injured patients.

Second, our findings underscore the need to retrofit both structural and non-structural components. While most retrofitting efforts have focused on structural safety, non-structural failures often drive hospital outages. In 2011, Christchurch Hospital remained structurally intact but lost ICU and radiology services due to generator damage[36,51]. Similarly, 80% of hospitals surveyed after the 2016 Kumamoto Earthquake reported water system failures that disrupted dialysis and sterilization, despite minimal structural damage[38,39]. California's mandate to retrofit hospitals with SPC ≤ 2 and NPC ≤ 3 by 2030 is ambitious—and needed—but difficult to achieve. Currently, 24% of buildings fall below the SPC threshold and 69% below the NPC threshold[22]. If the 2030 goal proves unrealistic, targeting retrofits for hospitals serving large or vulnerable populations could offer a more strategic path to resilience.

Third, access to acute care will become significantly more uneven. On average, travel time to the nearest functional hospital increases by 177%, from 6.1 to 16.9 minutes. Alameda County will see the sharpest increases—up 407% overall—with some zip codes experiencing 10 to 20 times longer travel times. These delays are critical. Patients with severe fractures or crush syndrome—a common injury in collapsed buildings—require immediate care, including x-rays, surgeries, or dialysis, all typically found in acute care hospitals. For many in Alameda, these services will be significantly harder to reach, which could have deadly consequences.

Fourth, the earthquake will reveal the deep interdependence between the hospital and transportation systems. Many communities will be forced to cross major bridges—such as the Richmond-San Rafael, Bay, San Mateo-Hayward, or Dumbarton—to access care, despite having relied on local hospitals pre-earthquake. Our analysis shows that hospital access hinges on the functionality of surrounding bridges. These critical connectors should be retrofitted to the same seismic standards as hospitals. In some cases, bridge failures could fully isolate neighborhoods—cutting off access not only to hospitals but also to basic services like food. While our model focuses on structural failures, additional disruptions—such as debris-blocked roads, fuel shortages, and emergency closures—could further exacerbate access losses. These findings highlight the need for a system-level approach to infrastructure planning—one that prioritizes interdependencies and regional importance rather than treating assets in isolation. Retrofitting the most critical bridges could greatly reduce the cascading effects of an earthquake on healthcare access.

## Methods

### Risk Formulation: From single to multiple infrastructure units

We utilize an extension of the performance-based earthquake engineering (PBEE) framework, initially established to assess earthquake consequences in an infrastructure unit, and analyze multiple hospital buildings[53–55]. Under Markovian (conditional independency) assumptions described in canonical PBEE formulations[54,55], we find the probability distribution of an earthquake consequence, e.g., economic losses or fatalities, as

$$P_{DV,DS,IM}(dv, ds, im) = P_{DV|DS}(dv|ds)P_{DS|IM}(ds|im)P_{IM}(im), \quad (1)$$

where $P()$ is a probability distribution (or mass) function, and $DV$ is a random variable representing an earthquake consequence (also called a decision variable). For example, $DV$ can track repair costs in buildings. In this case, $DV$ will be a positive number with an upper bound of $dv_u$, i.e., the total replacement cost of the building. In other applications, $DV$ has a different variable space, e.g., $DV \in \mathbb{Z}$ for the number of injured people in a building[9,10]. $DS$ is an ordinal random variable that evaluates structural damage in an infrastructure unit, and

**Table 2 | Predicted post-earthquake functionality of bridges across Bay Area counties under the baseline scenario (bridges with at most slight damage remain functional)**

| County | Pre-earthquake Bridges | Post-earthquake Bridges | | Post-earthquake Capacity | |
|---|---|---|---|---|---|
| | | Mean | Std. Dev. | Mean | Std. Dev. |
| Alameda | 642 | 282 (44%) | 147 (23%) | 52% | 23% |
| Contra Costa | 604 | 407 (67%) | 114 (19%) | 73% | 17% |
| Marin | 195 | 125 (64%) | 46 (23%) | 71% | 22% |
| Napa | 158 | 135 (86%) | 27 (17%) | 89% | 14% |
| San Francisco | 126 | 102 (81%) | 21 (17%) | 84% | 15% |
| San Mateo | 346 | 280 (81%) | 64 (19%) | 85% | 16% |
| Santa Clara | 946 | 648 (69%) | 215 (23%) | 75% | 20% |
| Solano | 360 | 319 (89%) | 49 (14%) | 91% | 11% |
| Sonoma | 590 | 531 (90%) | 74 (13%) | 92% | 10% |

The expected capacity represents the average ratio of post-earthquake to pre-earthquake maximum speeds across all bridges in each county.

**Table 3 | Travel times (in minutes) to reach the closest functional acute care hospital before and after the earthquake for different Bay Area counties**

| County | Pre-earthquake | Post-earthquake | | Population (thousands) |
|---|---|---|---|---|
| | | Mean | Std. Dev. | |
| Alameda | 5.1 | 25.6 (507%) | 34.8 (688%) | 1682 |
| Contra Costa | 7.9 | 19.6 (247%) | 27.7 (350%) | 1166 |
| Marin | 8.1 | 33.6 (414%) | 47.7 (588%) | 251 |
| Napa | 7.9 | 17.9 (228%) | 27.8 (354%) | 138 |
| San Francisco | 2.8 | 6.0 (215%) | 24.0 (857%) | 874 |
| San Mateo | 5.6 | 10.9 (195%) | 25.6 (461%) | 764 |
| Santa Clara | 6.1 | 13.8 (226%) | 27.1 (442%) | 1936 |
| Solano | 7.3 | 13.9 (191%) | 29.2 (401%) | 453 |
| Sonoma | 9.5 | 15.5 (162%) | 22.1 (231%) | 489 |
| Total | 6.1 | 16.9 (277%) | 23.9 (391%) | 7753 |

Post-earthquake values include the mean and standard deviation, expressed as a percentage of pre-earthquake travel times (in parentheses).

typically $DS \in$ {None, Slight, Moderate, Extensive, Complete}. Finally, $IM$ is a random variable representing an intensity measure of shaking at the building site, and generally, $IM \in \mathbb{R}_{\geq 0}$. Note that earthquake shaking on the Earth's crust has a physical upper bound; however, this bound is not generally modeled, as large shaking cannot produce a $DS$ greater than Complete, thereby having a minimal impact on seismic risk analysis. Frequently, $DS$ and $IM$ are marginalized from $P_{DV,DS,IM}(dv, ds, im)$ (e.g., through summation and integration) to find $P_{DV}(dv)$[53-55].

Lee and Kiredmjian[56] first formalized the extension of the PBEE formulation to multiple infrastructure units[57–59,59,60], focusing on transportation infrastructure. After finding marginal distributions of damage in single units, they formulate joint distributions of damage for all units by defining spatial correlations and their decay for distant sites. However, in the last two decades, empirical studies have better characterized spatial correlation on shaking (i.e., $IM$) rather than building damage[61–64]. To account for it, Ceferino et al[9]. first formulated an extension of regional PBEE incorporating earthquake shaking's spatial correlation, defining fully the set of conditional independencies in state-of-the-art regional risk models and applications[65–68]. Ceferino et al[9]. defined the regional model for earthquake casualties in the residential building portfolio, and here we apply it to hospital and transportation infrastructure portfolios in a region.

We first formulate the model for hospital infrastructure. We redefine the traditional PBEE notation to keep the equations concise in the extension to a regional analysis with many buildings. We denote a random variable $X$'s probability distribution $P_X(x) = \pi(x)$. Similarly, for a

multi-variate vector $\boldsymbol{X}$, we denote its probability distribution $P_X(x) = \pi(x)$, where $x$ is a specific realization of $\boldsymbol{X}$. Using this notation, we define the damage ordinal variable $D$, instead of $DS$, and call $D_k^s$ and $D_k^n$ the structural and non-structural damage of building $k$. Thus, we are interested in the damage vector $\boldsymbol{D} = \{D_1^s, D_1^n, \ldots, D_m^s, D_m^n\}$, where $m$ is the total number of buildings in the region. We also define the shaking variable $I$, instead of $IM$, and call $I_k^s$ and $I_k^n$ the shaking measure affecting structural and non-structural damage of building $k$. We use the more concise notation $I$ rather than $IM$ to improve the readability of expressions in high-dimensional regional models, where large sets of correlated variables are simulated. For example, $I_k^s$ can be the peak ground acceleration or spectral acceleration affecting the structural components of buildings, and $I_k^n$ is the peak floor acceleration affecting acceleration-sensitive non-structural components, e.g., ceilings. Thus, we are interested in the shaking intensity vector $\boldsymbol{I} = \{I_1^s, I_1^n, \ldots, I_m^s, I_m^n\}$, where $m$ is the total number of buildings in the region. We extend the Markovian (or conditional independence) assumptions from single buildings[54,55] to multiple buildings[9] to estimate the joint probability distribution of damage and intensity vectors as

$$\pi(\boldsymbol{d}, \boldsymbol{i}) = \pi(\boldsymbol{d}|\boldsymbol{i})\pi(\boldsymbol{i}) \qquad (2a)$$

$$\pi(\boldsymbol{d}|\boldsymbol{i}) = \prod_{k=1}^{m} \pi(d_k^s|i_k^s)\pi(d_k^n|i_k^n), \qquad (2b)$$

where $\boldsymbol{d} = \{d_1^s, d_1^n, \ldots, d_m^s, d_m^n\}$ and $\boldsymbol{i} = \{i_1^s, i_1^n, \ldots, i_m^s, i_m^n\}$ are specific realizations of $\boldsymbol{D}$ and $\boldsymbol{I}$, respectively. Eq. (2b) assumes damage is

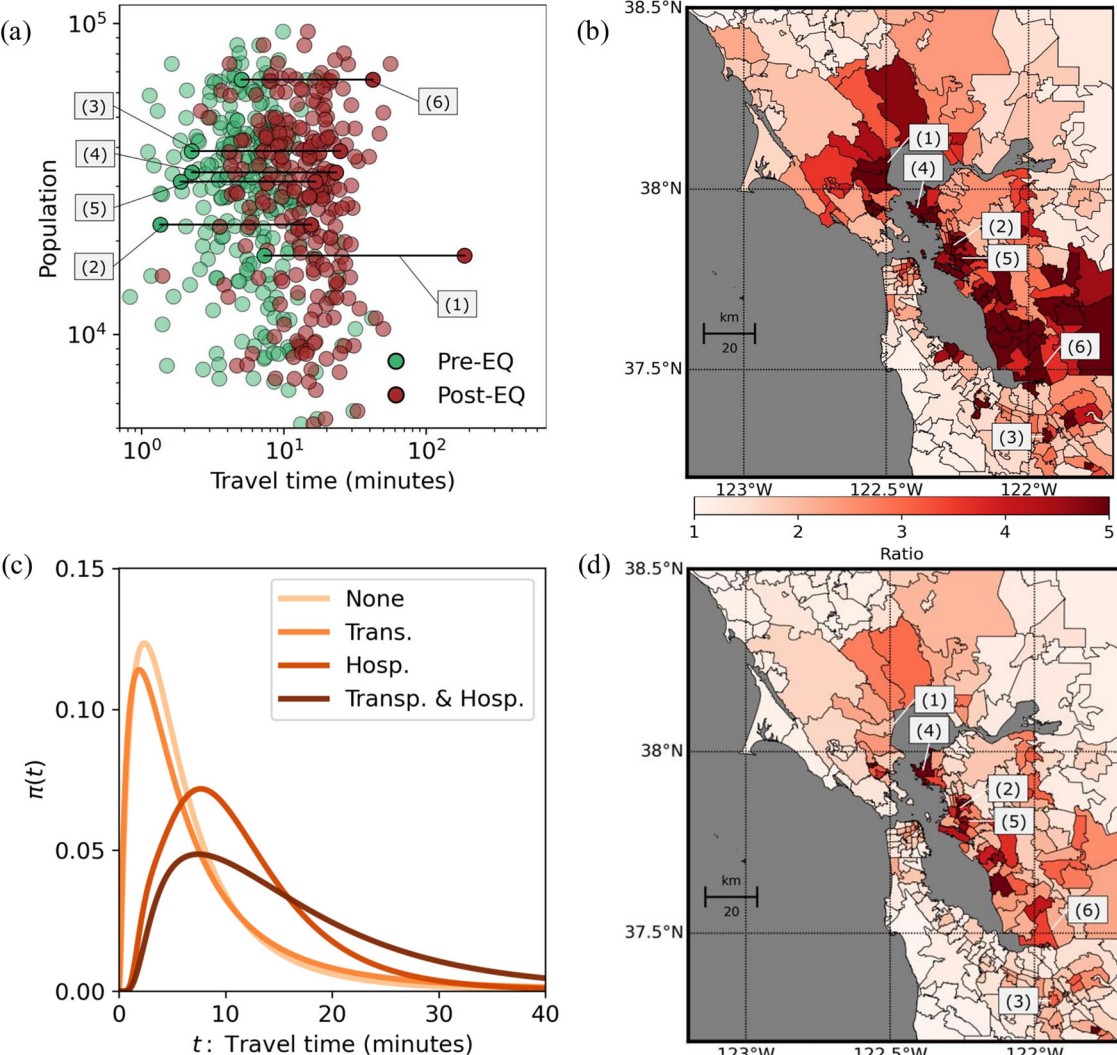

**Fig. 4 | Access to functional acute care hospitals in the Bay Area after the earthquake. a** Post-earthquake (Post-EQ) increases in travel time at the zip-code level compared to pre-earthquake (Pre-EQ) conditions, highlighting densely populated areas that are most severely impacted. **b** Ratios of post- to pre-earthquake travel times to the closest acute care hospitals. Map data from https://www.openstreetmap.org/copyrightOpenStreetMap ©contributors. **c** Distribution of travel times per zip code under four scenarios: None (no infrastructure disruptions), Transp. (only transportation infrastructure is vulnerable), Hosp. (only hospital infrastructure is vulnerable), and Transp. & Hosp. (both transportation and hospital infrastructures are vulnerable). **d** Ratios of post- to pre-earthquake travel times when only hospital infrastructure (Hosp.) is vulnerable. Map data from https://www.openstreetmap.org/copyrightOpenStreetMap ©contributors. Results generated from 5000 Monte Carlo simulations.

independent at different buildings conditioned on their respective shaking intensities; thus $\pi(\boldsymbol{d}|\boldsymbol{i})$ can be estimated as the product of probability distributions of damage in each building. However, unconditional damages will be correlated through the joint probability distribution of shaking in the region, $\pi(\boldsymbol{i})$. This formulation follows the chain rule of probability, where Eq. (2a) represents the joint distribution of damage and intensity. Eq. (2b) assumes that, conditional on shaking intensity, the damage at each building is independent—an assumption that simplifies regional modeling. This allows the conditional joint distribution $\pi(\boldsymbol{d}|\boldsymbol{i})$ to be expressed as the product of per-building fragility functions. While this assumes no direct interaction between buildings once shaking is known, the spatial correlation of ground motion in $\pi(\boldsymbol{i})$ introduces dependence across damage outcomes.

Similarly, we assume that structural and non-structural damage ($D_k^s$ and $D_k^n$) within a building $k$ are conditionally independent given their respective shaking intensities ($I_k^s$ and $I_k^n$). Yet, these two random variables of damage will be correlated since these shaking intensities in the building are also correlated, e.g., peak ground acceleration and

peak floor acceleration. Supplementary Fig. 5 illustrates and summarizes all conditional dependencies through a probabilistic graphical model for the hospital buildings.

Although non-structural damage may not be conditionally independent of structural damage—given that structural damage can alter the dynamic properties of a building and, in turn, affect demands on acceleration-sensitive non-structural components—we consider the independence assumption to be reasonable within the failure space of hospitals. Specifically, our model assumes that hospital functionality is lost once early damage thresholds (e.g., slight damage) are exceeded; at this stage, structural degradation is unlikely to significantly modify the system's dynamic characteristics. While we are not aware of empirical studies explicitly validating this hypothesis, the assumption allows us to simplify the model without compromising its ability to capture key failure mechanisms relevant to hospital functionality.

We assess the building functionality vector $\boldsymbol{F} = \{F_1, …, F_m\}$, where $F_m$ is a Bernoulli random variable that assesses whether the hospital building $k$ will work ($F_k = 1$) after the earthquake. We model $F_k$ as a deterministic function of the structural and non-structural damage of

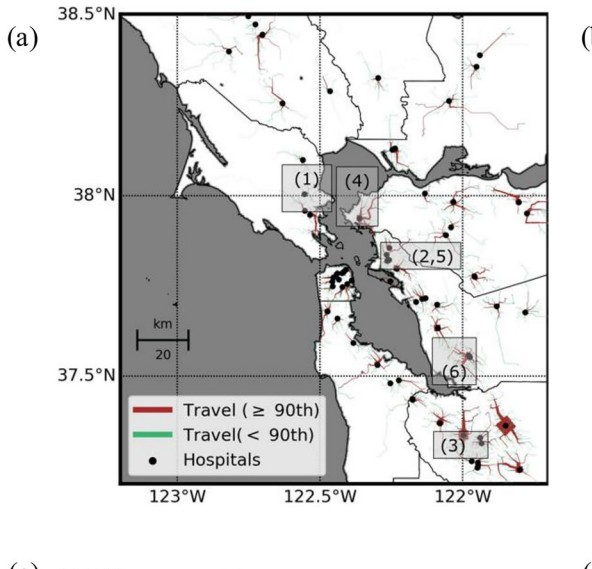

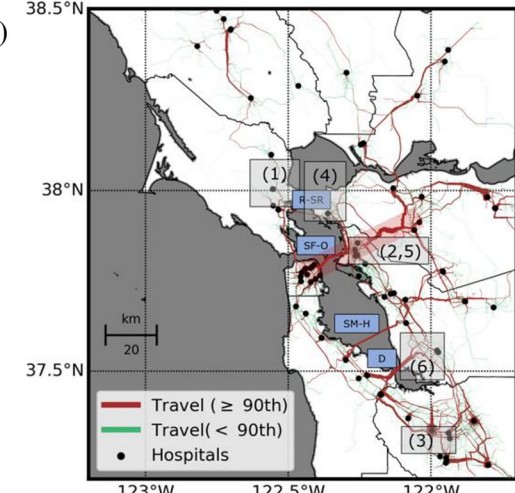

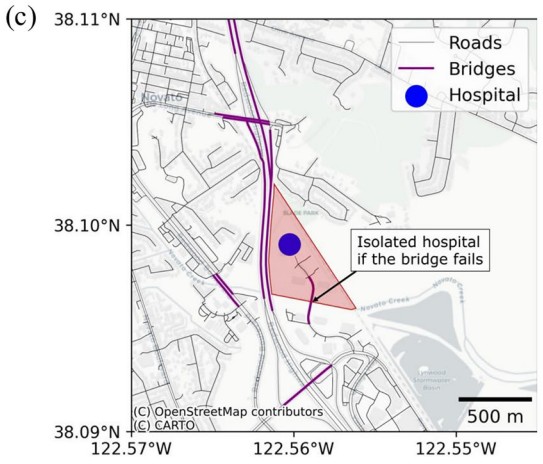

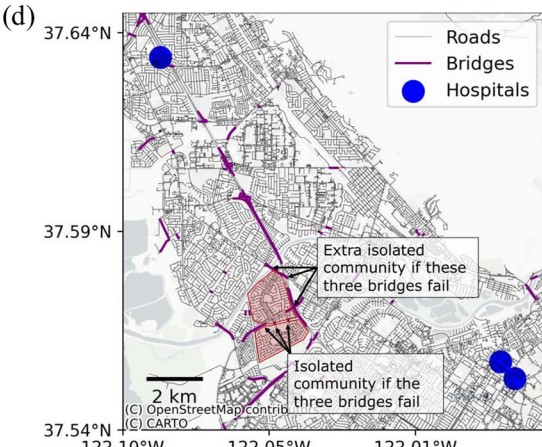

**Fig. 5 | Mobility to reach the closest functional acute care hospital in the San Francisco Bay Area. a** Pre-earthquake travel volumes to the closest acute care hospitals. Numbers 1 to 6 indicate the hospitals serving the dense zip codes with the highest post-earthquake increases in travel time. Map data from https://www.openstreetmap.org/copyrightOpenStreetMap ©contributors. **b** Projected travel volumes after the earthquake, showing major shifts in patient mobility. The red-shaded corridor illustrates how Oakland residents (#2 and #5) must travel west to the Peninsula or east to Walnut Creek due to hospital and transportation disruptions.

Map data from https://www.openstreetmap.org/copyrightOpenStreetMap ©contributors. **c** The hospital serving zip code #1 (Novato) relies entirely on a single bridge; if the bridge fails, all access to its 64 beds is lost. Map data from https://www.openstreetmap.org/copyrightOpenStreetMap ©contributors. **d** Community near zip code #6 (Fremont), where the southern and northern sections become fully isolated if three different bridges fail in each area. Map data from https://www.openstreetmap.org/copyrightOpenStreetMap ©contributors. Results are based on 5000 Monte Carlo simulations of post-earthquake patient mobility.

the building as an input. Thus, under a change of variables, Eq. 2 becomes

$$\pi(\boldsymbol{f}, \boldsymbol{i}) = \pi(\boldsymbol{f}|\boldsymbol{i})\pi(\boldsymbol{i}) \tag{3}$$

where $\boldsymbol{f} = \{f_1, ..., f_m\}$ is a specific realization of $\boldsymbol{F}$. Supplementary Fig. 6 illustrates and summarizes all conditional dependencies through a probabilistic graphical model. Note that a probabilistic formulation that links damage to functionality could also be incorporated, introducing an additional term to Eq. (3), similar to Eq. (1). Thus, our approach admits such extensions.

Hospitals can rapidly lose functionality at early stages of structural and non-structural damage, as observed in the 2023 M 7.8 Kahramanmaras Earthquake in Turkey[8], 2011 M 6.1 Christchurch Earthquake in New Zealand[24], and 2010 M 8.8 Maule Earthquake in Chile[69]. We consider that damage thresholds, $d^s$ and $d^n$, for the structural and non-structural components trigger the disruption of hospital functionality, i.e., if either fails, the hospital loses functionality. We can estimate the probabilities of not exceeding these thresholds as $p_k^s = \pi(D_k^s \leq d^s | I_k^s = i_k^s)$ and $p_k^n = \pi(D_k^n \leq d^n | I_k^n = i_k^n)$, respectively, which can

be evaluated with earthquake fragility functions. Thus, we can model functionality as the intersection of both random events, i.e., the structural and non-structural components work after the earthquake. Since $D_k^s$ and $D_k^n$ are conditionally independent, the $F_k$ is a Bernoulli random variable with probability $p_k^s p_k^n$. Accordingly,

$$\pi(\boldsymbol{f}|\boldsymbol{d}) = \prod_{k=1}^{m} \left[ p_k^s p_k^n \right]^{f_k} \left[ 1 - p_k^s p_k^n \right]^{1-f_k} \tag{4}$$

The functionality of the hospital relies on the structural and non-structural components as described in Eq. (4). This formulation reflects our focus on estimating hospital functionality during the initial post-earthquake period, where service disruption is most directly tied to structural and non-structural damage. Rather than modeling the full trajectory of hospital recovery[70]—which depends on permitting, logistics, and staffing—we assess whether each building is likely to remain operational immediately after the event. To enable this analysis across hundreds of facilities and correlated scenarios, we use Bernoulli random variables derived from fragility-based exceedance probabilities, accounting for spatial dependencies in seismic intensity and building response (Supplementary Fig. 6).

The hospital portfolio has various levels of SPC and NPC that control the final functionality of the hospital.

We are also interested in the probability distribution of the total number of functional beds $B_t$ in the region, which can be estimated as

$$B_t = \sum_{k=1}^{m} \beta_k F_k \tag{5}$$

where $\beta_k$ is the number of beds (all functional before the earthquake) at the hospital building $k$. We are also interested in estimating the distributions of the total number of functional beds $B_h$ in hospital $h$ as each hospital can have multiple buildings with beds. Let $k_h$ be the set containing all indexes of the buildings belonging to the hospital $h$. Thus,

$$B_h = \sum_{k \in k_h} \beta_k F_k \tag{6}$$

In addition, we want to evaluate the contributions of structural and non-structural damage in buildings to the loss of functionality on the hospital campus. We can model the expected number of buildings with structural damage in a hospital campus given that the hospital only has a portion $\phi$ of functional beds as

$$\mathbb{E}\left(\sum_{k \in k_h} 1\{D_k^s > d^s\} \,\middle|\, B_h < \phi \sum_{k \in k_h} \beta_k\right) = \sum_{k \in k_h} \frac{\pi(D_k^s > d^s \cap B_h < \phi \sum_{k \in k_h}\beta_k)}{\pi(B_h < \phi \sum_{k \in k_h}\beta_k)} \tag{7}$$

Similarly, we can evaluate the number of buildings with non-structural damage in a hospital campus given that the hospital only has a portion $\phi$ of functional beds as

$$\mathbb{E}\left(\sum_{k \in k_h} 1\{D_k^n > d^n\} \,\middle|\, B_h < \phi \sum_{k \in k_h} \beta_k\right) = \sum_{k \in k_h} \frac{\pi(D_k^n > d^n \cap B_h < \phi \sum_{k \in k_h}\beta_k)}{\pi(B_h < \phi \sum_{k \in k_h}\beta_k)} \tag{8}$$

We formulate the model for damage to the bridge infrastructure and its post-earthquake functionality following the state of the art in previous implementations[71]. As with buildings, we use fragility functions to estimate structural damage to bridges. Post-earthquake functionality is represented in terms of reduced maximum free-flow travel speed. We adopt deterministic relationships between damage levels and speed reductions based on previous research, as fully described later below.

**Approaches for numerical solutions**
Regional seismic risk analysis is a high-dimensional problem. While the joint distribution $\pi(f, i)$ in Eq. (3) can be analytically evaluated due to closed-form expressions for $\pi(f|i)$ and $\pi(i)$[62–64,72], computing the marginal distribution $\pi(f)$ is not analytically tractable. This is because it involves integrating over a set of correlated random variables in $i$, for which no closed-form marginalization exists. Numerical integration methods (e.g., Riemann integration) quickly become computationally infeasible, even for a modest number of buildings[9]. To address this, several approaches have been proposed to efficiently approximate $\pi(f)$ in large-scale settings. For example, Ceferino et al[9,10]. demonstrated−both theoretically and empirically−that under certain conditions, Central Limit Theorem (CLT)-based approximations can accurately and efficiently estimate $\pi(f)$. More recently, Heresi and Miranda[60] showed that similar approximations remain valid even under mild correlation structures in seismic risk models. However, in situations where the CLT does not apply, modern risk models often rely on Monte Carlo methods to estimate $\pi(f)$. In our case, we seek to estimate

the distribution of $b_t$, the total number of functional beds across hospital buildings, where functionality follows the distribution $\pi(f)$. Since the number of hospital buildings varies widely−from a few to many−CLT approximations are not generally applicable. Therefore, we use Monte Carlo simulation to evaluate $\pi(b_t)$. We implemented this using the NHERI SimCenter's R2D Tool[73–75], which enables scalable regional risk assessments. Specifically, we used R2D to generate 5000 realizations of $i$ and $d$ for both hospitals and bridges, which form the basis for evaluating Eq. 2.

**Earthquake rupture and shaking modeling**
We study an M 7.25 earthquake scenario on the Hayward Fault. Similar scenarios have been extensively studied to inform resilience policy-making in the Bay Area[48]. The rupture geometry was obtained from the Uniform California Earthquake Rupture Forecast (UCERF) 2[76]. The earthquake ruptures the Hayward South and North sections over a total length of ~110 km.

Estimates of shallow shear wave velocities (averages at the top 30 m of soil) are utilized over the entire Bay Area[77]. With this information, we built the joint probability distribution of shaking intensities $\pi(i)$ in R2D. We utilized a ground motion model[78] for shallow crustal earthquakes to estimate medians and logarithm standard deviations of $i$. $i$'s uncertainty is divided into two components[79]. The first component captures between-event uncertainty and affects the entire region equally, but it varies per intensity measure type, e.g., peak ground acceleration versus spectral acceleration. The first component captures correlation across different intensity measures[80]. The second component captures within-event uncertainty and affects the entire region and intensity measure types differently. This component captures spatial correlation and correlation across different intensity measure types. We use a computationally efficient method to account for the second component[64,81]. As stated earlier, we sampled 5000 realizations of $i$ and show expected values $\mathbb{E}(i)$ for the entire region (Fig. 1). Notice that for illustration purposes, we show $\mathbb{E}(i)$ for the entire Bay Area (+10,000 locations), but for the hospital network's risk analysis, we only need to quantify $i$ at the 426 building locations and 5163 bridge locations.

**Hospital vulnerability modeling**
This paper utilized building-level lognormal fragility functions to determine damage to structural and non-structural components. For example, to determine the likelihood of reaching or exceeding a structural damage threshold $d^s$ in building $k$ as a function of the shaking intensity measure, we use fragility functions like

$$\pi\left(D_k^s \geq d^s | I_k^s = i_k^s\right) = \Phi\left(\frac{log(i_k^s) - log(\alpha)}{\beta}\right) \tag{9}$$

where $\Phi$ is the standard normal cumulative distribution function[82]. The parameters $\alpha$ and $\beta$ define the fragility function and vary according to the damage threshold $d^s$ and the building's structural type and vulnerability (e.g., SPC rating) for building $k$. $\alpha$ equals the shaking intensity (e.g., PGA) that exposes the building to a 50% probability of damage of at least $d^s$. $\beta$ is a normalizing factor that defines the width of the transition range between shaking with low and high damage probability, and it is a measure of aleatory uncertainty in the vulnerability analysis. In the limit, when $\beta \to 0$, Eq. (9) becomes equivalent to a deterministic assessment, where the building would fail after a fixed shaking threshold. An analogous equation is used for non-structural damage.

We assessed structural damage across a variety of structural system types and five levels of structural vulnerability (Fig. 1 and Supplementary Fig. 1). To represent this range, we used and adapted structural fragility functions from HAZUS[49], also available through the R2D tool[83]. Based on the definitions of SPC ratings, we mapped SPC 1

and 2 buildings to the HAZUS pre-code and moderate-code fragility functions, respectively, using the corresponding $\alpha$ and $\beta$ parameters in Eq. (9). This mapping reflects the fact that SPC 1 and 2 buildings do not comply with the structural provisions of the Alquist Hospital Facilities Seismic Safety Act and are known to be older and more vulnerable structures. While no empirical fragility curves directly link SPC ratings to seismic performance, post-earthquake damage reports from events such as the 1971 San Fernando and 1994 Northridge earthquakes show that hospitals with older structural systems experienced significant damage. Although these observations are not sufficient to derive fragility functions, they support the alignment of SPC 1 and 2 facilities with more vulnerable construction classes, consistent with HAZUS classifications. In contrast, SPC 3, 4, and 5 comply with the Alquist Act. Fragility functions for SPC 5 buildings are obtained by increasing all the $\alpha$ values for high-code structures by 50% higher PGAs. These adjustments were made to represent that hospitals designed to meet the Alquist Act (SPC 5) according to the ASCE7-16 building code are designed to withstand 50% higher seismic loads than regular buildings, i.e., an importance factor of 1.5[84]. SPC 3 buildings are steel structures that comply with the Alquist Act but are pre-Northridge. We used regular building fragility functions to represent the vulnerability of SPC buildings as pre-Northridge's non-ductile connections as it takes ~30% less seismic demands to make them reach moderate and extensive levels of damage[85], i.e., $0.7 \times 1.5\alpha = \sim \alpha$. Finally, we modeled SPC 4 buildings with $1.25\alpha$ to represent that these buildings have lower performance than SPC 5 buildings ($1.5\alpha$). SPC 4 buildings comply with the Alquist Act but can have some structural conditions that make them more prone to damage, e.g., lack of weak beam/strong column, presence of short captive columns[86].

We followed a similar approach for the fragility functions of buildings' non-structural components. We mapped NPC 1, 2, and 3 to pre-code, moderate-code, and high-code fragility functions for acceleration-sensitive non-structural components from HAZUS[87]. For NPC 4 and 5, we adjusted the fragility functions of high-code fragility functions by increasing the median peak floor acceleration (PFA) to reach different damage states by 25% and 50%, respectively. Unlike structural components, we defined the same fragility functions for all building types because hospitals have similar non-structural components, e.g., equipment for acute care. However, as stated earlier, the input PFA for each hospital building differs and depends on the structural type.

After defining these fragility functions for the portfolio of hospital buildings, we computed realizations of building damage utilizing R2D[88]. As mentioned, we generated 5000 samples of the buildings' structural and non-structural damage for each ground-shaking simulation. As stated earlier, we initially tested multiple thresholds of damage, $d^s$ and $d^n$ (e.g., slight, moderate, extensive damage) to evaluate hospital disruptions (Fig. 3). However, we used the slight damage threshold for most of the analysis later in the study since most hospitals lose functionality at quite early stages of damage. The damage state definitions adopted in this paper for building structural components and non-structural components are consistent with the damage states defined by HAZUS[89].

At each building $k$, we used the Peak Ground Acceleration (PGA) as the shaking intensity random variable $I_k^s$ to estimate structural damage. While spectral accelerations can also be used to improve damage predictability through the inclusion of structural properties, such as period of vibration[10], we did not follow this approach here since fragility spectral acceleration-based fragility functions are not available for the diversity of building typologies in the San Francisco Bay Area.

We used PFA as the shaking intensity random variable $I_k^n$ to estimate non-structural damage at building $k$. Our focus on PFA stems from the fact that acceleration-sensitive non-structural components—such as ceilings, shelves, and mechanical or medical equipment—tend to fail at lower shaking levels than drift-sensitive components like partition walls. Furthermore, in hospitals, acceleration-sensitive components (e.g., x-ray machines) often play a more critical role in maintaining functionality after an earthquake.

PFA varies with height within a building, and total non-structural damage depends on how acceleration-sensitive components are distributed across floors. However, detailed data on the floor-by-floor distribution of non-structural elements are rarely available in regional-scale analyses. To address this limitation, we used a simplified proxy to approximate the variation of PFA with building height while incorporating key building characteristics. Specifically, we assumed a linear distribution of floor accelerations, with the base acceleration set equal to the peak ground acceleration (PGA) and the acceleration at the effective height equal to the spectral acceleration corresponding to the building's dynamic period. The effective height is estimated for each structural type based on period and height values from the HAZUS exposure model[49]. We then compute the average PFA across the building height from this linear profile and use an average value across height in our non-structural damage analysis. Further studies can enhance the fidelity of this analysis if higher-resolution information for non-structural components is available, e.g., on each floor.

## Bridge vulnerability modeling

This paper assesses highway bridge damage due to both ground shaking and soil liquefaction. To estimate shaking-induced damage, we adopt the methodology developed in HAZUS[49], which defines 28 representative bridge archetypes. Each archetype is assigned a fragility function of the form in Eq. (9), using spectral acceleration at a 1-second period as the intensity measure (denoted as $i$). To account for variations among individual bridges within each archetype, modification factors are applied to the fragility parameter $\alpha$. These include a three-dimensional modification to capture deck arching effects in multi-span bridges and a skew factor to reflect reduced capacity at skewed deck-pier connections. Additionally, for estimating slight damage, a spectral shape modification converts the intensity measure to an equivalent 0.3-second spectral acceleration, acknowledging that bridges are typically elastic and sensitive to short-period ground motions.

To assess damage from ground failure, HAZUS[89] defines fragility curves that use permanent ground deformation (PGD) as the intensity measure. Given the high liquefaction susceptibility in the Bay Area, we adopt the HAZUS-recommended geologic modelHAZUS[89] to estimate liquefaction-induced PGD at each bridge location. This model first uses empirical correlations between soil types (e.g., intertidal mud, Holocene and Pleistocene alluvium, rock) and liquefaction susceptibility. Liquefaction probability is then estimated based on peak ground acceleration (PGA), earthquake magnitude (to capture shaking duration), and groundwater depth (to reflect pore pressure effects). PGD is subsequently estimated using empirical relationships with the calculated liquefaction probability. We utilize the California statewide geologic map[90] and a global groundwater depth model(2010)[91] for this analysis. Once PGD is estimated, ground failure-induced bridge damage is determined using HAZUS fragility curves through the R2D software platform.

These fragility curves yield the probability of each bridge reaching damage states—slight, moderate, extensive, or complete—under a given ground motion realization. HAZUS defines the physical characteristics associated with each state (e.g., slight damage involves minor cracking at abutments, while complete damage corresponds to column collapse or loss of bearing support). To estimate post-earthquake traffic performance, we adopt the damage-capacity model from Guo et al. (2017)[50], which relates bridge damage to residual free-flow speed. Travel time is computed as road length divided by this speed under the assumption that emergency vehicles receive priority and travel at free-flow speed. More realistic travel times could incorporate background traffic using volume-delay relationships such as the Bureau of Public Roads (1964) curve[92], though this requires

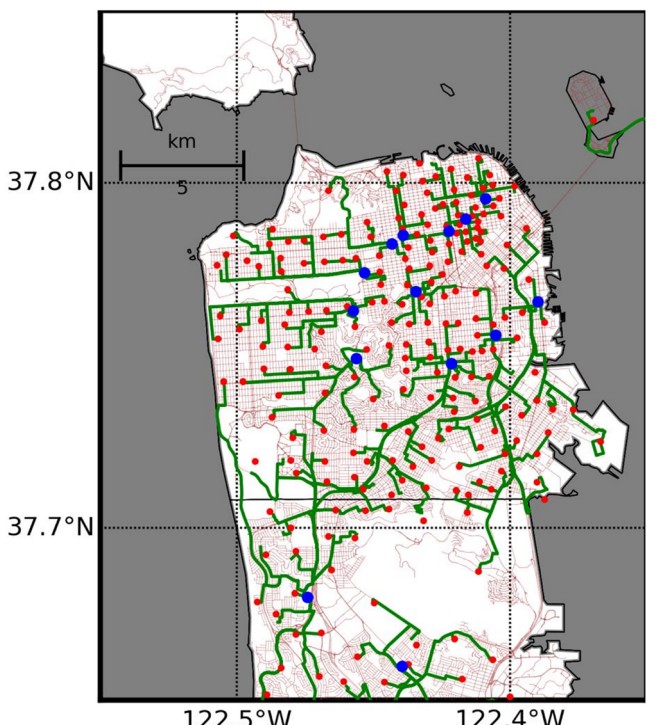

**Fig. 6 | Shortest paths from patient vertices ($p \in \boldsymbol{p}$) to hospital vertices ($h \in \boldsymbol{h}$) before the earthquake in the San Francisco county.** The transportation network is shown in brown lines, the shortest paths in green lines, hospital vertices in blue dots, and patient vertices in red dots. Map data from https://www.openstreetmap.org/copyright OpenStreetMap ©contributors.

post-disaster traffic data, which remains a key challenge in traffic simulations after major events.

### Network model: acute care accessibility modeling

We denote $G = (\boldsymbol{v}, \boldsymbol{e})$ a graph, where $\boldsymbol{v}$ is the set of $|\boldsymbol{v}|$ vertices and $\boldsymbol{e}$ the set of $|\boldsymbol{e}|$ edges. This graph will represent the infrastructure system of hospitals and roads supporting healthcare access in the Bay Area. Let $p \in \boldsymbol{p}$ be a vertex representing the location of a patient needing acute care and $\boldsymbol{p}$ be the set of all patient vertices. Also, let $h \in \boldsymbol{h}$ be a vertex representing the location of an acute care hospital and $\boldsymbol{h}$ be the set of all hospital nodes. In this case, $\boldsymbol{p} \subseteq \boldsymbol{v}$, and $\boldsymbol{h} \subseteq \boldsymbol{v}$. $\boldsymbol{e} = \{(u, v) | u, v \in \boldsymbol{v}\}$ is the set of $|\boldsymbol{v}|$ edges representing different roads connecting different locations (vertices) in the region. We denote $\tau(u, v) \in \mathbb{R}_{\geq 0}$ the travel time between vertices $u$ and $v$.

To analyze healthcare access, we evaluate the shortest travel times for a patient in node $p$ to reach any hospital vertex in the set $\boldsymbol{h}$. We modeled hospital $h$ as a source vertex and found the shortest paths to all vertices $p \in \boldsymbol{p}$ simultaneously, resulting in a shortest-path tree problem. This approach is faster than computing multiple shortest for each pair $h$ and $p$, separately. Thus, for each vertex $h$, we find the shortest time to reach the patient:

$$t^*(h) = \min_{(u,v) \in \boldsymbol{e}} \sum \tau(u, v) x(u, v) \tag{10a}$$

$$s.t. \quad \sum_{v:(u,v) \in \boldsymbol{e}} x(u, v) - \sum_{v:(v,u) \in \boldsymbol{e}} x(v, u) = b_u, \quad \forall u \in \boldsymbol{v} \tag{10b}$$

$$b_u = |\boldsymbol{p}| \quad \text{for} \quad u = h \tag{10c}$$

$$b_p = -1 \quad \text{for} \quad u \in \boldsymbol{p} \tag{10d}$$

$$b_u = 0 \quad \text{for} \quad u \neq h, p \tag{10e}$$

$$x(u, v) \in \{0, 1\}, \forall (v, u) \in \boldsymbol{e} \tag{10f}$$

This optimization problem seeks to find the minimum travel time from vertex $h$ to all vertices in $\boldsymbol{p}$ simultaneously. This optimization problem can also be interpreted as a special network flow problem where we seek to find a directed path with minimum cost from a source node $h$ to multiple destinations. $\tau(u, v)$ can be interpreted as edge flow cost, and the shortest path problem can be seen as sending a flow unit to each destination $p \in \boldsymbol{p}$. We solve Eq. 10 to find the shortest travel time $t^*(h)$ from each node $p \in \boldsymbol{p}$ to each (hospital) vertex $h$ (Fig. 6). We can use multiple algorithms to solve Eq. 10, including Dijkstra and Bellman-Ford[93]. For reference, the computational complexity of the classical Dijkstra's algorithm is $\mathcal{O}(|\boldsymbol{v}|^2)$ and Bellman-Ford is $\mathcal{O}(|\boldsymbol{v}||\boldsymbol{e}|)$. Note that these algorithms will find the shortest paths to all vertices in $\boldsymbol{v}$, and not just on $\boldsymbol{p}$, at once.

After finding $t^*(h)$ from all $h \in \boldsymbol{h}$, we compute the minimum travel time $t^*(h)$ to any hospital by comparing the different options each patient $p$ has. Thus,

$$T^* = \min_{h \in H} t^*(h) \tag{11}$$

In the post-earthquake scenario, all hospitals will be functional. Then, Eq. (11) finds the shortest time to reach an acute care hospital $p$ (Fig. 6).

In an earthquake, we adjust Eq. (11) to

$$\overline{T}^* = \min_{h \in \overline{\boldsymbol{h}}} t^*(h) \tag{12}$$

where $\overline{\boldsymbol{h}} = \{h | B_h > 0.5\beta_h \quad \forall h \in \boldsymbol{h}\}$ to account for the reduction in hospital capacity. Thus, $\overline{\boldsymbol{h}}$ is a set with a random selection of elements, i.e., hospital vertices. This means that in this model, the tipping for receiving new patients in hospitals is to have at least 50% of their beds ($\beta_h$) functional. Studies on hospital disaster resilience during earthquakes suggest that a significant reduction in functional bed availability can severely strain hospital operations, particularly in accommodating new patients in the emergency department[30,94].

We also further adjust the edge capacity and the network topology to solve Eq. 10. As mentioned earlier, the free-flow travel times $\tau$ are adjusted to post-earthquake conditions $\overline{\tau}$ according to the bridge damage level. Thus, we use $\overline{\tau}$ instead of $\tau$ to find $t^*(h)$ in Eq. 10. While in practice, fully collapsed bridges would have travel time $\tau = \infty$, we found it was more computationally efficient to remove these edges the network. Thus, we also used a slightly different edge $\overline{\boldsymbol{e}} = \{(u, v) | \overline{\tau}(u, v) < \infty \quad \forall (u, v) \in \boldsymbol{e}\}$ for each simulation. Thus, $\overline{T}^*(p)$ is a random variable. We solve Eq. (12) 5000 times using the 5000 Monte Carlo simulations obtained before.

### Bay Area's Transportation System

We study the entire San Francisco Bay, where the transportation network is massive, with $|\boldsymbol{v}| = 0.5$ million vertices and $|\boldsymbol{e}| = 1.5$ million edges. The edges in the graph represent the roads in the transportation system, whose information was obtained from the San Francisco Region Roadways[95] and OpenStreetMap (OSM)[96]. We incorporated directional and travel time data into the network models using OSM's OSMnx library[97]. We couple the transportation network data to the hospital and the patient data. The 76 hospitals are embedded in the network models by identifying the network's vertices closest to the hospital locations, forming the set $\boldsymbol{h}$. In addition, we obtained population data at 1613 zip codes, and we assumed that the number of patients is proportional to the population.

Our goal was to quantify hospital accessibility losses beyond the immediate emergency period, recognizing that hospitals often take

months or even years to be repaired. Because people are typically injured within damaged buildings, patient surges after earthquakes generally last only a few days—except in rare cases such as the 2023 Türkiye Earthquake. To focus on long-term access rather than short-term emergency response, we did not model injury locations directly. Instead, we assumed that patient demand is proportional to population distribution and modeled accessibility accordingly. For the Bay Area, we based our analysis on population-weighted access loss and identified the network's vertices closest to the centroids of census tracts to define the set *p* (Fig. 6).

## Data availability
The hospital data are publicly available in the California Department of Health Care Access and Information's Portal: https://hcai.ca.gov/facilities/building-safety/facility-detail/. The transportation data are publicly available in the San Francisco's Metropolitan Transportation Commission's Portal https://opendata.mtc.ca.gov/. In addition, we made the hospital disruption simulations and the pre-processed network data accessible in the following NSF's DesignSafe repository: https://doi.org/10.17603/ds2-pc7z-7227. Notice that files can be downloaded individually. Source data are provided with this paper.

## Code availability
The risk model was run with NHERI SimCenter's R2D software, open and publicly available: https://simcenter.designsafe-ci.org/research-tools/r2dtool/. Also, we made the code to run the network model coupled to the risk model publicly available in the following NSF's DesignSafe repository: https://doi.org/10.17603/ds2-pc7z-7227. Notice that scripts can be downloaded individually.

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

## Acknowledgements

We thank the financial support of the New York University's Tandon School of Engineering for this research. The authors are also thankful for the financial support provided by the National Science Foundation through the Grant No. CMMI-2410291 (L.C.). We also acknowledge using LLMs to correct grammar and improve the readability of the writeup.

## Author contributions

L.C. conceived and formulated the risk-network model. C.K., D.M., X.X., J.W., and J.Z. compiled and pre-processed the hospital and transportation data. L.C., C.K., J.Z., and A.Z. run the risk-network simulations. L.C. drafted the manuscript with contributions and editing from all the authors.

## Competing interests

The authors declare no competing interests.
