## [Peer Review File · Nature Communications]

Accessing Acute Care Hospitals in the San Francisco Bay after a Major Hayward Earthquake

Corresponding Author: Professor Luis Ceferino

Version 0:

Reviewer comments:

Reviewer #1

(Remarks to the Author)

This manuscript presents the results from a numerical study designed to evaluate the capacity of hospitals and the anticipated travel times for acute care patients in San Francisco Bay, California, following a potential M7.25 earthquake. This crucial and timely study innovatively integrates a risk model with a transportation network to illustrate the possible cascading effects. However, certain critical assumptions made within the model, which significantly impact the study's conclusions, lack sufficient justification. Details of these assumptions are outlined below. The manuscript requires significant revisions before it can be reconsidered for publication in Nature Communications.

The authors are advised to address the following while revising their manuscript:

[Page 5] The definitions adopted for the damage levels are not defined in the manuscript. As a result, it is difficult to interpret the plausibility of the damage probabilities presented in Figure 2 as well as the post-earthquake capacity estimates (in Table 1, Figures 3 and 4). The definitions of the damage states and their impact of the type of damage on the capacity should be explained well.

[Page 8, Fig. 3, lower left] The legend entry related to markers with small white circles is missing.

[Page 10, Fig. 4, upper right] The figure legend is missing.

[Page 11] In the last paragraph, all "Figure 4"s should read "Figure 5".

[Page 12] The discussion related to the transportation network is difficult to follow since it requires knowledge of the names of the places. It would be very informative to add labels on the maps (e.g. Figure 1) or to cite the numbers indicated in Figure 4 in the text so that the non-residents can follow the observations related to cascading effects with ease.

[Page 15] In Equation 1, " $P_{IM}(I_m)$ " should read " $P_{IM}(i_m)$ ".

[Page 16] It is stated that "structural and non-structural damage (D_{ks} and D_{kn}) within a building k are conditionally independent given their respective shaking intensities (I_{ks} and I_{kn}).". Physical mechanisms that drive non-structural damage, as well as past reconnaissance observations, indicate the opposite. Given a specific shaking intensity, the non-structural damage is very sensitive to the extent of structural damage. The level of structural damage has a direct influence on the acceleration and deformation demands on the non-structural components as well as the changes in the resistance capacities at the anchorage locations.

[Page 16] Building functionality is represented using a Bernoulli random variable which can only have two states: building is functional or not functional. This implies that if there is slight non-structural damage in some part of a hospital building, the entire building will become completely dysfunctional. At present, the definition of slight damage adopted in the manuscript is unclear. However, for the typical definition of slight damage, this is not a realistic assumption. This is particularly true after severe disasters where there is a significant demand for urgent acute care. Many of the hospital buildings with slight non-structural damage would continue to operate with some reduced level of functionality. In order to address this issue, the definitions of damage states should be explained. Moreover, the assumption of complete loss of functionality of the entire

building needs to be justified based on the provided damage definitions and past experiences. Otherwise, the formulation should be revised accordingly.

[Page 18] Difficulties associated with the analytical evaluation of $\rho(f)$ and $\rho(b)$ are essentially very similar. The related sentences can be collected together into a single paragraph so as not to repeat very similar issues twice.

[Page 19] Structural Performance Categories (SPCs) of the hospitals have a critical function in the presented investigation. At present it is unclear if these categories are the revised categories obtained as a result of the HAZUS Reassessment Program for SPC-1 buildings or not. In general, the technical basis of this categorization should be clarified in the manuscript.

[Page 19] Authors state that "Following the definitions of SPC ratings (Supplementary Table 1), we mapped SPC 1 and 2 to pre-code and moderate-code fragility functions for regular buildings from HAZUS ...". This is a critical assumption which would have a direct impact on all calculations. The assumed fragility parameters need to be justified. For this purpose, the extent of the structural and non-structural damages observed in hospitals designed and constructed in other parts of the state (or other states with similar design codes) that experienced earthquakes within the last 50 years can be considered as a basis. If the fragility characteristics show a similar trend to the assumed model, the selected parameters can be considered as suitable. At present, the introduced assumptions are not justified based on any empirical or analytical evidence.

[Page 20] Authors note that they "utilize a proxy for a representative PFA equal to the spectral acceleration at the building's period of vibration.". The assumption of peak floor acceleration being the same as the spectral acceleration at the site of the building is not a suitable one. Such an assumption ignores the height of the building, the type of the system, and any irregularity in the structural layout. These factors significantly influence the resulting floor accelerations.

[Page 22] It is stated that the number of patients is assumed to be proportional to the population. Past disasters show that the number of acute care patients is typically distributed proportional to the intensity of building damage together with the population density. The building damage is a function of shaking intensity and vulnerability of the residential or commercial building stock. In the proposed approach, it appears that these critical factors are not taken into account while modeling the distribution of patients with acute care needs following an earthquake. This assumption needs to be justified or the approach needs to be revised. The assumptions related to the spatial distribution of patients over the region are expected to have a considerable impact on the calculated travel times and the related conclusions.

[Page 22] It appears that the potential for any closure of roads after the earthquake is not considered in the proposed framework. Past earthquakes have shown that havoc-related jams, toppling of electric lines and commercial pole signs, landslides, surface fault ruptures, underground pipe ruptures, building façade debris, damages to bridges, flyovers, and tunnels have led to the closure of roads after earthquakes. In urgent cases, the roads are closed as a part of the disaster response operations and only the emergency vehicles are allowed on the main roads. Furthermore, very often, a scarcity of gas is caused after such earthquakes by the demand surge, which cannot be accommodated quickly. As a result, it becomes a common problem to have vehicles on the sides of the roads left without any gas. In the case of California, where electric vehicles are more common as compared to other regions, the post-earthquake functioning of the grid and the likely charging durations may also have a critical impact. In the present investigation, the potential impacts of such effects on the travel time estimates, as well as transportation network connectivity, are not taken into account. These should be noted as a limitation of the study to inform the reader properly about the complexity of the subject matter.

[Supp. Fig. 3] The damage probabilities are compared against a "reference" fragility model for structural wall buildings. However, the original source of this reference model, as well as the related building typology, is unclear. Therefore, it is not easy to interpret the meaning of this comparison.

(Remarks on code availability)

I downloaded the code and checked its contents very briefly. It contains the related files. I did not run it to reproduce the results.

Reviewer #2

(Remarks to the Author)

As the authors know, one of the most critical elements of any study is the motivation of the work. The authors motivated the work by indicating that, unlike the past studies on hospital networks, this study focuses on a community exposed to earthquakes only within a few kilometers of them. First, the papers cited by the authors include hospitals a few kilometers from the hazard. Second, the justification overall is very weak and does not represent uniqueness. The study is motivated by the hospital being a few kilometers from the hazard, which is irrelevant - not to mention that this was the case in past studies. Furthermore, assessing healthcare networks under earthquakes has been studied quite a bit using more rigorous approaches. It is hard to see valuable contributions in this study.

In addition to the lack of sufficient motivation, the author did not utilize the state-of-the-art in healthcare resilience analysis and used a simplistic approach in their assessment. For example, it is assumed that damage to the building is a proxy for bed availability and resilience, which is not true. Many recent studies describe healthcare resilience as a function of physical infrastructure, staff, and stuff. So, it appears that the authors are going back in time to use a definition that is no longer recognized in the research community.

Moreover, the paper suffers from many technical issues/mistakes, which causes the analysis to be unsound. For example, their predictions show that many hospitals will experience violent shaking at a scale not seen in over a century under a peak ground acceleration of 0.2. This is not possible. A peak ground acceleration of 0.2 is extremely low and is expected to cause only slight damage. The literature is overwhelmed with studies that show how a 0.2 PGA typically causes slight damage to different archetypal structures. In addition, the authors noted that many hospitals will have a probability of damage above 0.25. What is the significance of 0.25 as a damage probability? and What type of damage corresponds to 0.25 or more? Damage probability ranges from slight to moderate to extensive to complete, and each plays a role in quantifying healthcare functionality. To list that the damage probability is 0.25 has no structural engineering relevance. In earthquake engineering, there is no relevance in saying that many hospitals had a damage probability of more than 0.25. The reviewer recommends that the authors look at various FEMA documents on seismic performance assessment to become more familiar with damage assessment approaches, including the various engineering demand parameters used for seismic evaluation of buildings and hospitals.

Concerning this comment, the authors note that their analysis predicts that 8,501 beds will be lost throughout the Bay, and such predictions are based on slight damage (justified by previous studies that used slight damage to indicate loss of beds). Again, the literature is way past the idea of using damage to indicate bed losses and that stuff, staff, and physical systems should be used to indicate bed losses. In addition, many studies show that hospitals continue to operate beyond slight damage, including the studies referenced by the authors. None of the studies cited by the authors show that hospitals will lose functionality at slight damage. The STEER report cited by the authors indicates that only fully collapsed or severely damaged hospitals were evacuated after the Turkey EQ event. Unfortunately, the authors used this assumption to draw the study's main conclusions. Using slight damage drastically skews the results towards showing significant bed losses, which is very unrealistic. Also, the authors note that services within hospital buildings will be highly disrupted – a statement with no basis since the authors did not model any hospital units or services. Another significant issue is that bridge damage and failure analysis are not included in determining travel time. The travel time calculated in the paper can not be correct due to the lack of bridge failure analysis.

The methods section also suffers from statements that are either very simplistic and outdated by more detailed analysis, incorrect, or indicate problems with how the study was conducted. For example, the authors suggested that building functionality is modeled with Bernoulli distribution. The state of the art on building functionality (plz see work by Henry Burton or Naiyu Wang) has shown building functionality to be much more complex than this, requiring more details analysis and the inclusion of various probabilities, including the availability of crew workers, the time it takes to issue a permit, material availability, and many other things. These parameters will impact the hospitals' functionality and the staff's viability since their homes could be damaged - criteria that were not included in this study. Assuming a simple Bernoulli function has no basis.

Another major issue in the methods section is that the authors tried to connect the data collected from the vulnerability rating to the HAZUS fragility. This connection is based on assumptions not validated or supported by previous studies. The authors tried to use ASCE-7 to come up with some numbers that reflect a lack of understanding of the code philosophy. I recommend utilizing fragility functions that reflect the building characteristics; otherwise, the calculated damage does not represent reality.

The authors also assumed that some hospitals could accept patients if only one bed was functional, which is a weak assumption. If only one bed is functional, then the hospital must be substantially damaged, and if most of the hospital is damaged, then it will be unsafe to utilize. Please check previous studies reporting that many hospitals evacuated, even when they had more than 50% of their capacity due to severe damage. I doubt whether any case study or expert opinion will support the assumption made by the authors.

Another major issue in this paper is that the authors ignored damage to the transportation network, which is a fundamental component when calculating the estimated demand at different hospitals. Damage to bridges and the road network will significantly change the shortest paths calculated in this study, which can lead to cases where some functional hospitals see less demand due to reduced accessibility. Many previous studies on healthcare systems reported that impact and some even included the dynamic effect of the change in travel time on hospital demand and accessibility.

The authors also assumed the number of patients proportional to the number of populations. However, considering the variation in patient distribution, especially during disasters, is critical for accurate demand assessment, and again, many studies have already done that. Most earthquake casualties are related to building collapse, a function of the building characteristics and earthquake intensity measured at the building locations. So, the assumption made by the authors has no basis and will significantly impact the results and main conclusions of the study.

Overall, the paper addresses a problem investigated in the past using simple approaches that are not necessarily detailed and, in some cases, infused with incorrect assumptions and/or calculations. Unfortunately, on these grounds, this paper can not be published.

(Remarks on code availability)

I evaluated the authors' online sources, and they all have sufficient information.

Reviewer #3

(Remarks to the Author)

The manuscript presents a case study on the impacts of a scenario earthquake on the Hayward fault on the San Francisco

Bay Area's hospital capacity. It is very interesting work and implemented using state of the art approaches, giving good weight to the discussion points provided.

The biggest issue I found with the work was the length of the text that made it rather difficult to read. I understand that the authors wish to discuss their results in detail, but I felt much of the discussion could have been delivered in half the amount of text. I would strongly urge the authors to revise the initial parts of the paper and trim unnecessary sentences or anecdotes that do not contribute to the main points being delivered.

Furthermore, some of the text is written in a manner that seems less like a technical article, and more like a magazine article for non-technical readers. The excessive use of adjectives (i.e. violent shaking) and adverbs (i.e. we rigorously study) and terminology like "we predict" seems a little exaggerated, at least in this Reviewer's opinion. Reducing these would help the article's conciseness.

Overall, it is good and interesting work that should be accepted following some minor adjustments.

Some minor observation:

Abstract:

Correct "In East Bay" to "In the East Bay"

Change "are way higher" to more technical terminology "are much higher"

Page 1, the term "mass-casualty earthquake" sounds a little dramatic for a technical publication. Consider revising.

The authors note that the the Alfred E. Alquist Hospital Seismic Safety Act was enacted in 1973 but twelve Pre-Alquist Act hospital buildings received red tags following the 1994 Northridge earthquake. Does this mean that the act was focused on new hospitals and existing hospitals were left as is? Please comment.

Also, the authors write 12 hospitals, but a few lines earlier, 11 were mentioned. Check that these are correct.

Does the "California Senate Bill 1953" refer to the year 1953 or just the number 1953? I presume not, so it would be better to specify the year so as to avoid confusion in the timeline of these legislative developments.

Results:

It may be better to specify that "The Bay" refers to the "The Bay Area of San Francisco" for unfamiliar readers.

Statements like "Our predictions show that many hospitals will experience violent shaking at a scale not seen in over a century." seem a little dramatic and unnecessary. It also conveys a distorted view to the public that the earthquake shaking can be predicted. In general, words like estimate would be a better reflection of the science behind the methods.

Equation 1. It seems "l'm" is a typo.

Would it not make more sense to define the shaking variable as $IM = im$ instead of $I = I$? It would maintain a consistency with the previous text.

Should the left hand term of Equation 2a contain the i term as $\pi(d, i)$? The formulation on the right hand side would suggest it should be independent of the intensity and listed simple as $\pi(d)$

The authors mention that the damage estimation in Equation 2b assumes independence. Is this realistic? Are there cases where this assumption may not hold? Are there alternative methods that could be adopted? Please comment.

Figure 6. The term D_1^n 's appears twice. One of them should be D_1^n .

Again, the authors use the term "state of the art" when referring to a ground motion model developed over a decade ago.

(Remarks on code availability)

Version 1:

Reviewer comments:

Reviewer #1

(Remarks to the Author)

This manuscript presents a numerical investigation into the anticipated hospital capacity and patient travel times in the San Francisco Bay Area, California, following a simulated M7.25 earthquake. The study focuses on expected travel times during the earthquake's recovery phase. A key innovation lies in integrating a risk model with a transportation network to elucidate potential cascading effects. The results demonstrate that the fragility of both healthcare facilities and transportation infrastructure significantly contributes to the probable elongation of patient travel times.

Compared to a previous iteration, this manuscript has undergone a comprehensive revision. Several critical assumptions employed in the earlier version have been relaxed, and the impact of the remaining assumptions is thoroughly discussed. A limitation of the study is the use of relatively generic vulnerability models to estimate the probability of specific damage levels for each hospital, given the expected ground shaking. While individual hospital buildings may exhibit fragility characteristics that deviate from these generic models, this approach provides a reasonable approximation considering the large number of buildings considered. Overall, the manuscript is suitable for publication in Nature Communications.

(Remarks on code availability)

The code reaches 3.4GB and exceeds the 2GB transfer limit set by the server. It is not practical to download and try to run it on the local computer. It would be much more convenient if the file size was reduced.

Reviewer #2

(Remarks to the Author)

I had the chance to reread the paper and review the author's responses to my original comment. I appreciate how the authors have elaborated on their work and clarified their contribution, and this manuscript is an improved version of what they submitted originally. For example, adding bridge failure analysis is a significant improvement. The authors also provided various clarifications on issues. They explained different confusing elements, including, for example, that their intent was not to use a value of 0.2 PGA as an indicator of significant damage to the buildings, and how a probability of 0.25 was not used as a cut-off for damage. Similarly, the use of HAZSU fragilities and their adjustment according to the SPC/NPC mapping provided a clearer reflection on the actions taken to obtain damage fragilities. However, it is essential to note that the validity of HAZSU facilities has been a subject of extensive discussions within the earthquake community. It is good to see, however, that they have been adjusted based on some logic.

While these improvements helped the paper in general, I still consider the analysis simplistic in many ways, and I struggled to see the novelty of the work. As a case study, the findings are indeed important and relevant to the city of San Francisco. However, I remain unconvinced about the novelty of the work and the study's motivation. Integrating transportation infrastructure into the modeling of post-earthquake hospital accessibility has been done numerous times, and publications with a similar motive appear in many journals worldwide, not only for seismic applications but also for almost all other natural hazards. The authors further elaborated on the study's motivation, noting that the San Francisco Bay area's uniqueness is characterized by a high concentration of seismically vulnerable hospitals, a complex and interconnected healthcare network that spans county and jurisdictional boundaries, and spatially heterogeneous seismic hazards and vulnerabilities across communities. These three elements are not unique to San Francisco. In fact, most major cities around the world exhibit similar characteristics, including a complex network of aging infrastructure. So, in my opinion, the argued uniqueness represents a form of baseline risk environment rather than a novel or case-specific anomaly. The authors further noted that the uniqueness is in that "this is the first study to evaluate the performance of a realistic regional healthcare network using publicly available, high-resolution data on both hospital infrastructure and transportation systems.". As I noted, numerous studies have investigated accessibility to healthcare on a large scale, which limits the uniqueness of your study (please see some examples below). Of course, the studies listed below are not "exactly" similar to yours in any way. Still, the overall idea of examining accessibility to healthcare systems on a large scale remains the same.

Post-earthquake health care service accessibility assessment framework and its application in a medium-sized city:

10.1016/j.res.2022.108782

Accessibility of medical services following an earthquake: A case study of traffic and economic aspects affecting the Istanbul roadway: <https://www.sciencedirect.com/science/article/abs/pii/S2212420918305314>

Measuring spatiotemporal accessibility to healthcare with multimodal transport modes in the dynamic traffic environment: <https://www.degruyterbrill.com/document/doi/10.1515/geo-2022-0461/html?lang=en>

I appreciated the elaboration on the intent of the paper by the authors and I also recognize the computational demand in integrating staff, stuff, and supplies in calculating bed availability at a single room level, which is the state of the art, as opposed to your approach where you assumed hospital damage as a proxy for bed availability. A better approximation would have been to perform that integration at the hospital level, rather than at the single-room level. Not only would this be computationally reasonable, but it would also give you a much better approximation of bed availability. Similarly, modeling building functionality as a Bernoulli process and not considering the state-of-the-art that integrates various elements to determine the functionality level adds to the uncertainty of the results. This gross approximation, combined with the hospital bed approximation, increases the uncertainty level in the results presented, rendering them somewhat questionable.

(Remarks on code availability)

Reviewer #3

(Remarks to the Author)

The authors have addressed all comments by this reviewer

(Remarks on code availability)

NA

Reviewer #1

Reviewers' comments:

Reviewer #1 (Remarks to the Author):

Overall comment:

This manuscript presents the results from a numerical study designed to evaluate the capacity of hospitals and the anticipated travel times for acute care patients in San Francisco Bay, California, following a potential M7.25 earthquake. This crucial and timely study innovatively integrates a risk model with a transportation network to illustrate the possible cascading effects. However, certain critical assumptions made within the model, which significantly impact the study's conclusions, lack sufficient justification. Details of these assumptions are outlined below. The manuscript requires significant revisions before it can be reconsidered for publication in Nature Communications.

Overall response:

We thank the reviewer for their thoughtful and constructive comments. We sincerely apologize for the delay in our response—this was due to major revisions made to both the manuscript and the computational framework to strengthen the analysis and address the issues raised.

In particular, we significantly expanded the scope of the study by fully integrating transportation infrastructure into the modeling of post-earthquake hospital accessibility. This addition allowed us to identify new critical outcomes—such as the potential isolation of a hospital and a community (“urban island”)—that were not captured in the original version of the manuscript.

We also conducted new sensitivity analyses to test the robustness of our results under different modeling assumptions, especially those related to damage thresholds and hospital functionality. These changes were made in direct response to the reviewer’s comments, which we found extremely helpful in improving the clarity and technical rigor of the work.

We carefully addressed each point raised in the review and implemented all suggested revisions. Reviewer comments are shown in **blue**, and corresponding changes in the manuscript are indicated in **red**. We are grateful for the reviewer’s input, which has led to a much stronger and more complete study.

The authors are advised to address the following while revising their manuscript:

Comment 1: [Page 5] The definitions adopted for the damage levels are not defined in the manuscript. As a result, it is difficult to interpret the plausibility of the damage probabilities

presented in Figure 2 as well as the post-earthquake capacity estimates (in Table 1, Figures 3 and 4). The definitions of the damage states and their impact of the type of damage on the capacity should be explained well.

We appreciate the reviewer's comment and agree that clearer definitions of the damage states are essential for interpreting the damage probabilities and the post-earthquake capacity estimates. In response, we have now added detailed definitions of the damage states for the three most common structural systems in our analysis—Steel Moment Frames, Concrete Shear Walls, and Steel Braced Frames—in *Supplementary Table 3*. We have also included definitions of non-structural damage for suspended ceilings and electrical-mechanical equipment—components that often drive hospital disruptions—in *Supplementary Table 4*.

To further address the reviewer's concern, we conducted a sensitivity analysis evaluating how hospital functionality changes under varying damage thresholds. We acknowledge that the actual thresholds are more complex in practice and can depend heavily on human decisions influenced by factors such as risk perception, preparedness levels, and patient surges. These behavioral and operational dimensions are still underexplored in the literature, but we hope our study encourages deeper investigation by the research community. We have also discussed these challenges in our related fieldwork-based publication following hospital visits in Turkey [1].

[1] Ceferino, L., Merino, Y., Pizarro, S. et al. Placing engineering in the earthquake response and the survival chain. *Nat Commun* 15, 4298 (2024). <https://doi.org/10.1038/s41467-024-48624-3>

Modifications to the main manuscript, including sensitivity analysis (line 169):

Our risk analysis predicts a substantial loss of hospital capacity throughout the Bay Area following the earthquake scenario (see Methods). We estimate a loss of 8,165 hospital beds, resulting in only 51% of total acute care beds remaining functional (standard deviation: 21%). This significant reduction in capacity is highly uneven across the region (Figure 3, Table 1). Our baseline predictions assume hospitals lose functionality when structural or non-structural damage exceeds a threshold of slight damage (see Methods). These damage conditions are often shown to disrupt hospital services (see descriptions of multiple damage levels in Supplementary Tables 3 and 4). [21, 28–31] We also analyzed more optimistic scenarios with moderate (“favorable”) and extensive (“idealistic”) damage thresholds, resulting in increased overall functionality of 79% and 93%, respectively (Figure 3 and Supplementary Table 5). However, empirical evidence supports our baseline assumption, as hospitals often lose operations at early stages of damage. [21, 28–31]

Figure 3. Post-earthquake functionality of hospitals and bridges across the Bay Area. [...] Probability distribution of functional acute care beds under baseline (slight), favorable (moderate), and idealistic (extensive) damage thresholds.

Additional changes in Supplementary Information:

Supplementary Table 3. Definitions of structural damage states for most common structural types in the hospital building portfolio: S1: Steel Moment Frame; C2: Concrete Shear Walls; S2: Steel Braced Frame. These definitions come from HAZUS and were adopted in this study.⁴

[Table Redacted]

Supplementary Table 4. Definitions of non-structural damage states for two acceleration-sensitive components whose failures often disrupt hospital services. *These definitions come from HAZUS and were adopted in this study.*⁴

[Table Redacted]

Editorial Note: The map data in the figure on this page of the Peer Review file is from © OpenStreetMap contributors openstreetmap.org/copyright.

Comment 2: [Page 8, Fig. 3, lower left] The legend entry related to markers with small white circles is missing.

Thank you for catching this. We have added “beds” next to the white circles in the legend. We also added an extra line on the caption:

“The size of the circles indicates the number of functional beds before and after the earthquake.”

Please see the new Figure:

Comment 3: [Page 10, Fig. 4, upper right] The figure legend is missing.

We added the legend to the pre-earthquake and the post-earthquake cases. We also adjusted the figure’s caption to make it clearer:

Post-earthquake (Post-EQ) increases in travel time at the zip-code level compared to pre-earthquake (Pre-EQ) conditions, highlighting densely populated areas that are most severely impacted.

Please see the updated figure:

Comment 4: [Page 11] In the last paragraph, all "Figure 4"s should read "Figure 5".

Fixed

Comment 5: [Page 12] The discussion related to the transportation network is difficult to follow since it requires knowledge of the names of the places. It would be very informative to add labels on the maps (e.g. Figure 1) or to cite the numbers indicated in Figure 4 in the text so that the non-residents can follow the observations related to cascading effects with ease.

This is an excellent point. We added the names of the counties in Figure 1. We also adjusted the figure's caption to make it clearer:

The Bay Area's county names are also included for reference.

Please see the updated figure:

Editorial Note: The map data in the figure on this page of the Peer Review file is from © OpenStreetMap contributors openstreetmap.org/copyright.

Comment 6: [Page 15] In Equation 1, "P_IM(l'm)" should read "P_IM(im)"

Fixed

Comment 7: [Page 16] It is stated that "structural and non-structural damage (D_{ks} and D_{kn}) within a building k are conditionally independent given their respective shaking intensities (I_{ks} and I_{kn}).". Physical mechanisms that drive non-structural damage, as well as past reconnaissance observations, indicate the opposite. Given a specific shaking intensity, the non-structural damage is very sensitive to the extent of structural damage. The level of structural damage has a direct influence on the acceleration and deformation demands on the non-structural components as well as the changes in the resistance capacities at the anchorage locations.

This comment touches on an important aspect of the model, and we appreciate it. We spent quite a bit of time thinking through this assumption when developing our approach. We believe the independence assumption is reasonable in the context of hospital failure, where functionality is often lost at early damage stages (e.g., slight damage). At these levels, we think it's reasonable to assume that the structural system hasn't experienced enough damage to significantly alter the building's dynamic properties or the demands on non-structural components.

In other words, since the hospital is assumed to stop functioning once slight damage is reached, we're operating in a space where the interaction between structural and non-structural damage

may be less critical. We didn't find specific studies that test this hypothesis directly, but we think the assumption is appropriate given how early in the damage process hospitals can fail.

We've now added a brief explanation in the manuscript to clarify this point and acknowledge that future studies could explore the effects of correlated damage mechanisms in more detail, especially in the later stages of damage (line 393).

Although non-structural damage may not be conditionally independent of structural damage—given that structural damage can alter the dynamic properties of a building and, in turn, affect demands on acceleration-sensitive non-structural components—we consider the independence assumption to be reasonable within the failure space of hospitals. Specifically, our model assumes that hospital functionality is lost once early damage thresholds (e.g., slight damage) are exceeded; at this stage, structural degradation is unlikely to significantly modify the system's dynamic characteristics. While we are not aware of empirical studies explicitly validating this hypothesis, the assumption allows us to simplify the model without compromising its ability to capture key failure mechanisms relevant to hospital functionality.

Comment 8: [Page 16] Building functionality is represented using a Bernoulli random variable which can only have two states: building is functional or not functional. This implies that if there is slight non-structural damage in some part of a hospital building, the entire building will become completely dysfunctional. At present, the definition of slight damage adopted in the manuscript is unclear. However, for the typical definition of slight damage, this is not a realistic assumption. This is particularly true after severe disasters where there is a significant demand for urgent acute care. Many of the hospital buildings with slight non-structural damage would continue to operate with some reduced level of functionality. In order to address this issue, the definitions of damage states should be explained. Moreover, the assumption of complete loss of functionality of the entire building needs to be justified based on the provided damage definitions and past experiences. Otherwise, the formulation should be revised accordingly.

Dear reviewer, thank you for raising this point. Now, we include full descriptions of the damage and a sensitivity analysis for different damage thresholds. We believe our response to Comment 1 should also address this comment.

Comment 9: [Page 18] Difficulties associated with the analytical evaluation of $\pi(f)$ and $\pi(bt)$ are essentially very similar. The related sentences can be collected together into a single paragraph so as not to repeat very similar issues twice.

Thank you very much for the constructive suggestions. This section has been revised to be more concise to the following version (line 450):

Regional seismic risk analysis is a high-dimensional problem. While the joint distribution $\pi(f, i)$ in Eq. 3 can be analytically evaluated due to closed-form expressions for $\pi(f|i)$ and $\pi(i)$, computing the marginal distribution $\pi(f)$ is not analytically tractable. This is because it involves integrating over a set of correlated random variables in i , for which no closed-form marginalization exists. Numerical integration methods (e.g., Riemann integration) quickly become computationally infeasible, even for a modest number of buildings. To address this, several approaches have been proposed to efficiently approximate $\pi(f)$ in large-scale settings. For example, Ceferino et al. demonstrated—both theoretically and empirically—that under certain conditions, Central Limit Theorem (CLT)-based approximations can accurately and efficiently estimate $\pi(f)$.

Heresi and Miranda showed that similar approximations remain valid even under mild correlation structures in seismic risk models. However, in situations where the CLT does not apply, modern risk models often rely on Monte Carlo methods to estimate $\pi(f)$. In our case, we seek to estimate the distribution of bt , the total number of functional beds across hospital buildings, where functionality follows the distribution $\pi(f)$. Since the number of hospital buildings varies widely—from a few to many—CLT approximations are not generally applicable. Therefore, we use Monte Carlo simulation to evaluate $\pi(bt)$. We implemented this using the NHERI SimCenter's R2D Tool, which enables scalable regional risk assessments. Specifically, we used R2D to generate 5,000 realizations of i and d for both hospitals and bridges, which form the basis for evaluating Eq. 2.

Comment 10: [Page 19] Structural Performance Categories (SPCs) of the hospitals have a critical function in the presented investigation. At present, it is unclear if these categories are the revised categories obtained as a result of the HAZUS Reassessment Program for SPC-1 buildings or not. In general, the technical basis of this categorization should be clarified.

Thank you for pointing this out. We utilized the official SPC rating from the California Department of Health Care Access and Information (HCAI). Thus, hospitals that went under the reassessment and passed it were considered SPC-2. Fortunately, we did not have to do the reassessment analysis as the data provided by HCAI already pointed to the latest building rating.

To make this clearer, we added this explanation to Supplementary Table 1 for SPC 2:

*Supplementary Table 1. Structural Performance Categories (SPC) according to California's Department of Health Care Access and Information*³

[Table Redacted]

We also added more clarity to the main manuscript regarding the SPC and made the reference to the table shown before (line 117):

In this paper, we used the Structural Performance Categories (SPC) and Non-structural Performance Categories (NPC), established and reported by California's Department of Health Care Access and Information (HCAI), to characterize structural and non-structural deficiencies.[44] Each hospital building receives vulnerability ratings (SPC or NPC) ranging from 1 to 5, from the most to the least vulnerable. Supplementary Tables 1 and 2 fully describe SPC and NPC.

Comment 11: [Page 19] Authors state that "Following the definitions of SPC ratings (Supplementary Table 1), we mapped SPC 1 and 2 to pre-code and moderate-code fragility functions for regular buildings from HAZUS ...". This is a critical assumption which would have a direct impact on all calculations. The assumed fragility parameters need to be justified. For this purpose, the extent of the structural and non-structural damages observed in hospitals designed

and constructed in other parts of the state (or other states with similar design codes) that experienced earthquakes within the last 50 years can be considered as a basis. If the fragility characteristics show a similar trend to the assumed model, the selected parameters can be considered as suitable. At present, the introduced assumptions are not justified based on any empirical or analytical evidence.

Comment 12: Thank you for raising this important point. We agree that the assumption of mapping SPC 1 and 2 to HAZUS pre-code and moderate-code fragility functions plays a central role and needs a clearer justification.

Our main rationale is that the SPC classification itself was developed by engineers specifically to evaluate seismic risk in California hospitals. SPC 1 and 2 buildings are known to be older and more vulnerable structures—mainly designed before modern seismic codes were in place (pre-1973)—and have been flagged by the state as not acceptable for continued acute care use in future earthquakes. These characteristics align closely with the assumptions underlying HAZUS pre-code and moderate-code fragility functions, which were developed to represent similarly vulnerable building stock.

In response to your comment, we reviewed post-earthquake damage reports from events such as the 1971 San Fernando and 1994 Northridge Earthquakes. In many cases, hospitals with older structural systems—such as non-ductile concrete frames—exhibited significant structural and non-structural damage. Although this information is too limited and variable to develop fragility functions directly and use them in our models, it does support the idea that SPC 1 and 2 facilities represent a more vulnerable class of buildings. This general vulnerability is consistent with the assumptions behind HAZUS pre-code and moderate-code fragility functions. In fact, HAZUS recommends that “older buildings located in High-Code seismic design areas should be evaluated using damage functions for either Moderate-Code buildings or Pre-Code buildings (page 5.74 of HAZUS technical manual).

We have now added this explanation to the manuscript and cited relevant reconnaissance reports. We also acknowledge the limitation of this assumption and hope that future work will enable the development of empirically calibrated fragility models for SPC-rated hospitals (line 500).

We assessed structural damage across a variety of structural system types and five levels of structural vulnerability (Figure 3 and Supplementary Figure 1). To represent this range, we used and adapted structural fragility functions from HAZUS (FEMA, 2020), also available through the R2D tool (Zsarnóczy et al., 2020). Based on the definitions of SPC ratings (Supplementary Table 1), we mapped SPC 1 and 2 buildings to the HAZUS pre-code and moderate-code fragility functions, respectively, using the corresponding α and β parameters in Equation 1. This mapping reflects the fact that SPC 1 and 2 buildings do not comply with the structural provisions of the Alquist Hospital Facilities Seismic Safety Act and are known to be older and more vulnerable structures. While no empirical fragility curves directly link SPC ratings to seismic performance, post-earthquake damage reports from events such as the 1971 San Fernando and 1994 Northridge earthquakes show that hospitals with older structural systems experienced significant

damage. Although these observations are not sufficient to derive fragility functions, they support the alignment of SPC 1 and 2 facilities with more vulnerable construction classes, consistent with HAZUS classifications.

Comment 12: [Page 20] Authors note that they "utilize a proxy for a representative PFA equal to the spectral acceleration at the building's period of vibration.". The assumption of peak floor acceleration being the same as the spectral acceleration at the site of the building is not a suitable one. Such an assumption ignores the height of the building, the type of the system, and any irregularity in the structural layout. These factors significantly influence the resulting floor accelerations.

Thank you for pointing out that floor accelerations are influenced by several building-specific factors. We revised our approach based on this suggestion. A key challenge is that existing methods for estimating peak floor accelerations typically require solving dynamic equations and rely on detailed building information. These methods are more suitable for individual building analyses, rather than regional assessments involving large building portfolios (e.g., over 700 facilities in our case), where such detailed information (e.g., to capture irregularities) is generally unavailable.

To improve our model in this regional context, we refined our original approach to better account for building height and structural system. Specifically, we now assume that floor acceleration varies linearly with height, starting from PGA at the base and reaching the spectral acceleration (S_a) at the building's effective height. This approach is consistent with procedures for estimating acceleration demands in ASCE 7.

In our disaster risk model, both PGA and spectral acceleration are estimated using ground motion prediction equations. The spectral acceleration is based on the structural type and height of each hospital building, with periods of vibration estimated from the HAZUS exposure model. The effective height is computed using the ratio between the equivalent height and total height. We use the HAZUS exposure model to estimate these values, which also depends on structural type and number of stories. We then compute the average floor acceleration across the building from this linear profile and use that value in our analysis.

This updated method resulted in slightly lower probabilities of non-structural damage compared to the previous version of the model, which assumed PFA equal to S_a (i.e., from an equivalent single-degree-of-freedom system) across the entire building height. The revised estimates are lower because the average acceleration across the height is less than S_a , especially since the effective height is typically above mid-height.

We edited our manuscript according and explained the following (line 551):

PFA varies with height within a building, and total non-structural damage depends on how acceleration-sensitive components are distributed across floors. However, detailed data on the floor-by-floor distribution of non-structural elements are rarely available in regional-scale analyses. To address this limitation, we used a simplified proxy to approximate the variation of PFA with building height while incorporating key building characteristics. Specifically, we assumed a linear distribution of floor accelerations, with the base acceleration set equal to the peak ground acceleration (PGA) and the acceleration at the effective height equal to the spectral acceleration corresponding to the building's dynamic period. The effective height is estimated for each structural type based on period and height values from the HAZUS exposure model. We then compute the average PFA across the building height from this linear profile and use an average value across height in our non-structural damage analysis. Further studies can enhance the fidelity of this analysis if higher-resolution information for non-structural components is available, e.g., on each floor.

Comment 13: [Page 22] It is stated that the number of patients is assumed to be proportional to the population. Past disasters show that the number of acute care patients is typically distributed proportional to the intensity of building damage together with the population density. The building damage is a function of shaking intensity and vulnerability of the residential or commercial building stock. In the proposed approach, it appears that these critical factors are not taken into account while modeling the distribution of patients with acute care needs following an earthquake. This assumption needs to be justified or the approach needs to be revised. The assumptions related to the spatial distribution of patients over the region are expected to have a considerable impact on the calculated travel times and the related conclusions.

Thank you for pointing this out. We spent significant time considering whether to explicitly model injury distributions from earthquakes in the Bay Area as part of this study. Ultimately, we decided not to proceed for two main reasons.

First, our goal in this analysis was to document hospital accessibility losses beyond the emergency response period. Hospitals often remain closed for months or even years after an earthquake, and understanding long-term access disruptions is critical for recovery planning. Patient surges, on the other hand, typically last only a few days—except under extreme conditions, such as the 2023 Türkiye Earthquake, where demand persisted for three weeks.[1] While we have modeled injuries in other regions with high expected casualties, such as Peru,[2] we expect a much smaller emergency phase in the Bay Area. For this reason, we focused on population-weighted access losses rather than explicitly modeling injury locations. We have clarified this point in the revised manuscript (see line 645).

Our goal was to quantify hospital accessibility losses beyond the immediate emergency period, recognizing that hospitals often take months or even years to be repaired. Because people are typically injured within damaged buildings, patient surges after earthquakes generally last only a few days—except in rare extreme cases such as the 2023 Turkiye Earthquake. To focus on long-term access rather than short-term emergency response, we did not model injury locations directly. Instead, we assumed that patient demand is proportional to population distribution and modeled accessibility accordingly. For the Bay Area, we based our analysis on population-weighted access loss and identified the network’s vertices closest to the centroids of census tracts to define the set p (Figure 8).

Second, the state of the art in earthquake casualty modeling has not advanced significantly in recent decades, and predictions remain highly uncertain—especially in places like California, where recent damaging earthquakes have not produced large casualty datasets. We are currently leading an NSF-funded project (NSF Award #2410291) to address this gap, with a focus on developing high-fidelity, empirically grounded casualty models for the Bay Area. This includes new fieldwork and modeling efforts that will inform a dedicated paper focused on spatial injury distributions and emergency care needs. We plan to incorporate the reviewer’s suggestion as part of that forthcoming work.

[1] Ceferino, L., Merino, Y., Pizarro, S. et al. Placing engineering in the earthquake response and the survival chain. *Nat Commun* 15, 4298 (2024). <https://doi.org/10.1038/s41467-024-48624-3>

[2] Ceferino, L., Mitrani-Reiser, J., Kiremidjian, A. et al. Effective plans for hospital system response to earthquake emergencies. *Nat Commun* 11, 4325 (2020). <https://doi.org/10.1038/s41467-020-18072-w>

Comment 14: [Page 22] It appears that the potential for any closure of roads after the earthquake is not considered in the proposed framework. Past earthquakes have shown that havoc-related jams, toppling of electric lines and commercial pole signs, landslides, surface fault ruptures, underground pipe ruptures, building façade debris, damages to bridges, flyovers, and tunnels have led to the closure of roads after earthquakes. In urgent cases, the roads are closed as a part of the disaster response operations and only the emergency vehicles are allowed on the main roads. Furthermore, very often, a scarcity of gas is caused after such earthquakes by the demand surge, which cannot be accommodated quickly. As a result, it becomes a common problem to have vehicles on the sides of the roads left without any gas. In the case of California, where electric vehicles are more common as compared to other regions, the post-earthquake functioning of the grid and the likely charging durations may also have a critical impact. In the present investigation, the potential impacts of such effects on the travel time estimates, as well as transportation network connectivity, are not taken into account. These should be noted as a limitation of the study to inform the reader properly about the complexity of the subject matter.

Dear Reviewer, we fully agree with your comment and appreciate the detailed observations. After reading your reviews (along with those from the other reviewers), we invested significant effort into improving our modeling of transportation infrastructure disruptions in the Bay Area. This became a major component of our revision and is the main reason for the time it took us to respond.

In this revised version, we integrated detailed information for over 5,000 bridges in the Bay Area and significantly expanded the transportation network model, which now includes approximately 1.5 million edges and 0.5 million nodes. This refinement substantially increased computational demands: the 5,000 Monte Carlo simulations required for the analysis took about two weeks to run on a laptop equipped with an Apple M2 chip and 64 cores. We ran multiple iterations to ensure the model was fully debugged and stable.

Thanks to this updated formulation—and your review—we are now able to capture how hospital and transportation failures interact to influence post-earthquake hospital accessibility. Our revised analysis highlights several key cascading effects:

- Widespread transportation failures lead to compound access losses, with travel times increasing by over a factor of 10 in some areas.
- Some hospitals in the Bay Area may become completely isolated due to bridge failures.
- Urban communities—particularly in Fremont and Novato—may lose access to essential services due to localized bridge disruptions.

We acknowledge that other important factors raised in your comment—such as debris-blocked roads, emergency vehicle prioritization, fuel shortages, electric vehicle limitations, and traffic congestion—are not included in this simulation. These are highly relevant and could further exacerbate accessibility losses. However, due to a lack of standardized methods and data to model these phenomena at a regional scale, we did not include them in this analysis. We have now added sentences in the manuscript (in the discussion) explicitly stating these limitations and encouraging future work in this direction.

Despite these limitations, we believe this revised version represents an important step toward understanding the interdependencies between hospital and transportation networks in post-earthquake accessibility. We sincerely thank you for pushing us to deepen the analysis. The changes we made based on your feedback have significantly strengthened the study.

We include some of the updated results and manuscript revisions below.

Abstract (line 15):

By integrating seismic hazard with hospital and transportation infrastructure’s vulnerability and connectivity data, we analyze 76 hospitals (426 buildings with 16,639 beds) and 5,163 bridges within a vast network of ~1.5 million edges and ~0.5 million nodes [...] Widespread transportation failures further restrict access, increasing regional travel times by 177% and exceeding 1000% in parts of East Bay, potentially fully isolating hospitals and entire urban communities. These findings underscore the urgent need for resilient healthcare and transportation infrastructure to mitigate life-threatening disruptions following major earthquakes.

Introduction (line 80):

To fill this critical gap, we investigate, for the first time, joint failures of healthcare and transportation networks across the San Francisco Bay Area after a major earthquake. By examining diverse urban communities near active seismic faults—often underrepresented in previous studies—this research provides valuable insights into post-earthquake healthcare access, particularly relevant for earthquake-prone regions worldwide with diverse populations and similarly vulnerable hospitals and bridges, such as those in the U.S., Japan, and Türkiye.

To our knowledge, this is the first study to evaluate post-earthquake healthcare access at this scale using publicly available data on both hospital and transportation infrastructure. Whereas prior work has focused on individual facilities[20-23] or small systems[42-46]—often relying on synthetic data—our approach couples regional seismic risk models with large-scale network analysis to capture interdependent failures across an entire metropolitan area. By integrating vulnerability data from 426 hospital buildings and 5,163 bridges, and simulating disruptions across a transportation network with approximately 1.5 million edges and 0.5 million nodes, we offer one of the most extensive assessments to date of how earthquakes affect healthcare access. This study advances the state of the art in regional risk modeling globally, offering a scalable, computationally intensive framework that supports risk-informed decision-making in other high-risk urban areas.

New Section on Post-earthquake Bridge Functionality (line 196):

Our risk analysis model predicts that 1,469 bridges out of 5,163 would be damaged due to the earthquake scenario (see Methods). Consequently, 3,693 bridges (72%, standard deviation: 17%) are expected to remain undamaged and fully operational (Figure 3). Some damaged bridges could continue operating, albeit at reduced capacities. Under our “baseline scenario”, where only bridges with at most slight damage remain operational, a total of 3,964 bridges (77%, standard deviation: 15%) would function. In a more optimistic “favorable scenario”, where bridges with at most moderate damage also remain partially operational, 4,069 bridges (79%, standard deviation: 14%) could provide service. In an idealistic scenario, which assumes that bridges even with extensive damage retain emergency functionality, 4,527 bridges (88%, standard deviation: 11%) would remain operational.

Editorial Note: The map data in figure 3 in this Peer Review file is from © OpenStreetMap contributors openstreetmap.org/copyright.

Similar to hospitals, Alameda County faces the most severe transportation impacts (Figure 3, Table 2). Under the baseline scenario, Alameda retains only 44% functional bridges (from 642 down to 282). Marin County is second most impacted, retaining 64% (from 195 to 125 bridges). Additionally, estimates of reductions in bridge travel speeds (see Methods)—calculated from their damage probabilities—highlight Alameda’s higher vulnerability due to proximity to the Hayward Fault rupture (Figure 3, Table 2).

Figure 3. [...] Lower left: Probability distribution of fully functional bridges under baseline (slight), favorable (moderate), and idealistic (extensive) damage thresholds. Lower right: Spatial distribution highlighting bridges with post-earthquake travel capacities below 75%. Results generated from 5,000 Monte Carlo simulations.

Revised findings on Accessibility (line 223):

At a more detailed neighborhood scale, these disruptions are even more pronounced (Figure 4). We identified six densely populated zip codes (each exceeding 15,000 residents, labeled #1 to #6 in Figure 4) facing substantial reductions in healthcare access. The most severely impacted is Novato (#1, Marin County), where travel times increase nearly 25-fold, from 7.3 to 185.5 minutes, affecting approximately 18,000 residents. Similarly, a Fremont zip code (#6, Alameda County) with roughly 66,000 residents experiences an eightfold increase (from 8.4 to 42.2 minutes). Other significantly impacted zip codes in Richmond (#4), Oakland (#2 and #5, Alameda County), and San Jose (#3, Santa Clara County) face travel time increases of 10, 11, 9, and 11 times their pre-earthquake levels, respectively. These results underscore the dramatic reshaping of healthcare access throughout East Bay communities along the earthquake rupture.

Finally, we explored the relative contributions of hospital and transportation infrastructure disruptions through scenario analyses (Figure 4). We compared the baseline (None, no infrastructure disruptions) against scenarios modeling probabilistic damage only to transportation

infrastructure (Transp., hospitals fully resilient), only to hospital infrastructure (Hosp., transportation fully resilient), and simultaneous probabilistic disruptions to both (Transp. & Hosp.). When considering disruptions only to transportation infrastructure (Transp.), average Bay Area travel times increase by 41% (from 6.1 to 8.6 minutes), notably less than the combined scenario (Transp. & Hosp., 177% increase). Conversely, modeling only hospital infrastructure disruptions (Hosp.) yields a 79% increase (from 6.1 to 10.9 minutes), highlighting greater relative fragility in hospital networks.

At localized urban scales, the compounding effects of simultaneous transportation and hospital infrastructure failures lead to disproportionately severe impacts (Figure 4). For example, as mentioned earlier, in the most severely impacted zip code (Novato, labeled #1 in Figure 4), travel times dramatically increase from 7.3 to 185.5 minutes under combined disruptions (Transp. & Hosp.), compared to only 14.1 minutes when transportation disruptions are excluded (Hosp. only). Similarly, for other highly affected zip codes (labeled #2 to #6 in Figure 4), travel-time ratios notably decrease—from 11, 11, 10, 9, and 8 under combined disruptions, to 6.4, 3.1, 6.1, 4.7, and 3.4 when transportation disruptions are not considered—highlighting the critical influence of compounding infrastructure failures on healthcare access.

Editorial Note: The map data in figure 4 in this Peer Review file is from © OpenStreetMap contributors openstreetmap.org/copyright.

Figure 4: Access to functional acute care hospitals in the Bay Area after the earthquake. Upper left: Post-earthquake (Post-EQ) increases in travel time at the zip-code level compared to pre-earthquake (Pre-EQ) conditions, highlighting densely populated areas that are most severely impacted. Upper right: Ratios of post- to pre-earthquake travel times to the closest acute care hospitals. Lower right: Distribution of travel times per zip code under four scenarios: None (no infrastructure disruptions), Transp. (only transportation infrastructure is vulnerable), Hosp. (only hospital infrastructure is vulnerable), and Transp. & Hosp. (both transportation and hospital infrastructures are vulnerable). Lower left: Ratios of post- to pre-earthquake travel times when only hospital infrastructure (Hosp.) is vulnerable. Results generated from 5,000 Monte Carlo simulations.

New section on Isolated Communities and Hospitals (line 252):

To understand the drivers behind the sharp increases in travel times observed earlier, we analyzed travel volumes to the closest acute care hospitals before and after the earthquake. Figure 5 shows pre-earthquake road usage, with line thickness representing travel volume. Roads in brown highlight those whose usage falls above the 90th percentile based on pre-earthquake conditions, emphasizing the most critical travel routes. As expected, the highest-usage roads prior to the earthquake were located near hospitals, especially in densely populated areas with fewer hospitals. For instance, communities in Fremont (#6 in Figures 4 and 5) are served by only two hospitals, while Richmond (#4) relies on just one.

Figure 5 also displays post-earthquake road usage, using the same pre-earthquake 90th percentile threshold to highlight heavily traveled roads. After the earthquake, patient travel patterns shift significantly, with many communities forced to travel longer distances along major corridors. For example, residents in Oakland (#2 and #5 in Figures 4 and 5), who previously relied on nearby hospitals, will now have to travel west to the San Francisco Peninsula or east to Walnut Creek due to simultaneous disruptions in both hospital and transportation infrastructure. These shifts help explain the steep increases in travel times observed in East Bay communities along the rupture zone.

Four major bridges connect the East and West Bay: the Richmond–San Rafael, Oakland–San Francisco Bay, San Mateo–Hayward, and Dumbarton bridges. Before the earthquake, no patients needed to cross these bridges to reach their nearest acute care hospital (Figure 5). After the earthquake, however, these bridges become critical, with travel volumes increasing to 3.6, 9.3, 6.3, and 13.6 times the pre-earthquake 90th percentile values—reflecting the large-scale redistribution of healthcare demand.

Our analysis also revealed a more complex post-disaster phenomenon: micro-scale isolation of hospitals and communities due to localized bridge failures. For example, the hospital serving the most severely impacted zip code in Novato (#1 in Figures 4 and 5), which has 64 beds, depends entirely on a single bridge for access. If this bridge fails, the hospital becomes inaccessible, increasing travel times from 7.3 to 185.5 minutes for the surrounding population.

More concerning, we found that urban neighborhoods near Fremont (#6) could become fully isolated (Figure 5). In one area, failure of three bridges would trap one part of the neighborhood; in another, a different set of three bridge failures would isolate the remaining part. These communities would lose access not only to hospitals but also to other essential services such as grocery stores and pharmacies—illustrating how cascading infrastructure failures can sever lifeline access in disaster scenarios.

Editorial Note: The map data in figure 5 in this Peer Review file is from © OpenStreetMap contributors openstreetmap.org/copyright.

Figure 5: Mobility to reach the closest functional acute care hospital in the San Francisco Bay Area. Upper left: Pre-earthquake travel volumes to the closest acute care hospitals. Numbers 1 to 6 indicate the hospitals serving the dense zip codes with the highest post-earthquake increases in travel time. Upper right: Projected travel volumes after the earthquake, showing major shifts in patient mobility. The red-shaded corridor illustrates how Oakland residents (#2 and #5) must travel west to the Peninsula or east to Walnut Creek due to hospital and transportation disruptions. Lower left: The hospital serving zip code #1 (Novato) relies entirely on a single bridge; if the bridge fails, all access to its 64 beds is lost. Lower right: Community near zip code #6 (Fremont), where the southern and northern sections become fully isolated if three different bridges fail in each area. Results are based on 5,000 Monte Carlo simulations of post-earthquake patient mobility.

Comment 15: [Supp. Fig. 3] The damage probabilities are compared against a "reference" fragility model for structural wall buildings. However, the original source of this reference model, as well as the related building typology, is unclear. Therefore, it is not easy to interpret the meaning of this comparison.

Thank you for pointing this out. We fully agree with the reviewer. This plot was confusing and did not add much relevant information. We removed it.

Comment 16: Reviewer #1 (Remarks on code availability):

I downloaded the code and checked its contents very briefly. It contains the related files. I did not run it to reproduce the results.

We have uploaded updated code to the same data repository. We carefully prepared a README with step-by-step instructions to ensure that all analysis results presented in the paper can be reproduced by running the provided code. This updated version contains the new model version.

Reviewer #2

Reviewer #2 (Remarks to the Author):

Overall response:

We thank the reviewer for their feedback and recognize that the previous version of the manuscript did not sufficiently convey the motivation and contributions of our study. We have taken the comments seriously and used them to substantially revise and strengthen the work.

We also note that concerns raised in this review, especially regarding the motivation and novelty of the study, were not shared by the other reviewers, whose feedback was more positive and encouraging. Nonetheless, we carefully considered all reviewer input and undertook a major revision effort.

The delay in resubmission reflects the time required to expand the analysis and clarify our contributions. In particular, we significantly broadened the scope of the study by fully integrating transportation infrastructure into the modeling of post-earthquake hospital accessibility. This addition allowed us to identify new critical outcomes—such as the potential isolation of a hospital and a community (“urban island”)—that were not captured in the original version. We also introduced new metrics of accessibility, conducted sensitivity analyses, and revised the introduction and discussion to better position the study within the broader literature.

All specific responses are provided in **blue** text, and corresponding changes in the revised manuscript are shown in **red**. We hope these revisions demonstrate the rigor and value of the updated study.

Comment 1: As the authors know, one of the most critical elements of any study is the motivation of the work. The authors motivated the work by indicating that, unlike the past studies on hospital networks, this study focuses on a community exposed to earthquakes only within a few kilometers of them. First, the papers cited by the authors include hospitals a few kilometers from the hazard. Second, the justification overall is very weak and does not represent uniqueness. The study is motivated by the hospital being a few kilometers from the hazard, which is irrelevant - not to mention that this was the case in past studies. Furthermore, assessing healthcare networks under earthquakes has been studied quite a bit using more rigorous approaches. It is hard to see valuable contributions in this study.

Response 1: Thank you very much for the constructive feedback. We appreciate the opportunity to clarify the motivation and contribution of our work. We have revised the introduction of the manuscript to better emphasize these points.

Our motivation stems from the unique combination of challenges present in the San Francisco Bay Area, including:

- (a) a high concentration of seismically vulnerable hospitals, many of which are pre-1973 structures located near active faults;
- (b) a complex and interconnected healthcare network that spans county and jurisdictional boundaries; and
- (c) spatially heterogeneous seismic hazards and vulnerabilities across communities.

This combination creates critical management challenges. For example, our results show that hospitals located near rupture faults face higher risks of damage, and their potential loss would disproportionately reduce access to care in nearby communities. From the perspective of regional healthcare managers (e.g., HCAI, formerly OSHPD), addressing earthquake impacts requires a regional risk approach—one that goes beyond individual building assessments to account for system-wide hospital vulnerabilities and transportation disruptions, as modeled in the current version of this study, to capture changing access patterns across the network.

We have clarified in the manuscript (see changes to Introduction below) that the intent is not to claim uniqueness based solely on hospital proximity to faults. Rather, we emphasize that this proximity, combined with widespread vulnerability and regional interdependence, demands a systems-level analysis. To our knowledge, this is the first study to evaluate the performance of a realistic regional healthcare network using publicly available, high-resolution data on both hospital infrastructure and transportation systems.

In terms of methodology, we recognize that high-fidelity simulation models have been developed to study the earthquake dynamics of individual hospitals. These are valuable for understanding detailed functional disruptions but are not feasible for regional-scale analyses due to their intensive data requirements and computational demands. In contrast, our model is designed to capture interdependent failures—such as simultaneous damages to hospitals and critical bridges—at a regional scale, while balancing spatial coverage, data availability, and computational feasibility.

In our previous work, we have developed discrete event simulation models to capture detailed hospital functionality, which can be coupled with structural models to analyze patient flow and operations at the building level [1]. However, applying such models across an entire region remains impractical. This study therefore adopts a campus-level functionality model, allowing us to compare hospital capacity losses and assess how those losses—combined with bridge damage—affect access to care.

Our regional modeling approach represents the state of the art in large-scale healthcare system analysis, integrating detailed hospital and transportation vulnerability data, seismic hazard models, and transportation network connectivities to assess post-earthquake access to care. To achieve this level of integration and fidelity using publicly available data, we developed a modeling and computational framework capable of simulating interdependent failures across hundreds of facilities and thousands of network links.

These simulations are computationally intensive, requiring approximately two weeks to run, which underscores the technical effort involved in performing this level of analysis and highlights the practical challenges of scaling such models to other high-risk urban regions.

Finally, while we appreciate the reviewer's concerns, we respectfully disagree with the assessment that the study lacks value. We believe the scale, integration, and policy relevance of our findings—such as the identification of communities with disparate access to healthcare and even isolated hospitals and communities—represent a meaningful contribution to the literature. If there are specific references the reviewer believes have already addressed this problem at a similar level of scope and resolution, we would be happy to engage with them. Based on our comprehensive literature review, we are confident that this study advances the state of the art in regional healthcare system resilience under seismic risk.

Please see the new version of the Introduction that is revised to highlight better our contributions according to your comments (lines 50, 57, 63, and 86):

Recognizing these vulnerabilities, this paper investigates the effects of future earthquakes on healthcare access in the entire San Francisco Bay Area, California. [...]

Yet, hospitals do not function in isolation—particularly in dense urban areas [...].

Failures in transportation infrastructure significantly compound healthcare disruptions [...] These interconnected disruptions underscore the urgent need to evaluate joint vulnerabilities within healthcare and transportation infrastructure.

To our knowledge, this is the first study to evaluate post-earthquake healthcare access at this scale using publicly available data on both hospital and transportation infrastructure. Whereas prior work has focused on individual facilities[20-23] or small systems[42-46]—often relying on synthetic data—our approach couples regional seismic risk models with large-scale network analysis to capture interdependent failures across an entire metropolitan area. By integrating vulnerability data from 426 hospital buildings and 5,163 bridges, and simulating disruptions across a transportation network with approximately 1.5 million edges and 0.5 million nodes, we offer one of the most extensive assessments to date of how earthquakes affect healthcare access. This study advances the state of the art in regional risk modeling globally, offering a scalable, computationally intensive framework that supports risk-informed decision-making in other high-risk urban areas.

[1] Merino Y, Ceferino L, Pizarro S, de la Llera JC. Modeling hospital resources based on global epidemiology after earthquake-related disasters. *Earthquake Spectra*. 2024;40(4):2402-2429. doi:10.1177/87552930241262788

Comment 2: In addition to the lack of sufficient motivation, the author did not utilize the state-of-the-art in healthcare resilience analysis and used a simplistic approach in their assessment. For example, it is assumed that damage to the building is a proxy for bed availability and resilience,

which is not true. Many recent studies describe healthcare resilience as a function of physical infrastructure, staff, and stuff. So, it appears that the authors are going back in time to use a definition that is no longer recognized in the research community.

We thank the reviewer for this important point and the opportunity to clarify the scope of our modeling framework. We fully recognize that healthcare resilience is multidimensional, and that seminal work—including that of Dr. Mitrani-Reiser—has expanded the definition of hospital functionality to include structure, staff, and stuff. In fact, we referenced this work in the first version of the paper [1].

As noted in our literature review, these innovative models have been developed for analyses on single hospital buildings, where blueprints, space, and staff information are available to construct a high-fidelity finite element model of the structure and analyze how different spaces of the structure lose functionality. We have actually extended these high-fidelity modeling techniques to analyze single hospitals in Chile using discrete event simulations [2]. These models can often be very computationally expensive and not able to handle uncertainty exploration.

While these models are powerful at the single-building scale, they do not work at the regional level because of the lack of data and the enormous demands for computational power. We are modeling 76 hospitals with +400 buildings (not just one) in the Bay Area. For these types of problems, we must change the approach as we are not interested in capturing how functionality changes in each specific room of a building but instead in the relative resilience of one hospital campus versus another. We created these regional models for hospitals a few years ago together with Dr. Mitrani (who created the models you are referring to) to analyze effective plans for hospital system response to earthquake emergencies in the city of Lima, Peru [3]. In the field of regional risk modeling, we create a model based on the available vulnerability information. In the Bay Area, we created a regional risk model based on the available information on structural type, building height, year of construction, structural performance category, and non-structural performance category. This is the state of the art in regional risk modeling as we are interested at loss of functionality at the whole hospital campus level and not at each hospital room. We acknowledge, though that these models could be improved if high-resolution information for all hospitals would become available, which has not happened, unfortunately.

[1] Jacques CC, McIntosh J, Giovinazzi S, Kirsch TD, Wilson T, Mitrani-Reiser J. Resilience of the Canterbury Hospital System to the 2011 Christchurch Earthquake. *Earthquake Spectra*. 2014;30(1):533-554. doi:10.1193/032013EQS074M

[2] Merino Y, Ceferino L, Pizarro S, de la Llera JC. Modeling hospital resources based on global epidemiology after earthquake-related disasters. *Earthquake Spectra*. 2024;40(4):2402-2429. doi:10.1177/87552930241262788

[3] Ceferino, L., Mitrani-Reiser, J., Kiremidjian, A. et al. Effective plans for hospital system response to earthquake emergencies. *Nat Commun* 11, 4325 (2020).
<https://doi.org/10.1038/s41467-020-18072-w>

We rewrote this part in the introduction to make this point clearer (line 50):

Recognizing these vulnerabilities, this paper investigates the effects of future earthquakes on healthcare access in the entire San Francisco Bay Area, California. Previous research has largely focused on individual hospital vulnerability, using methods such as structural engineering analyses, performance-based assessments, and fault-tree evaluations to examine structural and non-structural damages within isolated facilities.[20–23] Other studies evaluated patient treatment dynamics and medical resource demands through flow models and discrete event simulations in post-earthquake scenarios but are similarly limited to single-building analyses.[24–27] Yet, hospitals do not function in isolation—particularly in dense urban areas—as demonstrated by seismic events in Chile, New Zealand, and Japan.[21, 28–31] Failures in individual hospitals can trigger broader network impacts, overwhelming remaining facilities and forcing patients to travel greater distances for care, severely exacerbating health outcomes for critically injured individuals. Thus, understanding healthcare resilience requires examining interconnected failures across healthcare and transportation infrastructure.[32–34]

Comment 3: Moreover, the paper suffers from many technical issues/mistakes, which causes the analysis to be unsound. For example, their predictions show that many hospitals will experience violent shaking at a scale not seen in over a century under a peak ground acceleration of 0.2. This is not possible. A peak ground acceleration of 0.2 is extremely low and is expected to cause only slight damage. The literature is overwhelmed with studies that show how a 0.2 PGA typically causes slight damage to different archetypal structures.

Comment 3: We appreciate the reviewer’s attention to this point and the opportunity to clarify. It was unclear whether the reviewer’s concern is that 0.2g is not a possible level of ground shaking, or that 0.2g does not constitute unusually high shaking for hospitals in the Bay Area. To clarify, we reference 0.2g only as an illustrative example—to note that many hospitals in the region have not experienced even this moderate level of shaking since the 1906 earthquake.

Importantly, our risk analysis does not rely on this deterministic value. As described in the Methods section, our analysis is based on probabilistic seismic hazard models, which sample the full distribution of ground motion intensities across a wide range of scenarios. For damage and accessibility assessments, we use spectral acceleration values at building-specific periods rather than PGA.

To avoid confusion, we have revised the Methods text to clarify that 0.2g was mentioned purely as a historical reference point, and that our modeling approach is fully probabilistic and consistent with established regional risk frameworks (line 469):

We study an M 7.25 earthquake scenario on the Hayward Fault. Similar scenarios have been extensively studied to inform resilience policy-making in the Bay Area.⁴⁷ The rupture geometry was obtained from the Uniform California Earthquake Rupture Forecast (UCERF) 2.7² The earthquake ruptures the Hayward South and North sections over a total length of ~110 km. Estimates of shallow shear wave velocities (averages at the top 30 m of soil) are utilized over the entire Bay Area.⁷³ With this information, we built the joint probability distribution of shaking intensities $\pi(i)$ in R2D. We utilized a state-of-the-art ground motion model⁷⁴ for shallow crustal earthquakes to estimate medians and logarithm standard deviations of i . i 's uncertainty is divided into two components.⁷⁵ The first component captures between-event uncertainty and affects the entire region equally, but it varies per intensity measure type, e.g., peak ground acceleration versus spectral acceleration. The first component captures correlation across different intensity measures.⁷⁶ The second component captures within-event uncertainty and affects the entire region and intensity measure types differently. This component captures spatial correlation and correlation across different intensity measure types. We use a computationally efficient method to account for the second component.^[62, 77] As stated earlier, we sampled 5,000 realizations of i and show expected values $E(i)$ for the entire region (Figure 1).

We have also clarified the manuscript to better explain that a PGA of 0.2g is not only plausible, but expected for hospitals located near active faults. For magnitude 7.0 earthquakes at short fault distances—as is the case for many sites along the Hayward Fault—PGAs well above 0.2g are consistent with empirical ground motion prediction equations. As illustrated in Figure 1, several hospitals in our study area are within 5 km of the Hayward Fault, making higher levels of ground shaking entirely realistic.

Historical earthquake records further support this point. While the Bay Area has experienced only moderate earthquakes in recent decades, most of the 76 hospitals in our dataset have not been exposed to shaking above 0.2g since the 1906 event. This is the motivation for referencing 0.2g in the text: even moderate levels of shaking would represent a significant increase over the historical experience of many healthcare facilities in the region.

To prevent further confusion, we revised the manuscript to clarify that the 0.2g value is used as a historical reference point, not as a basis for damage estimation. We also emphasize that our damage predictions are derived from spectral acceleration values at building-specific periods, and that the analysis accounts for the close proximity of hospitals to the Hayward Fault and the corresponding hazard expected from a magnitude 7.0 event (line 149).

We modeled the rupture geometry (see Methods) and found that 10 acute care hospitals

with 2167 beds (13% of the Bay's total) are within just 5 km of the projected rupture. Our shaking intensity estimates indicate that many hospitals could experience levels of ground motion not observed in recent earthquakes (See Methods). On average, we predict 51 hospitals will experience peak ground accelerations above 0.2g (Figure 2). By comparison, the 1989 M 6.9 Loma Prieta Earthquake—previously the strongest since the devastating 1906 M 7.9 San Francisco Earthquake—subjected only 14 hospitals to similar shaking intensities (Figure 2). Supplementary Figure 2 further contextualizes these findings, illustrating how recent events such as the 2014 Napa Earthquake (M 6.0) exposed just three hospitals to significant shaking (>0.2g). In contrast, the historic 1906 earthquake exposed nearly the entire portfolio (72 out of 76 hospitals)

Comment 4: In addition, the authors noted that many hospitals will have a probability of damage above 0.25. What is the significance of 0.25 as a damage probability? and What type of damage corresponds to 0.25 or more? Damage probability ranges from slight to moderate to extensive to complete, and each plays a role in quantifying healthcare functionality. To list that the damage probability is 0.25 has no structural engineering relevance. In earthquake engineering, there is no relevance in saying that many hospitals had a damage probability of more than 0.25. The reviewer recommends that the authors look at various FEMA documents on seismic performance assessment to become more familiar with damage assessment approaches, including the various engineering demand parameters used for seismic evaluation of buildings and hospitals.

Comment 4: We thank the reviewer for this comment and for recommending the FEMA guidance documents. These are indeed excellent resources, and we are well acquainted with them. The lead author has used FEMA P-58, FEMA P-695, and HAZUS extensively for both research and teaching, including in the graduate course “Disaster Risk Analysis” at UC Berkeley. One co-author is a risk specialist with years of experience applying these frameworks to develop regional risk models and policy interventions in collaboration with the World Bank. Another co-author has contributed to the development of NSF-funded tools at the NHERI SimCenter that operationalize these frameworks for both regional and single-building risk assessments. We have referenced these methodologies throughout our work and remain committed to aligning with best practices in the field.

Regarding the reviewer's specific point: the mention of a damage probability threshold of 0.25 was intended solely as an illustrative example—to demonstrate how the likelihood of damage increases with proximity to the fault. It was not used as a cutoff or performance threshold in the actual analysis. As detailed in the Methods section, we compute full probability distributions of multiple damage states for each hospital building using fragility functions and combine these with a fault tree to assess hospital functionality.

We understand that our use of the 0.25 value may have caused confusion, and we have revised the manuscript to make it clear that this number was only used for communication purposes, not as a substitute for engineering demand parameters or damage state classifications. Our

underlying modeling framework is aligned with established seismic performance assessment approaches and incorporates state-based damage probabilities in a probabilistic risk analysis. To improve clarity, we revised the Methods section to make this explicit. Below, we provide the revised text in plain format, noting that the full mathematical expressions and formatting appear properly in the main manuscript (line 410):

Hospitals can rapidly lose functionality at early stages of structural and non-structural damage, as observed in the 2023 M 7.8 Kahramanmaraş Earthquake in Turkey, the 2011 M 6.1 Christchurch Earthquake in New Zealand, and the 2010 M 8.8 Maule Earthquake in Chile. We consider that damage thresholds, d_s and d_n , for the structural and non-structural components, respectively, trigger the disruption of hospital functionality—that is, if either fails, the hospital loses functionality.

We estimate the probabilities of not exceeding these thresholds as:

- $p_k^s = \pi(D_k^s \leq d_s \mid I_k^s = i_k^s)$, and
- $p_k^n = \pi(D_k^n \leq d_n \mid I_k^n = i_k^n)$,

which can be evaluated using earthquake fragility functions. We then model functionality as the intersection of these two events, meaning the hospital component remains functional only if both structural and non-structural elements perform adequately after the earthquake. Since D_k^s and D_k^n are assumed conditionally independent, F_k (hospital building functionality) becomes a Bernoulli random variable with probability $p_k^s \times p_k^n$. Accordingly, the joint probability of functionality given the data d is:

$$\pi(f \mid d) = \prod_{k=1}^m [(p_k^s \times p_k^n)^{f_k} \times (1 - p_k^s \times p_k^n)^{(1 - f_k)}] \quad (\text{Equation 4})$$

The functionality of each hospital depends on the structural and non-structural components, as described in Equation 4. The hospital portfolio includes various levels of SPC (Structural Performance Category) and NPC (Non-structural Performance Category), which influence the final functionality outcome.

We also estimate the probability distribution of the total number of functional beds B_t across the region, computed as:

$$B_t = \sum \beta_k \times F_k \quad (\text{Equation 5})$$

where β_k is the number of beds at hospital building k (assumed fully functional before the earthquake).

To account for hospitals with multiple buildings, we estimate the distribution of the total number of functional beds B_h for hospital h , where \mathcal{L}_h is the set of buildings belonging to hospital h :

$$B_h = \sum \beta_k \times F_k \quad (\text{Equation 6})$$

In addition, we believe that the previous version of the paper may have overemphasized the 0.25 value, which may have led some readers to interpret it as a threshold used in the computation of hospital functionality. To avoid this misunderstanding, we have removed this reference from the main text.

Comment 5: Concerning this comment, the authors note that their analysis predicts that 8,501 beds will be lost throughout the Bay, and such predictions are based on slight damage (justified by previous studies that used slight damage to indicate loss of beds). Again, the literature is way past the idea of using damage to indicate bed losses and that stuff, staff, and physical systems should be used to indicate bed losses. In addition, many studies show that hospitals continue to operate beyond slight damage, including the studies referenced by the authors. None of the studies cited by the authors show that hospitals will lose functionality at slight damage. The StEER report cited by the authors indicates that only fully collapsed or severely damaged hospitals were evacuated after the Turkey EQ event. Unfortunately, the authors used this assumption to draw the study's main conclusions. Using slight damage drastically skews the results towards showing significant bed losses, which is very unrealistic. Also, the authors note that services within hospital buildings will be highly disrupted – a statement with no basis since the authors did not model any hospital units or services.

Response 5:

We refer back to our response to Comment 1, where we explain that we do not apply the “structure, staff, and stuff” framework in this study because, although valuable, it is not scalable to the regional level given current data limitations. While we acknowledge the broader dimensions of healthcare resilience, our study focuses on structural and non-structural damage as proxies for loss of hospital functionality—an approach consistent with the current state of the art in regional seismic risk modeling.

The lead author has participated in post-earthquake field investigations in Türkiye, Ecuador, and New Zealand and has directly observed instances where hospitals experienced disruption of services even at slight damage levels. In fact, the lead author coordinated the hospital section of the StEER report on the 2023 Türkiye Earthquake. Contrary to the reviewer’s comment, that report does not state that “only fully collapsed or severely damaged hospitals were evacuated.” Our fieldwork, including interviews with hospital staff, documented several cases where hospitals with slight or moderate damage lost functionality or required partial shutdowns due to non-structural failures, dislodged equipment, or internal hazards. These findings are described in our peer-reviewed article [1]. We made the following notes on the paper and added references to these cases (line 172):

Our baseline predictions assume hospitals lose functionality when structural or non-structural damage exceeds a threshold of slight damage—conditions often shown to disrupt hospital services (see Methods).[28-31]

[1] Ceferino, L., Merino, Y., Pizarro, S. et al. Placing engineering in the earthquake response and the survival chain. *Nat Commun* 15, 4298 (2024). <https://doi.org/10.1038/s41467-024-48624-3>

[28] Ardagh, M. W. et al. The initial health-system response to the earthquake in Christchurch, New Zealand, in February, 2011. *The Lancet* 379, 2109–2115 (2012). URL [http://dx.doi.org/10.1016/S0140-6736\(12\)60313-4](http://dx.doi.org/10.1016/S0140-6736(12)60313-4). Publisher: Elsevier Ltd ISBN: 1474-547X (Electronic)\r0140-6736 (Linking).

[29] Kirsch, T. D. et al. Impact on hospital functions following the 2010 Chilean earthquake. *Disaster medicine and public health preparedness* 4, 122–128 (2010). ArXiv: 1011.1669v3 ISBN: 1938-744X (Electronic)\r1935-7893 (Linking).

[30] Shimoto, M. et al. Hospital Evacuation Implications After the 2016 Kumamoto Earthquake. *Disaster Medicine and Public Health Preparedness* 16, 2680–2682 (2022). URL https://www.cambridge.org/core/product/identifier/S1935789322000258/type/journal_article.

[31] Achour, N. & Miyajima, M. Post-earthquake hospital functionality evaluation: The case of Kumamoto Earthquake 2016. *Earthquake Spectra* 36,

We do recognize that there is uncertainty in defining the precise threshold of functionality loss, especially in crisis settings. In some extreme situations, hospital staff were able to creatively adapt by cleaning up debris and isolating safe areas to continue providing care despite building damage. These heroic efforts, while admirable, also underscore the risks involved for both staff and patients.

To address this uncertainty and better reflect the range of plausible responses, we have added a sensitivity analysis that explores alternative (higher) thresholds for functionality loss. We label these scenarios “favorable” and “idealistic,” as we believe they represent optimistic assumptions that may not hold in a context like the Bay Area, where liability, regulatory, and safety concerns may prevent continued operation under damaged conditions (line 169).

Our risk analysis predicts a substantial loss of hospital capacity throughout the Bay Area following the earthquake scenario (see Methods). We estimate a loss of 8,165 hospital beds, resulting in only 51% of total acute care beds remaining functional (standard deviation: 21%). This significant reduction in capacity is highly uneven across the region (Figure 3, Table 1). Our baseline predictions assume hospitals lose functionality when structural or non-structural damage exceeds a threshold of slight damage (see Methods). These damage conditions are often shown to disrupt hospital services (see descriptions of multiple damage levels in Supplementary Tables 3 and 4).[21, 28–31] We also analyzed more optimistic scenarios with moderate (“favorable”) and extensive (“idealistic”) damage thresholds, resulting in increased overall functionality of 79% and 93%, respectively (Figure 3 and Supplementary Table 5). However, empirical

evidence supports our baseline assumption, as hospitals often lose operations at early stages of damage. [21, 28–31]

Figure 3. Post-earthquake functionality of hospitals and bridges across the Bay Area. [...] Probability distribution of functional acute care beds under baseline (slight), favorable (moderate), and idealistic (extensive) damage thresholds.

Supplementary Table 5. Predicted post-earthquake hospital bed capacities across Bay Area counties under thresholds of moderate (favorable) and extensive (idealistic) damage scenarios. The numbers in parentheses indicate the percentage in proportion to the pre-earthquake capacity.

	Favorable		Idealistic	
County	Mean	Std. Dev.	Mean	Std. Dev.
Alameda	1,743 (51%)	748 (24%)	2,635 (82%)	554 (17%)
Contra Costa	1,372 (75%)	378 (22%)	1,649 (94%)	188 (11%)
Marin	482 (71%)	159 (25%)	581 (93%)	84 (13%)

Napa	319 (89%)	50 (14%)	345 (98%)	19 (5%)
San Francisco	3,258 (89%)	502 (14%)	3,547 (98%)	173 (5%)
San Mateo	1,269 (86%)	229 (16%)	1,390 (96%)	120 (8%)
Santa Clara	3,512 (80%)	725 (17%)	4,068 (94%)	346 (8%)
Solano	636 (88%)	116 (16%)	705 (98%)	50 (7%)
Sonoma	537 (90%)	64 (11%)	573 (98%)	26 (4%)
Total	13,129 (79%)	2,583 (16%)	15,492 (93%)	1,271 (8%)

We also appreciate that the reviewer is familiar with the “structure, staff, and stuff” model. While it brings valuable insight, we note that this framework still does not model human behavior, such as staff decision-making or willingness to work in damaged conditions—factors that play a critical role in hospital functionality. To date, these behaviors remain challenging to represent accurately in either building-level or regional modeling frameworks. We believe that more research is needed in this area to enable their integration into future probabilistic risk models.

Finally, we note that loss of “stuff” (e.g., medical equipment and supplies) is often directly linked to damage to non-structural components or structural failures in specific hospital zones. Even in high-fidelity single-building models, damage remains the primary driver of functionality loss, as confirmed by our field deployments and multiple case studies.

Comment 6: Another significant issue is that bridge damage and failure analysis are not included in determining travel time. The travel time calculated in the paper can not be correct due to the lack of bridge failure analysis.

Response 6: After reading your reviews (along with those from the other reviewers), we invested significant effort into improving our modeling of transportation infrastructure disruptions in the Bay Area. This became a major component of our revision and is the main reason for the time it took us to respond.

In this revised version, we integrated detailed information for over 5,000 bridges in the Bay Area and significantly expanded the transportation network model, which now includes approximately 1.5 million edges and 0.5 million nodes. This refinement substantially increased computational demands: the 5,000 Monte Carlo simulations required for the analysis took about two weeks to run on a laptop equipped with an Apple M2 chip and 64 cores. We ran multiple iterations to ensure the model was fully debugged and stable.

Thanks to this updated formulation—and your review—we are now able to capture how hospital and transportation failures interact to influence post-earthquake hospital accessibility. Our revised analysis highlights several key cascading effects:

- Widespread transportation failures lead to compound access losses, with travel times increasing by over a factor of 10 in some areas.
- Some hospitals in the Bay Area may become completely isolated due to bridge failures.
- Urban communities—particularly in Fremont and Novato—may lose access to essential services due to localized bridge disruptions.

We include some of the updated results and manuscript revisions below.

Abstract (line 15):

By integrating seismic hazard with hospital and transportation infrastructure’s vulnerability and connectivity data, we analyze 76 hospitals (426 buildings with 16,639 beds) and 5,163 bridges within a vast network of ~1.5 million edges and ~0.5 million nodes [...] Widespread transportation failures further restrict access, increasing regional travel times by 177% and exceeding 1000% in parts of East Bay, potentially fully isolating hospitals and entire urban communities. These findings underscore the urgent need for resilient healthcare and transportation infrastructure to mitigate life-threatening disruptions following major earthquakes.

Introduction (line 80):

To fill this critical gap, we investigate, for the first time, joint failures of healthcare and transportation networks across the San Francisco Bay Area after a major earthquake. By examining diverse urban communities near active seismic faults—often underrepresented in previous studies—this research provides valuable insights into post-earthquake healthcare

access, particularly relevant for earthquake-prone regions worldwide with diverse populations and similarly vulnerable hospitals and bridges, such as those in the U.S., Japan, and Türkiye.

To our knowledge, this is the first study to evaluate post-earthquake healthcare access at this scale using publicly available data on both hospital and transportation infrastructure. Whereas prior work has focused on individual facilities[20-23] or small systems[42-46]—often relying on synthetic data—our approach couples regional seismic risk models with large-scale network analysis to capture interdependent failures across an entire metropolitan area. By integrating vulnerability data from 426 hospital buildings and 5,163 bridges, and simulating disruptions across a transportation network with approximately 1.5 million edges and 0.5 million nodes, we offer one of the most extensive assessments to date of how earthquakes affect healthcare access. This study advances the state of the art in regional risk modeling globally, offering a scalable, computationally intensive framework that supports risk-informed decision-making in other high-risk urban areas.

New Section on Post-earthquake Bridge Functionality (line 196):

Our risk analysis model predicts that 1,469 bridges out of 5,163 would be damaged due to the earthquake scenario (see Methods). Consequently, 3,693 bridges (72%, standard deviation: 17%) are expected to remain undamaged and fully operational (Figure 3). Some damaged bridges could continue operating, albeit at reduced capacities. Under our “baseline scenario”, where only bridges with at most slight damage remain operational, a total of 3,964 bridges (77%, standard deviation: 15%) would function. In a more optimistic “favorable scenario”, where bridges with at most moderate damage also remain partially operational, 4,069 bridges (79%, standard deviation: 14%) could provide service. In an idealistic scenario, which assumes that bridges even with extensive damage retain emergency functionality, 4,527 bridges (88%, standard deviation: 11%) would remain operational.

Similar to hospitals, Alameda County faces the most severe transportation impacts (Figure 3, Table 2). Under the baseline scenario, Alameda retains only 44% functional bridges (from 642 down to 282). Marin County is second most impacted, retaining 64% (from 195 to 125 bridges). Additionally, estimates of reductions in bridge travel speeds (see Methods)—calculated from their damage probabilities—highlight Alameda’s higher vulnerability due to proximity to the Hayward Fault rupture (Figure 3, Table 2).

Editorial Note: The map data in figure 3 in this Peer Review file is from © OpenStreetMap contributors openstreetmap.org/copyright.

Figure 3. [...] Lower left: Probability distribution of fully functional bridges under baseline (slight), favorable (moderate), and idealistic (extensive) damage thresholds. Lower right: Spatial distribution highlighting bridges with post-earthquake travel capacities below 75%. Results generated from 5,000 Monte Carlo simulations.

Revised findings on Accessibility (line 223):

At a more detailed neighborhood scale, these disruptions are even more pronounced (Figure 4). We identified six densely populated zip codes (each exceeding 15,000 residents, labeled #1 to #6 in Figure 4) facing substantial reductions in healthcare access. The most severely impacted is Novato (#1, Marin County), where travel times increase nearly 25-fold, from 7.3 to 185.5 minutes, affecting approximately 18,000 residents. Similarly, a Fremont zip code (#6, Alameda County) with roughly 66,000 residents experiences an eightfold increase (from 8.4 to 42.2 minutes). Other significantly impacted zip codes in Richmond (#4), Oakland (#2 and #5, Alameda County), and San Jose (#3, Santa Clara County) face travel time increases of 10, 11, 9, and 11 times their pre-earthquake levels, respectively. These results underscore the dramatic reshaping of healthcare access throughout East Bay communities along the earthquake rupture.

Finally, we explored the relative contributions of hospital and transportation infrastructure disruptions through scenario analyses (Figure 4). We compared the baseline (None, no infrastructure disruptions) against scenarios modeling probabilistic damage only to transportation infrastructure (Transp., hospitals fully resilient), only to hospital infrastructure (Hosp., transportation fully resilient), and simultaneous probabilistic disruptions to both (Transp. & Hosp.). When considering disruptions only to transportation infrastructure (Transp.), average Bay Area travel times increase by 41% (from 6.1 to 8.6 minutes), notably less than the combined scenario (Transp. & Hosp., 177% increase). Conversely, modeling only hospital infrastructure disruptions (Hosp.) yields a 79% increase (from 6.1 to 10.9 minutes), highlighting greater relative fragility in hospital networks.

Editorial Note: The map data in figure 4 in this Peer Review file is from © OpenStreetMap contributors openstreetmap.org/copyright.

At localized urban scales, the compounding effects of simultaneous transportation and hospital infrastructure failures lead to disproportionately severe impacts (Figure 4). For example, as mentioned earlier, in the most severely impacted zip code (Novato, labeled #1 in Figure 4), travel times dramatically increase from 7.3 to 185.5 minutes under combined disruptions (Transp. & Hosp.), compared to only 14.1 minutes when transportation disruptions are excluded (Hosp. only). Similarly, for other highly affected zip codes (labeled #2 to #6 in Figure 4), travel-time ratios notably decrease—from 11, 11, 10, 9, and 8 under combined disruptions, to 6.4, 3.1, 6.1, 4.7, and 3.4 when transportation disruptions are not considered—highlighting the critical influence of compounding infrastructure failures on healthcare access.

Figure 4: Access to functional acute care hospitals in the Bay Area after the earthquake. Upper left: Post-earthquake (Post-EQ) increases in travel time at the zip-code level compared to pre-earthquake (Pre-EQ) conditions, highlighting densely populated areas that are most severely impacted. Upper right: Ratios of post- to pre-earthquake travel times to the closest acute care hospitals. Lower right: Distribution of travel times per zip code under four scenarios: None (no infrastructure disruptions), Transp. (only transportation infrastructure is vulnerable), Hosp. (only hospital infrastructure is vulnerable), and Transp. & Hosp. (both transportation and hospital infrastructures are vulnerable). Lower left: Ratios of post- to pre-earthquake travel times when only hospital infrastructure (Hosp.) is vulnerable. Results generated from 5,000 Monte Carlo simulations.

New section on Isolated Communities and Hospitals (line 252):

To understand the drivers behind the sharp increases in travel times observed earlier, we analyzed travel volumes to the closest acute care hospitals before and after the earthquake. Figure 5 shows pre-earthquake road usage, with line thickness representing travel volume. Roads in brown highlight those whose usage falls above the 90th percentile based on pre-earthquake conditions, emphasizing the most critical travel routes. As expected, the highest-usage roads prior to the earthquake were located near hospitals, especially in densely populated areas with fewer hospitals. For instance, communities in Fremont (#6 in Figures 4 and 5) are served by only two hospitals, while Richmond (#4) relies on just one.

Figure 5 also displays post-earthquake road usage, using the same pre-earthquake 90th percentile threshold to highlight heavily traveled roads. After the earthquake, patient travel patterns shift significantly, with many communities forced to travel longer distances along major corridors. For example, residents in Oakland (#2 and #5 in Figures 4 and 5), who previously relied on nearby hospitals, will now have to travel west to the San Francisco Peninsula or east to Walnut Creek due to simultaneous disruptions in both hospital and transportation infrastructure. These shifts help explain the steep increases in travel times observed in East Bay communities along the rupture zone.

Four major bridges connect the East and West Bay: the Richmond–San Rafael, Oakland–San Francisco Bay, San Mateo–Hayward, and Dumbarton bridges. Before the earthquake, no patients needed to cross these bridges to reach their nearest acute care hospital (Figure 5). After the earthquake, however, these bridges become critical, with travel volumes increasing to 3.6, 9.3, 6.3, and 13.6 times the pre-earthquake 90th percentile values—reflecting the large-scale redistribution of healthcare demand.

Our analysis also revealed a more complex post-disaster phenomenon: micro-scale isolation of hospitals and communities due to localized bridge failures. For example, the hospital serving the most severely impacted zip code in Novato (#1 in Figures 4 and 5), which has 64 beds, depends entirely on a single bridge for access. If this bridge fails, the hospital becomes inaccessible, increasing travel times from 7.3 to 185.5 minutes for the surrounding population.

Editorial Note: The map data in figure 5 of this Peer Review file is from © OpenStreetMap contributors openstreetmap.org/copyright.

More concerning, we found that urban neighborhoods near Fremont (#6) could become fully isolated (Figure 5). In one area, failure of three bridges would trap one part of the neighborhood; in another, a different set of three bridge failures would isolate the remaining part. These communities would lose access not only to hospitals but also to other essential services such as grocery stores and pharmacies—illustrating how cascading infrastructure failures can sever lifeline access in disaster scenarios.

Figure 5: Mobility to reach the closest functional acute care hospital in the San Francisco Bay Area. Upper left: Pre-earthquake travel volumes to the closest acute care hospitals. Numbers 1 to 6 indicate the hospitals serving the dense zip codes with the highest post-earthquake increases in travel time. Upper right: Projected travel volumes after the earthquake, showing major shifts in patient mobility. The red-shaded corridor illustrates how Oakland residents (#2 and #5) must travel west to the Peninsula or east to Walnut Creek due to hospital and transportation disruptions. Lower left: The hospital serving zip code #1 (Novato) relies entirely

on a single bridge; if the bridge fails, all access to its 64 beds is lost. Lower right: Community near zip code #6 (Fremont), where the southern and northern sections become fully isolated if three different bridges fail in each area. Results are based on 5,000 Monte Carlo simulations of post-earthquake patient mobility.

Comment 7: The methods section also suffers from statements that are either very simplistic and outdated by more detailed analysis, incorrect, or indicate problems with how the study was conducted. For example, the authors suggested that building functionality is modeled with Bernoulli distribution. The state of the art on building functionality (plz see work by Henry Burton or Naiyu Wang) has shown building functionality to be much more complex than this, requiring more details analysis and the inclusion of various probabilities, including the availability of crew workers, the time it takes to issue a permit, material availability, and many other things. These parameters will impact the hospitals' functionality and the staff's viability since their homes could be damaged - criteria that were not included in this study. Assuming a simple Bernoulli function has no basis.

Response 7: We thank the reviewer for the opportunity to clarify our modeling approach and the role of the Bernoulli distribution in our framework. We fully agree that the evolution of hospital functionality after an earthquake depends on a wide range of factors beyond structural integrity, including crew availability, permitting timelines, material logistics, and staff viability. These considerations are especially relevant for modeling long-term recovery trajectories, where the full restoration of services can take multiple years. For example, following major earthquakes such as the 2010 Chile Earthquake and the 2023 Türkiye Earthquake, the repair of residential and healthcare buildings often took 1 to 4 years, depending on the severity of damage, logistical constraints, and institutional capacity.

In contrast, our study focuses on a regional, scenario-based assessment of hospital system functionality and accessibility during the first days to months following a major earthquake. Within this timeframe, building functionality is estimated based on damage to structural and non-structural components, which serve as proxies for immediate operability. This approach is consistent with established practices in regional seismic risk modeling and is appropriate for identifying where healthcare access is likely to be most compromised in the aftermath of a major event.

To support this analysis, we model hospital building functionality using correlated Bernoulli random variables, not independent ones. These variables are derived from fragility-based

exceedance probabilities informed by spatially correlated ground motions and component damage. Simulating from high-dimensional correlated Bernoulli distributions is computationally intensive—especially at the scale of this study, which includes 76 hospital campuses and over 5,000 bridges—and requires careful approximations to ensure tractability. We have clarified these aspects in the revised Methods section (line 421).

This formulation reflects our focus on estimating hospital functionality during the initial post-earthquake period, where service disruption is most directly tied to structural and non-structural damage. Rather than modeling the full trajectory of hospital recovery\cite{Burton2016}—which depends on permitting, logistics, and staffing—we assess whether each building is likely to remain operational immediately after the event. To enable this analysis across hundreds of facilities and correlated scenarios, we use Bernoulli random variables derived from fragility-based exceedance probabilities, accounting for spatial dependencies in seismic intensity and building response (Figure 7).

We are also familiar with the recovery modeling frameworks developed by Henry Burton and Naiyu Wang. In fact, the lead author has previously collaborated with Henry Burton [1], using the same structural modeling techniques that underpin Burton's recovery framework [2]. While these frameworks provide valuable insights into building-level recovery and time-dependent functionality, they also rely on limit-state exceedance representations and fault-tree logic—concepts that closely align with our modeling structure. For example, Burton's framework uses three discrete functionality states (none, partial, full), whereas we adopt a binary representation that is appropriate for our study's scope, scale, and data constraints. Moreover, many of Burton's recovery paths transition directly from no functionality to full functionality, which is conceptually similar to the transitions used in our study.

[1] Burton, Deierlein, Lallemand, and Lin. "Framework for incorporating probabilistic building performance in the assessment of community seismic resilience." *Journal of Structural Engineering* 142.8 (2016): C4015007.

[2] Lallemand, Burton, Ceferino, Zullock, and Kiremidjian. "A framework and case study for earthquake vulnerability assessment of incrementally expanding buildings." *Earthquake spectra* 33.4 (2017): 1369-1384.

We acknowledge that a binary Bernoulli model does not capture partial functionality within individual buildings, but our framework allows for partial functionality at the hospital campus level, since each campus includes multiple buildings evaluated with different random variables. This structure enables us to estimate the proportion of functional capacity at each hospital across simulated scenarios.

To further address the reviewer's concern, we have added a sensitivity analysis (see response to Comment 5 and Supplementary Note 4) that explores how varying functionality thresholds affect accessibility outcomes.

Comment 8: Another major issue in the methods section is that the authors tried to connect the data collected from the vulnerability rating to the HAZUS fragility. This connection is based on assumptions not validated or supported by previous studies. The authors tried to use ASCE-7 to come up with some numbers that reflect a lack of understanding of the code philosophy. I recommend utilizing fragility functions that reflect the building characteristics; otherwise, the calculated damage does not represent reality.

Response 8: Thank you very much for the helpful feedback. We would like to clarify how the fragility functions used in our study are linked to building characteristics and why we believe this approach is appropriate for a regional-scale hospital risk assessment.

The fragility functions applied in our model do reflect key building characteristics, including structural system type, effective building height, and the official Structural Performance Categories (SPC) and Non-structural Performance Categories (NPC) assigned to each hospital by the California Department of Health Care Access and Information (HCAI). These SPC and NPC ratings reflect engineering assessments of seismic vulnerability and are widely used by the State of California to inform seismic upgrade policies.

In line with the guidance provided in the FEMA HAZUS Technical Manual (Section 5.7: Guidance for Expert Users), buildings "deemed not to conform to modern code provisions should be assigned a lower seismic design level or defined as Pre-Code buildings if not seismically designed." Following this recommendation, we mapped SPC 1 buildings to pre-code fragilities, SPC 2–3 to moderate-code, and SPC 4–5 to high-code fragility curves. This mapping approach is consistent with prior studies and allows us to leverage the extensive fragility data provided in HAZUS while making transparent assumptions about code conformance.

We acknowledge that this method involves simplifications and introduces uncertainty. However, such approximations are common in regional seismic risk modeling, where detailed high-resolution building-by-building structural data are typically unavailable. To address this, we incorporate Monte Carlo simulations to propagate uncertainties in damage and functionality estimates across the building portfolio. In addition, we conducted a sensitivity analysis to explore how varying damage thresholds influence our hospital functionality outcomes (see response to comment 5).

We also want to clarify that we did not derive fragility functions from ASCE 7 provisions, but only referenced ASCE 7 to support our assumptions about differences in design intent across building code eras. The fragility functions themselves are based on HAZUS and were adjusted according to the SPC/NPC mapping logic.

To ensure transparency, we have added more detail to the Methods section (see "Hospital Vulnerability Modeling") and Supplementary Table 1 describing how SPC 2 buildings were classified based on official HCAI reassessments. We also added Supplementary Table 3, which includes definitions of the damage states for the three most common structural systems in our dataset—Steel Moment Frames, Concrete Shear Walls, and Steel Braced Frames—and Supplementary Table 4, which defines non-structural damage for elements like suspended ceilings and mechanical/electrical equipment, which often drive hospital service disruptions. We added the following (line 117):

In this paper, we used the Structural Performance Categories (SPC) and Non-structural Performance Categories (NPC), established and reported by California's Department of Health Care Access and Information (HCAI), to characterize structural and non-structural deficiencies.[44] Each hospital building receives vulnerability ratings (SPC or NPC) ranging from 1 to 5, from the most to the least vulnerable. Supplementary Tables 1 and 2 fully describe SPC and NPC.

Supplementary Table 3. *Definitions of structural damage states for most common structural types in the hospital building portfolio: S1: Steel Moment Frame; C2: Concrete Shear Walls; S2: Steel Braced Frame. These definitions come from HAZUS and were adopted in this study.⁴*

[Table Redacted]

Supplementary Table 4. Definitions of non-structural damage states for two acceleration-sensitive components whose failures often disrupt hospital services. *These definitions come from HAZUS and were adopted in this study.*⁴

[Table Redacted]

Supplementary Table 1. Structural Performance Categories (SPC) according to California’s Department of Health Care Access and Information³

[Table Redacted]

Comment 9: The authors also assumed that some hospitals could accept patients if only one bed was functional, which is a weak assumption. If only one bed is functional, then the hospital must be substantially damaged, and if most of the hospital is damaged, then it will be unsafe to utilize. Please check previous studies reporting that many hospitals evacuated, even when they had more than 50% of their capacity due to severe damage. I doubt whether any case study or expert opinion will support the assumption made by the authors.

Response 9: We fully agree that treating a hospital as operational with just one functional bed does not reflect realistic clinical or safety conditions. In response to this concern, we have updated our model to require that hospitals retain at least 50% of their functional bed capacity to be considered able to receive patients.

This change was motivated by both prior studies and field observations. For example, during post-earthquake reconnaissance following the 2023 Kahramanmaraş earthquakes in Türkiye, the lead

author visited several hospitals that were evacuated or rendered inoperable despite partial structural integrity. Closure decisions were often based on internal system damage, safety assessments, or operational concerns—not merely bed availability. Similar outcomes were reported after the 2010 Chile and 1994 Northridge earthquakes, where hospitals with remaining capacity still ceased operations due to broader functional limitations.

While our model does not explicitly simulate these decision-making factors, they highlight the importance of establishing more conservative functionality thresholds. This is why we adopted the 50% bed capacity rule and also explored sensitivity to different damage thresholds in our updated travel time analysis (see response to Comment 6). These revisions are reflected in the manuscript. Please note that the equation formatting here is adapted for Google Docs and may differ slightly from the typeset version in the manuscript. For the full mathematical expression and formal notation, we kindly refer the reviewer to the main manuscript (line 619).

In the post-earthquake scenario, all hospitals are assumed to be functional. Then, Equation 11 finds the shortest time to reach an acute care hospital p (see Figure 8).

In an earthquake scenario, we adjust Equation 11 to:

$$T = \min \{ t(h) \mid h \in \hat{h} \}^{**}$$

where $\hat{h} = \{ h \mid B_h > 0.5 \times \beta_h \text{ for all } h \in \hat{h} \}$, to account for the reduction in hospital capacity. In this case, \hat{h} is the set of hospitals that retain more than 50% of their pre-earthquake bed capacity (β_h). This means that, in our model, a hospital must have at least 50% of its beds functional ($B_h > 0.5 \times \beta_h$) in order to be considered available to receive new patients. Studies on hospital disaster resilience during earthquakes suggest that a significant reduction in functional bed availability can severely strain hospital operations, particularly in accommodating new patients in the emergency department.²⁷

97

Comment 10: Another major issue in this paper is that the authors ignored damage to the transportation network, which is a fundamental component when calculating the estimated demand at different hospitals. Damage to bridges and the road network will significantly change the shortest paths calculated in this study, which can lead to cases where some functional hospitals see less demand due to reduced accessibility. Many previous studies on healthcare systems reported that impact and some even included the dynamic effect of the change in travel time on hospital demand and accessibility.

Response 10: Thanks for the constructive comment. We believe we have address this on the response to comment 6

Comment 11: The authors also assumed the number of patients proportional to the number of populations. However, considering the variation in patient distribution, especially during disasters, is critical for accurate demand assessment, and again, many studies have already done that. Most earthquake casualties are related to building collapse, a function of the building characteristics and earthquake intensity measured at the building locations. So, the assumption made by the authors has no basis and will significantly impact the results and main conclusions of the study.

Response 11:

We thank the reviewer for raising this important point. We spent significant time considering whether to explicitly model injury distributions from earthquakes in the Bay Area as part of this study. While we have modeled spatial injury distributions in other regions with high expected casualties—such as Peru[1]—we ultimately decided not to include this component here for two main reasons.

First, our focus is on hospital accessibility in the days to a few months following a major earthquake, rather than the immediate emergency response phase. While surges of severely injured patients are critical in the early aftermath, they typically last only a few days—except in rare large-scale disasters such as the 2023 Türkiye Earthquake, where demand persisted for several weeks.¹ In the Bay Area, where building code compliance is higher, we expect a shorter emergency phase and lower injury loads. Our goal was to evaluate accessibility during the transitional period when hospitals may still be closed or operating below capacity, and when routine or follow-up care becomes increasingly important. To model this, we used population-weighted demand, assigning it to the network based on census tract centroids (Figure 8). We added an explanation to the manuscript (line 645):

Our goal was to quantify hospital accessibility losses beyond the immediate emergency period, recognizing that hospitals often take months or even years to be repaired. Because people are typically injured within damaged buildings, patient surges after earthquakes generally last only a few days—except in rare extreme cases such as the 2023 Türkiye Earthquake. To focus on long-term access rather than short-term emergency response, we did not model injury locations directly. Instead, we assumed that

patient demand is proportional to population distribution and modeled accessibility accordingly. For the Bay Area, we based our analysis on population-weighted access loss and identified the network's vertices closest to the centroids of census tracts to define the set p (Figure 8).

Second, the state of the art in spatially explicit casualty modeling remains limited—especially in California, where recent damaging earthquakes have not produced large empirical injury datasets. To address this gap, we are currently leading an NSF-funded project (NSF Award #2410291) to develop high-resolution, empirically grounded casualty models for the Bay Area. This includes ongoing fieldwork, structural modeling, and healthcare systems simulation. We plan to incorporate the reviewer's suggestion as part of that forthcoming work.

[1] Ceferino, L., Merino, Y., Pizarro, S. et al. Placing engineering in the earthquake response and the survival chain. *Nat Commun* 15, 4298 (2024). <https://doi.org/10.1038/s41467-024-48624-3>

Comment 12: Overall, the paper addresses a problem investigated in the past using simple approaches that are not necessarily detailed and, in some cases, infused with incorrect assumptions and/or calculations. Unfortunately, on these grounds, this paper can not be published.

Response 12: We appreciate the reviewer's feedback and the opportunity to clarify the novelty and rigor of our work. We respectfully disagree with the assessment that this study revisits prior work using simple or incorrect approaches. As outlined in our responses to earlier comments, we have taken care to develop a regionally scalable, computationally intensive model that integrates structural and non-structural fragility modeling, correlated damage simulations across more than 5,000 infrastructure elements, and accessibility calculations on disrupted transportation networks—including bridge failures.

While our approach is necessarily approximate in some respects—as is common in regional risk modeling—we have been careful to follow established methodologies (e.g., HAZUS-based fragility models, Monte Carlo simulation), and we have incorporated improvements in model fidelity compared to previous studies. For example, we use official SPC and NPC classifications for hospital buildings, spatially correlated damage fields, and we explicitly model threshold-based hospital capacity loss and rerouting under bridge disruptions. In response to earlier reviewer comments, we have also:

- Revised our hospital functionality threshold to 50% capacity, based on field observations and supporting literature.
- Clarified and expanded our modeling of hospital vulnerability, including mapping procedures and fragility assumptions.

- Added a sensitivity analysis to assess the effects of alternative damage thresholds.
- Emphasized that our focus is on post-emergency accessibility over the weeks and months following an earthquake, rather than short-term surge demand (e.g., couple of days).
- Analyzed the transportation infrastructure risks, capturing more realistic travel times and identifying a potentially isolated hospital and community.

We believe these revisions significantly strengthen the manuscript and address concerns regarding assumptions and scope. We have been transparent about our modeling choices and limitations, and have taken steps to validate them against prior studies and post-earthquake observations (e.g., fieldwork in Türkiye, Chile, and California).

We respectfully hope the reviewer will reconsider the value of this contribution, especially in light of these clarifications and improvements.

Comment 13: Reviewer #2 (Remarks on code availability):

I evaluated the authors' online sources, and they all have sufficient information.

Response 13:

Thank you.

Reviewer #3

Reviewer #3 (Remarks to the Author):

Comment 1: The manuscript presents a case study on the impacts of a scenario earthquake on the Hayward fault on the San Francisco Bay Area's hospital capacity. It is very interesting work and implemented using state of the art approaches, giving good weight to the discussion points provided.

Comment 2: We thank the reviewer for their thoughtful and encouraging feedback. We are grateful for the recognition of our work's relevance and the implementation of state-of-the-art approaches, and we are especially glad that the discussion points resonated.

We sincerely apologize for the delay in submitting our response. A major revision was undertaken to expand the analysis and address reviewer suggestions thoroughly, which required substantial time and computational effort.

First, we have made significant improvements to the manuscript's structure and presentation. We streamlined the introduction and results sections to improve clarity, reduced redundancy, and ensured a concise and technically focused tone. We also revised scientific estimates for consistency and precision and addressed editorial corrections to improve the overall readability and professionalism of the paper.

Most notably, we expanded the scope of the analysis to explicitly incorporate transportation infrastructure disruptions, including bridge failures. This addition allowed us to assess hospital accessibility in greater detail and revealed the potential for complete isolation of both a hospital and a community following a major Hayward Fault earthquake. We believe this significantly enhances both the rigor and real-world applicability of the study. We provide more detail on this major addition at the end of our response.

To help the reviewer navigate our response, we have marked our replies in **blue** and used **red** text in the manuscript to indicate the changes. We hope the reviewer finds the revised version even more impactful and technically complete, and we truly appreciate the supportive and constructive review.

Comment 2: The biggest issue I found with the work was the length of the text that made it rather difficult to read. I understand that the authors wish to discuss their results in detail, but I felt much of the discussion could have been delivered in half the amount of text. I would strongly urge the authors to revise the initial parts of the paper and trim unnecessary sentences or anecdotes that do not contribute to the main points being delivered.

Response 2: Thank you for this comment. We reduced the main text from ~5,000 words to 4,000 words. We wanted to reduce it further but the addition of the transportation network required us to add more text.

Comment 3: Furthermore, some of the text is written in a manner that seems less like a technical article, and more like a magazine article for non-technical readers. The excessive use of adjectives (i.e. violent shaking) and adverbs (i.e. we rigorously study) and terminology like “we predict” seems a little exaggerated, at least in this Reviewer’s opinion. Reducing these would help the article’s conciseness.

Response 3: We revised the manuscript to reduce the use of adjectives, adverbs, and informal expressions such as “violent shaking,” and “we rigorously study” and “we predict.” Several sentences were re-written to eliminate redundant phrasing. We avoided all the above mentioned adjectives and the current manuscript looks more concise.

Comment 4: Overall, it is good and interesting work that should be accepted following some minor adjustments.

Response 4: Thank you. We have reviewed the paper according to all these comments.

Some minor observation:

Comment 5: Abstract: Correct “In East Bay” to “In the East Bay”

Response 5: Change executed

Comment 5: Change “are way higher” to more technical terminology “are much higher”

Response 6: Change executed

Comment 6: Page 1, the term “mass-casualty earthquake” sounds a little dramatic for a technical publication. Consider revising.

Response 6: We changed it to “major”.

Comment 7: The authors note that Alfred E. Alquist Hospital Seismic Safety Act was enacted in 1973 but twelve Pre-Alquist Act hospital buildings received red tags following the 1994 Northridge earthquake. Does this mean that the act was focused on new hospitals and existing hospitals were left as is? Please comment.

Response 7: Yes, the 1973 Alquist Act applied only to new hospital construction. As described in the manuscript, it was the 1994 Northridge Earthquake that revealed the vulnerability of older

hospital buildings and prompted the passage of Senate Bill 1953, which introduced retrofit requirements for existing facilities. As shown in Figure 1, many hospitals have yet to complete these retrofits. The current deadline is 2030, but many are unlikely to meet it. We have revised the text to clarify this timeline and policy progression (line 34).

In the U.S., even moderate earthquakes have severely disrupted healthcare access. The 1971 M 6.6 San Fernando Earthquake severely damaged four major hospitals, resulting in forced evacuations, two structural collapses, and more than 40 fatalities.¹³ These tragedies prompted legislative action, notably the Alfred E. Alquist Hospital Seismic Safety Act of 1973, mandating stronger, more resilient hospital infrastructure. However, the law primarily applied to new hospital construction and did not require immediate retrofits of existing facilities built before 1973. The law emphasizes that hospitals must be capable of protecting vulnerable patients and providing critical medical services immediately after disasters, highlighting the importance of resilience planning in disaster-prone regions.¹³

In 1994, the M 6.7 Northridge Earthquake severely disrupted healthcare services,¹⁴ forcing evacuations at eight acute care hospitals in Los Angeles County and causing around USD 3 billion in losses.¹⁵ Twelve hospital buildings constructed before the Alquist Act were deemed unsafe for occupancy. Post-Alquist Act buildings experienced less structural damage, but non-structural damage was still extensive.¹⁶ These disruptions prompted California Senate Bill 1953 (passed in 1994), mandating significant retrofits to acute care hospitals by 2030.¹⁷ ¹⁸ Notably, many hospitals face financial barriers that may prevent them from meeting these critical resilience goals.¹⁹

Comment 8: Also, the authors write 12 hospitals, but a few lines earlier, 11 were mentioned. Check that these are correct.

Response 8: We corrected this (paragraph is in the previous response). It should have been twelve. Thank you for pointing this out.

Comment 9: Does the “California Senate Bill 1953” refer to the year 1953 or just the number 1953? I presume not, so it would be better to specify the year so as to avoid confusion in the timeline of these legislative developments.

Response 9: This is just the number of the Bill. It was enacted in 1994. We added a clarification:

These disruptions prompted California Senate Bill 1953 (passed in 1994), [...]

Results:

Comment 10 :It may be better to specify that “The Bay” refers to the “The Bay Area of San Francisco” for unfamiliar readers.

We iterated this in the entire manuscript. We refer to it as the San Francisco Bay Area throughout the paper.

Comment 11: Statements like “Our predictions show that many hospitals will experience violent shaking at a scale not seen in over a century.” seem a little dramatic and unnecessary. It also conveys a distorted view to the public that the earthquake shaking can be predicted. In general, words like estimate would be a better reflection of the science behind the methods.

Response 11: We agree. We changed it to (line 150):

Our shaking intensity estimates indicate that many hospitals could experience levels of ground motion not observed in recent earthquakes

Comment 12: Equation 1. It seems “l’m” is a typo.

Response 12: We have corrected the notation:

²⁰¹ quake consequence, e.g., economic losses or fatalities, as

$$P_{DV,DS,IM}(dv, ds, im) = P_{DV|DS}(dv|ds)P_{DS|IM}(ds|im)P_{IM}(im) \quad (1)$$

²⁰² where $P()$ is a probability distribution (or mass) function, and DV is a random variable

Comment 13: Would it not make more sense to define the shaking variable as $IM = im$ instead of $l = l$? It would maintain a consistency with the previous text.

Response 13: We thank the reviewer for the thoughtful suggestion. In regional risk modeling, especially when simulating thousands of assets and correlated realizations, the dimensionality of the random variable space becomes quite large. To manage this complexity and maintain notational tractability, we adopt a convention of using concise single-letter variable names, such as l for intensity measures. This approach facilitates the presentation of joint distributions, conditional probabilities, and integrals involving multiple variables across infrastructure elements. While we agree that using IM could align more explicitly with “intensity measure,” we believe that the definition of l is clearly stated and consistently applied throughout the manuscript, and that the more compact notation is better suited to the mathematical structure of the regional framework. To make this clearer to the reader, we have revised the text in the Methods section to briefly explain our rationale for choosing a concise notation (line 366).

We also define the shaking variable l , instead of IM , and call l_k^s and l_k^n the shaking measures affecting structural and non-structural damage of building k . We use the more

concise notation I rather than IM to improve the readability of expressions in high-dimensional regional models, where large sets of correlated variables are simulated.

Comment 14: Should the left hand term of Equation 2a contain the i term as $p_i(d, i)$? The formulation on the right hand side would suggest it should be independent of the intensity and listed simple as $p_i(d)$

Response 14: We included the term i on the left-hand side of Equation 2a to indicate that $p(d, i)$ represents the joint probability distribution of damage and ground shaking intensity. The equation follows the standard chain rule: $p(d, i) = p(d | i) \cdot p(i)$, where $p(d | i)$ corresponds to the fragility functions used in regional risk modeling. This formulation does not assume independence; rather, it captures the dependence of damage on intensity. The purpose is to illustrate how fragility functions are defined and how spatial correlation in damage arises from the underlying correlated ground motions. We have clarified this point in the revised manuscript in the Methods section (line 380).

This formulation follows the chain rule of probability, where Equation 2a represents the joint distribution of damage and intensity. Equation 2b assumes that, conditional on shaking intensity, the damage at each building is independent—an assumption that simplifies regional modeling. This allows the conditional joint distribution $\pi(d | i)$ to be expressed as the product of per-building fragility functions. While this assumes no direct interaction between buildings once shaking is known, the spatial correlation of ground motion in $\pi(i)$ introduces dependence across damage outcomes.

Comment 15: The authors mention that the damage estimation in Equation 2b assumes independence. Is this realistic? Are there cases where this assumption may not hold? Are there alternative methods that could be adopted? Please comment.

Response 15: We thank the reviewer for raising this important point. We carefully considered the independence assumption when developing our model. Because hospitals in our framework are assumed to lose functionality at early damage stages (e.g., slight damage), we believe it is reasonable to assume conditional independence between structural and non-structural damage—since interaction effects are likely minimal before major degradation occurs. While we are not aware of empirical studies that directly validate this assumption, it aligns with observations from past events where hospitals were disrupted even at low damage states. We have added a brief explanation to the manuscript to clarify this point and note that future work could explore how correlated damage mechanisms may affect functionality, particularly at later stages (line 393).

Although non-structural damage may not be conditionally independent of structural damage—given that structural damage can alter the dynamic properties of a building and, in turn, affect demands on acceleration-sensitive non-structural components—we consider the independence assumption to be reasonable within the failure space of

hospitals. Specifically, our model assumes that hospital functionality is lost once early damage thresholds (e.g., slight damage) are exceeded; at this stage, structural degradation is unlikely to significantly modify the system’s dynamic characteristics. While we are not aware of empirical studies explicitly validating this hypothesis, the assumption allows us to simplify the model without compromising its ability to capture key failure mechanisms relevant to hospital functionality.

Comment 16: Figure 6. The term D_1^s appears twice. One of them should be D_1^n .

Response 16: We fixed it. Thank you for catching this. See the iterated figure (D_1^n appears next to D_1^s)

Comment 17: Again, the authors use the term “state of the art” when referring to a ground motion model developed over a decade ago.

Response 17: We removed state-of-the-art. The iterated text says (line 475):

We utilized a ground motion model for shallow crustal earthquakes to estimate the medians and logarithmic standard deviations of i .

Additional major edits: As mentioned earlier, we made substantial revisions to the computational analysis in response to feedback from the review process. Most notably, we incorporated transportation infrastructure failures into the modeling framework. This enhancement allowed us to better assess post-earthquake hospital accessibility and identify specific hospitals and communities that could become isolated due to bridge or road damage. These changes significantly improved the realism and scope of the study. A summary of the updates and corresponding modifications to the manuscript is provided below.

Abstract (line 15):

By integrating seismic hazard with hospital and transportation infrastructure's vulnerability and connectivity data, we analyze 76 hospitals (426 buildings with 16,639 beds) and 5,163 bridges within a vast network of ~1.5 million edges and ~0.5 million nodes [...] Widespread transportation failures further restrict access, increasing regional travel times by 177% and exceeding 1000% in parts of East Bay, potentially fully isolating hospitals and entire urban communities. These findings underscore the urgent need for resilient healthcare and transportation infrastructure to mitigate life-threatening disruptions following major earthquakes.

Introduction (line 80):

To fill this critical gap, we investigate, for the first time, joint failures of healthcare and transportation networks across the San Francisco Bay Area after a major earthquake. By examining diverse urban communities near active seismic faults—often underrepresented in previous studies—this research provides valuable insights into post-earthquake healthcare access, particularly relevant for earthquake-prone regions worldwide with diverse populations and similarly vulnerable hospitals and bridges, such as those in the U.S., Japan, and Türkiye.

To our knowledge, this is the first study to evaluate post-earthquake healthcare access at this scale using publicly available data on both hospital and transportation infrastructure. Whereas prior work has focused on individual facilities[20-23] or small systems[42-46]—often relying on synthetic data—our approach couples regional seismic risk models with large-scale network analysis to capture interdependent failures across an entire metropolitan area. By integrating vulnerability data from 426 hospital buildings and 5,163 bridges, and simulating disruptions across a transportation network with approximately 1.5 million edges and 0.5 million nodes, we offer one of the most extensive assessments to date of how earthquakes affect healthcare access. This study advances the state of the art in regional risk modeling globally, offering a scalable, computationally intensive framework that supports risk-informed decision-making in other high-risk urban areas.

New Section on Post-earthquake Bridge Functionality (line 196):

Our risk analysis model predicts that 1,469 bridges out of 5,163 would be damaged due to the earthquake scenario (see Methods). Consequently, 3,693 bridges (72%, standard deviation: 17%) are expected to remain undamaged and fully operational (Figure 3). Some damaged bridges could continue operating, albeit at reduced capacities. Under our “baseline scenario”, where only bridges with at most slight damage remain operational, a total of 3,964 bridges (77%, standard deviation: 15%) would function. In a more optimistic “favorable scenario”, where bridges with at most moderate damage also remain partially operational, 4,069 bridges (79%, standard deviation: 14%) could provide service. In an idealistic scenario, which assumes that

Editorial Note: The map data in figure 3 of this Peer Review file is from © OpenStreetMap contributors openstreetmap.org/copyright.

bridges even with extensive damage retain emergency functionality, 4,527 bridges (88%, standard deviation: 11%) would remain operational.

Similar to hospitals, Alameda County faces the most severe transportation impacts (Figure 3, Table 2). Under the baseline scenario, Alameda retains only 44% functional bridges (from 642 down to 282). Marin County is second most impacted, retaining 64% (from 195 to 125 bridges). Additionally, estimates of reductions in bridge travel speeds (see Methods)—calculated from their damage probabilities—highlight Alameda’s higher vulnerability due to proximity to the Hayward Fault rupture (Figure 3, Table 2).

Figure 3. [...] Lower left: Probability distribution of fully functional bridges under baseline (slight), favorable (moderate), and idealistic (extensive) damage thresholds. Lower right: Spatial distribution highlighting bridges with post-earthquake travel capacities below 75%. Results generated from 5,000 Monte Carlo simulations.

Revised findings on Accessibility (line 223):

At a more detailed neighborhood scale, these disruptions are even more pronounced (Figure 4). We identified six densely populated zip codes (each exceeding 15,000 residents, labeled #1 to #6 in Figure 4) facing substantial reductions in healthcare access. The most severely impacted is Novato (#1, Marin County), where travel times increase nearly 25-fold, from 7.3 to 185.5 minutes, affecting approximately 18,000 residents. Similarly, a Fremont zip code (#6, Alameda County) with roughly 66,000 residents experiences an eightfold increase (from 8.4 to 42.2 minutes). Other significantly impacted zip codes in Richmond (#4), Oakland (#2 and #5, Alameda County), and San Jose (#3, Santa Clara County) face travel time increases of 10, 11, 9, and 11 times their pre-earthquake levels, respectively. These results underscore the dramatic reshaping of healthcare access throughout East Bay communities along the earthquake rupture.

Finally, we explored the relative contributions of hospital and transportation infrastructure disruptions through scenario analyses (Figure 4). We compared the baseline (None, no infrastructure disruptions) against scenarios modeling probabilistic damage only to transportation infrastructure (Transp., hospitals fully resilient), only to hospital infrastructure (Hosp., transportation fully resilient), and simultaneous probabilistic disruptions to both (Transp. & Hosp.). When considering disruptions only to transportation infrastructure (Transp.), average Bay Area travel times increase by 41% (from 6.1 to 8.6 minutes), notably less than the combined scenario (Transp. & Hosp., 177% increase). Conversely, modeling only hospital infrastructure disruptions (Hosp.) yields a 79% increase (from 6.1 to 10.9 minutes), highlighting greater relative fragility in hospital networks.

At localized urban scales, the compounding effects of simultaneous transportation and hospital infrastructure failures lead to disproportionately severe impacts (Figure 4). For example, as mentioned earlier, in the most severely impacted zip code (Novato, labeled #1 in Figure 4), travel times dramatically increase from 7.3 to 185.5 minutes under combined disruptions (Transp. & Hosp.), compared to only 14.1 minutes when transportation disruptions are excluded (Hosp. only). Similarly, for other highly affected zip codes (labeled #2 to #6 in Figure 4), travel-time ratios notably decrease—from 11, 11, 10, 9, and 8 under combined disruptions, to 6.4, 3.1, 6.1, 4.7, and 3.4 when transportation disruptions are not considered—highlighting the critical influence of compounding infrastructure failures on healthcare access.

Figure 4: Access to functional acute care hospitals in the Bay Area after the earthquake. Upper left: Post-earthquake (Post-EQ) increases in travel time at the zip-code level compared to pre-earthquake (Pre-EQ) conditions, highlighting densely populated areas that are most severely impacted. Upper right: Ratios of post- to pre-earthquake travel times to the closest acute care hospitals. Lower right: Distribution of travel times per zip code under four scenarios: None (no infrastructure disruptions), Transp. (only transportation infrastructure is vulnerable), Hosp. (only hospital infrastructure is vulnerable), and Transp. & Hosp. (both transportation and hospital infrastructures are vulnerable). Lower left: Ratios of post- to pre-earthquake travel times when only hospital infrastructure (Hosp.) is vulnerable. Results generated from 5,000 Monte Carlo simulations.

New section on Isolated Communities and Hospitals (line 252):

To understand the drivers behind the sharp increases in travel times observed earlier, we analyzed travel volumes to the closest acute care hospitals before and after the earthquake. Figure 5 shows pre-earthquake road usage, with line thickness representing travel volume. Roads in brown highlight those whose usage falls above the 90th percentile based on pre-earthquake conditions, emphasizing the most critical travel routes. As expected, the highest-usage roads prior to the earthquake were located near hospitals, especially in densely populated areas with fewer hospitals. For instance, communities in Fremont (#6 in Figures 4 and 5) are served by only two hospitals, while Richmond (#4) relies on just one.

Figure 5 also displays post-earthquake road usage, using the same pre-earthquake 90th percentile threshold to highlight heavily traveled roads. After the earthquake, patient travel patterns shift significantly, with many communities forced to travel longer distances along major corridors. For example, residents in Oakland (#2 and #5 in Figures 4 and 5), who previously relied on nearby hospitals, will now have to travel west to the San Francisco Peninsula or east to Walnut Creek due to simultaneous disruptions in both hospital and transportation infrastructure. These shifts help explain the steep increases in travel times observed in East Bay communities along the rupture zone.

Four major bridges connect the East and West Bay: the Richmond–San Rafael, Oakland–San Francisco Bay, San Mateo–Hayward, and Dumbarton bridges. Before the earthquake, no patients needed to cross these bridges to reach their nearest acute care hospital (Figure 5). After the earthquake, however, these bridges become critical, with travel volumes increasing to 3.6, 9.3, 6.3, and 13.6 times the pre-earthquake 90th percentile values—reflecting the large-scale redistribution of healthcare demand.

Our analysis also revealed a more complex post-disaster phenomenon: micro-scale isolation of hospitals and communities due to localized bridge failures. For example, the hospital serving the most severely impacted zip code in Novato (#1 in Figures 4 and 5), which has 64 beds, depends entirely on a single bridge for access. If this bridge fails, the hospital becomes inaccessible, increasing travel times from 7.3 to 185.5 minutes for the surrounding population.

More concerning, we found that urban neighborhoods near Fremont (#6) could become fully isolated (Figure 5). In one area, failure of three bridges would trap one part of the neighborhood; in another, a different set of three bridge failures would isolate the remaining part. These communities would lose access not only to hospitals but also to other essential services such as grocery stores and pharmacies—illustrating how cascading infrastructure failures can sever lifeline access in disaster scenarios.

Editorial Note: The map data in figure 5 of this Peer Review file is from © OpenStreetMap contributors openstreetmap.org/copyright.

Figure 5: Mobility to reach the closest functional acute care hospital in the San Francisco Bay Area. Upper left: Pre-earthquake travel volumes to the closest acute care hospitals. Numbers 1 to 6 indicate the hospitals serving the dense zip codes with the highest post-earthquake increases in travel time. Upper right: Projected travel volumes after the earthquake, showing major shifts in patient mobility. The red-shaded corridor illustrates how Oakland residents (#2 and #5) must travel west to the Peninsula or east to Walnut Creek due to hospital and transportation disruptions. Lower left: The hospital serving zip code #1 (Novato) relies entirely on a single bridge; if the bridge fails, all access to its 64 beds is lost. Lower right: Community near zip code #6 (Fremont), where the southern and northern sections become fully isolated if three different bridges fail in each area. Results are based on 5,000 Monte Carlo simulations of post-earthquake patient mobility.

Reviewer #1

Comment 1: This manuscript presents a numerical investigation into the anticipated hospital capacity and patient travel times in the San Francisco Bay Area, California, following a simulated M7.25 earthquake. The study focuses on expected travel times during the earthquake's recovery phase. A key innovation lies in integrating a risk model with a transportation network to elucidate potential cascading effects. The results demonstrate that the fragility of both healthcare facilities and transportation infrastructure significantly contributes to the probable elongation of patient travel times.

Compared to a previous iteration, this manuscript has undergone a comprehensive revision. Several critical assumptions employed in the earlier version have been relaxed, and the impact of the remaining assumptions is thoroughly discussed. A limitation of the study is the use of relatively generic vulnerability models to estimate the probability of specific damage levels for each hospital, given the expected ground shaking. While individual hospital buildings may exhibit fragility characteristics that deviate from these generic models, this approach provides a reasonable approximation considering the large number of buildings considered. Overall, the manuscript is suitable for publication in Nature Communications.

Response 1: We sincerely thank the reviewer for their thoughtful and constructive comments throughout the revision process. Their feedback was instrumental in improving the quality of our manuscript. We agree that incorporating more refined structural models would enhance the fidelity of our regional analysis. Over the coming years, our research group will focus on developing computational methods that scale structurally detailed models to regional levels, as well as collecting building-specific data to support the construction of these models. We are grateful for the reviewer's insights and support.

Comment 2: The code reaches 3.4GB and exceeds the 2GB transfer limit set by the server. It is not practical to download and try to run it on the local computer. It would be much more convenient if the file size was reduced.

Response 2: DesignSafe allows for downloading each file individually (see screenshot below). That way, other researchers can download the scripts and files they are interested in without exceeding transfer limits. We added a line in the Code and Data Availability Statements pointing to this. Thank you for catching it.

[Figure Redacted]

Reviewer #2:

Comment 1: I had the chance to reread the paper and review the author's responses to my original comment. I appreciate how the authors have elaborated on their work and clarified their contribution, and this manuscript is an improved version of what they submitted originally. For example, adding bridge failure analysis is a significant improvement. The authors also provided various clarifications on issues. They explained different confusing elements, including, for example, that their intent was not to use a value of 0.2 PGA as an indicator of significant damage to the buildings, and how a probability of 0.25 was not used as a cut-off for damage. Similarly, the use of HAZSU fragilities and their adjustment according to the SPC/NPC mapping provided a clearer reflection on the actions taken to obtain damage fragilities. However, it is essential to note that the validity of HAZSU facilities has been a subject of extensive discussions within the earthquake community. It is good to see, however, that they have been adjusted based on some logic.

While these improvements helped the paper in general, I still consider the analysis simplistic in many ways, and I struggled to see the novelty of the work. As a case study, the findings are indeed important and relevant to the city of San Francisco. However, I remain unconvinced about the novelty of the work and the study's motivation. Integrating transportation infrastructure into the modeling of post-earthquake hospital accessibility has been done numerous times, and publications with a similar motive appear in many journals worldwide, not only for seismic applications but also for almost all other natural hazards. The authors further elaborated on the study's motivation, noting that the San Francisco Bay area's uniqueness is characterized by a high concentration of seismically vulnerable hospitals, a complex and interconnected healthcare network that spans county and jurisdictional boundaries, and spatially heterogeneous seismic hazards and vulnerabilities across communities. These three elements are not unique to San Francisco. In fact, most major cities around the world exhibit similar characteristics, including a complex network of aging infrastructure. So, in my opinion, the argued uniqueness represents a form of baseline risk environment rather than a novel or case-specific anomaly. The authors further noted that the uniqueness is in that "this is the first study to evaluate the performance of a realistic regional healthcare network using publicly available, high-resolution data on both hospital infrastructure and transportation systems.". As I noted, numerous studies have investigated accessibility to healthcare on a large scale, which limits the uniqueness of your study (please see some examples below). Of course, the studies listed below are not "exactly" similar to yours in any way. Still, the overall idea of examining accessibility to healthcare systems on a large scale remains the same.

Post-earthquake health care service accessibility assessment framework and its application in a medium-sized city: 10.1016/j.res.2022.108782

Accessibility of medical services following an earthquake: A case study of traffic and economic aspects affecting the Istanbul

roadway: <https://www.sciencedirect.com/science/article/abs/pii/S2212420918305314>

Measuring spatiotemporal accessibility to healthcare with multimodal transport modes in the dynamic traffic environment: <https://www.degruyterbrill.com/document/doi/10.1515/geo-2022-0461/html?lang=en>

I appreciated the elaboration on the intent of the paper by the authors and I also recognize the computational demand in integrating staff, stuff, and supplies in calculating bed availability at a single room level, which is the state of the art, as opposed to your approach where you assumed hospital damage as a proxy for bed availability. A better approximation would have been to perform that integration at the hospital level, rather than at the single-room level. Not only would this be computationally reasonable, but it would also give you a much better approximation of bed availability. Similarly, modeling building functionality as a Bernoulli process and not considering the state-of-the-art that integrates various elements to determine the functionality level adds to the uncertainty of the results. This gross approximation, combined with the hospital bed approximation, increases the uncertainty level in the results presented, rendering them somewhat questionable.

Response 1: We thank the reviewer for their thoughtful comments and for engaging deeply with our manuscript throughout all rounds of review. We carefully reviewed the suggested references and have incorporated two of them into the revised manuscript: the case study in Istanbul involving eight hospitals [1], and the simulation-based study in a medium-sized Chinese city [2]. We did not include the third reference, as it does not focus on earthquake-related disruptions, which are central to our study of post-earthquake healthcare accessibility. Nonetheless, we appreciate the recommendation and found it valuable for understanding broader methodological approaches.

In response to the reviewer's concerns about the framing of our study's novelty, we have revised the Introduction substantially. We have removed the initial claims regarding the uniqueness of our study and, instead, contextualize our contribution more carefully within the existing literature. We highlight how our study complements prior work while emphasizing the specific regional characteristics and data-driven elements of our San Francisco Bay Area analysis. Our revised framing focuses on the practical implications and transferability of our modeling framework rather than its singularity.

We also appreciate the reviewer's comments on structural modeling fidelity and acknowledge the limitations of using generic fragility functions and simplified assumptions (e.g., Bernoulli process for functionality). We agree that better approximations for hospital bed availability—such as integrating hospital-level data—would improve model realism and reduce uncertainty. Moving forward, our group is actively pursuing research efforts that aim to (1) scale structurally detailed models to regional levels, and (2) collect and integrate building-specific and hospital operational data to improve future analyses.

We are grateful for the reviewer's insights, which have helped us strengthen both the framing and technical direction of our study.

[1] Post-earthquake health care service accessibility assessment framework and its application in a medium-sized city: 10.1016/j.ress.2022.108782

[2] Accessibility of medical services following an earthquake: A case study of traffic and economic aspects affecting the Istanbul

roadway: <https://www.sciencedirect.com/science/article/abs/pii/S2212420918305314>

New part of the Introduction:

Past research has examined hospital seismic risk primarily at the building scale. Studies have used structural engineering methods, fault-tree analyses, and performance-based assessments to evaluate damage in isolated facilities.^{23–26} Others have used flow models and discrete-event simulations to explore patient care delays and medical bottlenecks in single-hospital scenarios.^{27–30} While valuable, these studies often overlook how failures can propagate across urban systems.

In parallel, a growing body of work has started to model access to hospital services at the regional scale. Studies in Lima, Peru,^{31, 32} and Butte County, California,³³ have shown that even small reductions in overall healthcare capacity can significantly affect local access to care. More recent efforts have begun to jointly consider disruptions to both healthcare and transportation systems, e.g., in Istanbul, Türkiye,³⁴ and in a synthetic medium-sized city in China.³⁵ We contribute to this growing literature by modeling these compounding effects at scale across the San Francisco Bay Area using detailed, real-world datasets on hospital and bridge infrastructure.

Recent earthquakes in Chile, New Zealand, and Japan have also underscored the importance of treating hospitals as part of broader interdependent systems.^{24, 36–39} Failures at one facility can cascade across a regional network—overwhelming nearby hospitals, increasing patient travel distances, and worsening health outcomes. To support resilience planning across large metropolitan areas, there is a need for region-wide models grounded in real exposure data that capture operational interdependencies between healthcare and transportation infrastructure.^{40–42} Developing such models remains challenging due to limited data, computational demands, and regional variation in infrastructure.

In this work, we evaluate post-earthquake healthcare access across the entire San Francisco Bay Area by simulating joint disruptions in healthcare and transportation systems. We integrate seismic risk models with regional-scale network analysis using publicly available data on 426 hospital buildings and 5,163 bridges across a transportation network comprising approximately 1.5 million edges and 0.5 million nodes. Our study contributes to the growing field of integrated

disaster risk modeling by providing one of the most extensive simulations to date of interdependent failures affecting healthcare access after an earthquake. Although grounded in the San Francisco Bay Area, our study highlights systemic barriers to accessing healthcare after earthquakes under conditions common to many dense urban regions—such as high seismic risk, aging infrastructure, complex emergency response systems, and population disparities. These findings underscore the need for integrated planning approaches that account for infrastructure interdependencies and can inform resilience policy in similarly exposed metropolitan areas

Reviewer #3:

Comment 1: The authors have addressed all comments by this reviewer

Response 1: The authors thank reviewer #3 for all the comments that helped us make the paper better.